# GRADIENT PLAY IN STOCHASTIC GAMES: STATIONARY POINTS, CONVERGENCE, AND SAMPLE COMPLEXITY

## ABSTRACT

We study the performance of the gradient play algorithm for stochastic games (SGs), where each agent tries to maximize its own total discounted reward by making decisions *independently* based on current state information which is shared between agents. Policies are directly parameterized by the probability of choosing a certain action at a given state. We show that Nash equilibria (NEs) and first-order stationary policies are equivalent in this setting, and give a local convergence rate around strict NEs. Further, for a subclass of SGs called Markov potential games (which includes the cooperative setting with identical rewards among agents as an important special case), we design a sample-based reinforcement learning algorithm and give a non-asymptotic global convergence rate analysis for both exact gradient play and our sample-based learning algorithm. Our result shows that the number of iterations to reach an $\epsilon$-NE scales linearly, instead of exponentially, with the number of agents. Local geometry and local stability are also considered, where we prove that strict NEs are local maxima of the total potential function and fully-mixed NEs are saddle points.

## 1 INTRODUCTION

Multi-agent systems find applications in a wide range of societal systems, e.g. electric grid, traffic networks, smart building and smart cities etc. Given the complexity of these systems, multi-agent reinforcement learning (MARL) has gained increasing attention in recent years (Daneshfar & Bevrani, 2010; Shalev-Shwartz et al., 2016; Vidhate & Kulkarni, 2017; Xu et al., 2020). Among MARL algorithms, policy gradient-type methods are highly popular because of their flexibility and capability to incorporate structured state and action spaces. However, while many recent works (Zhang et al., 2018; Chen et al., 2018; Wai et al., 2018; Li et al., 2019; Qu et al., 2020) have studied the sample complexity of multi-agent policy gradient algorithms, due to a lack of understanding of the optimization landscape in these multi-agent learning problems, most works can only show convergence to a first-order stationary point. Deeper understanding of the quality of these stationary points is missing even in the simple identical-reward multi-agent RL setting.

In this paper, we examine this problem from a game-theoretic perspective. We model the multi-agent system as a stochastic game (SG) where agents can have different reward functions, and study the dynamical behavior of first-order (gradient-based) learning methods. The study of SGs dates back to as early as the 1950s by Shapley (1953) with a series of followup works on developing NE-seeking algorithms, especially in the RL setting (e.g. (Littman, 1994; Bowling & Veloso, 2000; Shoham et al., 2003; Buşoniu et al., 2010; Lanctot et al., 2017; Zhang et al., 2019a) and citations within). While well-known classical algorithms for solving SGs are mostly value-based, such as Nash-Q learning (Hu & Wellman, 2003), Hyper-Q learning (Tesauro, 2003), and WoLF-PHC (Bowling & Veloso, 2001), gradient-based algorithms have also started to gain popularity in recent years due to their advantages as mentioned earlier (e.g. (Abdallah & Lesser, 2008; Zhang & Lesser, 2010; Foerster et al., 2017)). In this work, we aim to gain a deeper understanding of the structure and quality of first-order stationary points for these gradient-based methods, with a particular focus on answering the following questions: 1) How do the first-order stationary points relate to the NEs of the underlying game?, 2) Do gradient-based algorithms guarantee convergence to a NE?, 3) What is the stability of the individual NEs?, and 4) How should agents learn using local samples from the environment?.

These questions have already been widely discussed in other settings, e.g., one-shot (stateless) finite-action games (Shapley, 1964; Crawford, 1985; Jordan, 1993; Krishna & Sjöström, 1998; Shamma &

Arslan, 2005; Kohlberg & Mertens, 1986; Van Damme, 1991), one-shot continuous games (Mazumdar et al., 2020), zero-sum linear quadratic (LQ) games Zhang et al. (2019b), etc. There are both negative and positive results depending on the settings. For one-shot continuous games, (Mazumdar et al., 2020) proved a negative result suggesting that gradient flow has stationary points (even local maxima) that are not necessarily NEs. Conversely, Zhang et al. (2019b) designed projected nested-gradient methods that provably converge to NEs in zero-sum LQ games. However, much less is known in the tabular setting of SGs with finite state-action spaces.

**Contributions.** In our paper, we consider the *gradient play* algorithm for the infinite time-discounted reward SGs where an agent's individual policy is directly parameterized by the probability of choosing an action from the agent's own action space at a given state. We focus on the tabular setting where state and action spaces are finite. Through generalizing the gradient domination property in (Agarwal et al., 2020) to the multi-agent setting, we first establish the equivalence of first-order stationary policies and Nash equilibria. We then show that strict NEs are locally asymptotically stable under gradient play and provide a local convergence rate analysis.

Additionally, we study the global convergence for a special class of SGs called Markov potential games (MPGs) (González-Sánchez & Hernández-Lerma, 2013; Macua et al., 2018; Leonardos et al., 2021), which includes identical reward multi-agent RL (Tan, 1993; Claus & Boutilier, 1998; Panait & Luke, 2005) as an important special case. In this setting, we first show that exact gradient play can find an $\epsilon$-Nash equilibrium within $O\left(\frac{|\mathcal{S}|\sum_i|\mathcal{A}_i|}{\epsilon^2}\right)$ steps. Then, we design a sample-based gradient play algorithm and show that it can find an $\epsilon$-Nash equilibrium with high probability in a fully-decentralized manner using $\widetilde{O}\left(\frac{n}{\epsilon^6}\text{poly}\left(\frac{1}{1-\gamma}, |\mathcal{S}|, \max_i|\mathcal{A}_i|\right)\right)$ samples, where $|\mathcal{S}|, |\mathcal{A}_i|$ denote the size of the state space and action space of agent $i$ respectively. The convergence rate shows that the number of iterations to reach an $\epsilon$-NE scales linearly with the number of agents. In the sample-based learning, agents only need to observe the state, their own actions, and their own rewards. The key enabler of the learning is the existence of an underlying *averaged MDP* for each agent when other agents' policies are fixed. Our learning method can be viewed as a *model-based policy evaluation* method with respect to agents' averaged MDPs. This averaged MDP concept could be applied to design many other MARL algorithms, especially policy-evaluation-based methods. We also study the local geometry around some special classes of equilibrium points, showing that strict NEs are local maxima of the total potential function and that fully mixed NEs are saddle points. Lastly, all the algorithms studied in this paper have been numerical tested and results are provided in Appendix A.

**Comparison to other MARL algorithms:** For MPGs with continuous state and action spaces, there are studies about learning either the open-loop (González-Sánchez & Hernández-Lerma, 2013; Zazo et al., 2016) or the closed-loop (Macua et al., 2018) NEs for MPGs. These works generally assume full model information and solve the problem via optimal control. There are two recent arXiv preprints (Mguni, 2020; Leonardos et al., 2021; Mguni et al., 2021) studying MPGs that are similar to our MPG setting. In particular, Leonardos et al. (2021) also studies gradient play for MPG. Both of our papers share similar results on MPGs but the sample-based methods are designed from different perspectives.[1] In addition, our papers studies general SGs besides MPG. Moreover, our concept of "averaged" MDPs could also serve as a useful tool for the design and analysis of other MARL algorithms. Beyond these MPG works, decentralized Q-learning introduced in Arslan & Yüksel (2016) might be the closest to the setting considered in this paper. They consider the identical interest case and only show asymptotic convergence to the set of NEs. There are other recent works that also study learning for general-sum or zero-sum stochastic games. However the settings they consider are different from our setting, for example, Daskalakis et al. (2021) considers convergence to NE for two player zero-sum games, while Song et al. (2021) considers convergence to correlated equilibrium for finite time horizon general-sum games. On the other hand, Zhang et al. (2018); Li et al. (2019); Qu et al. (2019) consider slightly different MARL settings, where agents collaboratively maximize the summation of agents' reward with either full or partial state observation. They also require communication between neighboring agents for a better global coordination.

---

[1]Leonardos et al. (2021) considers Monte Carlo, model-free gradient estimation. The sample complexity is derived under the condition that the estimation is unbiased, which is difficult to hold in general. Interestingly, both sample complexities are $O(\frac{1}{\epsilon^6})$. It is an interesting question to study whether such dependence is fundamental or not. We also remark that Leonardos et al. (2021) and this work are done in parallel.

## 2 PROBLEM SETTING AND PRELIMINARIES

We consider a stochastic game (SG) $\mathcal{M} = (N, \mathcal{S}, \mathcal{A} = \mathcal{A}_1 \times \cdots \times \mathcal{A}_n, P, r = (r_1, \ldots, r_n), \gamma, \rho)$ with $n$ agents (Shapley, 1953) which is specified by an agent set $N = \{1, 2, \ldots, n\}$, a finite state space $\mathcal{S}$, a finite action space $\mathcal{A}_i$ for each agent $i \in N$, a transition model $P$ where $P(s'|s, a) = P(s'|s, a_1, \ldots, a_n)$ is the probability of transitioning into state $s'$ upon taking action $a := (a_1, \ldots, a_n)$ in state $s$ where $a_i \in \mathcal{A}_i$ is action of agent $i$, agent $i$'s reward function $r_i : \mathcal{S} \times \mathcal{A} \to [0, 1]$, a discount factor $\gamma \in [0, 1)$, and an initial state distribution $\rho$ over $\mathcal{S}$.

A stochastic policy $\pi : \mathcal{S} \to \Delta(\mathcal{A})$ (where $\Delta(\mathcal{A})$ is the probability simplex over $\mathcal{A}$) specifies a strategy in which agents choose their actions *jointly* based on the current state in a stochastic fashion, i.e. $\Pr(a_t|s_t) = \pi(a_t|s_t)$. A distributed stochastic policy is a special subclass of stochastic policies, with $\pi = \pi_1 \times \ldots \times \pi_n$, where $\pi_i : \mathcal{S} \to \Delta(\mathcal{A}_i)$. For distributed stochastic policies, each agent takes its action based on the current state $s$ *independently* of other agents' choices of actions, i.e.:

$$\Pr(a_t|s_t) = \pi(a_t|s_t) = \prod_{i=1}^n \pi_i(a_{i,t}|s_t), \quad a_t = (a_{1,t}, \ldots, a_{n,t}).$$

For notational simplicity, we define: $\pi_I(a_I|s) := \prod_{i \in I} \pi_i(a_i|s)$, where $I \subseteq N$ is an index set. Further, we use the notation $-i$ to denote the index set $N \backslash \{i\}$.

We consider *direct distributed policy parameterization*, where agent $i$'s policy is parameterized by $\theta_i$:

$$\pi_{i,\theta_i}(a_i|s) = \theta_{i,(s,a_i)}, \quad i = 1, 2, \ldots, n. \tag{1}$$

For notational simplicity, we abbreviate $\pi_{i,\theta_i}(a_i|s)$ as $\pi_{\theta_i}(a_i|s)$, and $\theta_{i,(s,a_i)}$ as $\theta_{s,a_i}$. Here $\theta_i \in \Delta(\mathcal{A}_i)^{|\mathcal{S}|}$, i.e. $\theta_i$ is subject to the constraints $\theta_{s,a_i} \geq 0$ and $\sum_{a_i \in \mathcal{A}_i} \theta_{s,a_i} = 1$ for all $s \in \mathcal{S}$. The global joint policy is given by: $\pi_\theta(a|s) = \prod_{i=1}^n \pi_{\theta_i}(a_i|s) = \prod_{i=1}^n \theta_{s,a_i}$. We use $\mathcal{X}_i := \Delta(\mathcal{A}_i)^{|\mathcal{S}|}$, $\mathcal{X} := \mathcal{X}_1 \times \cdots \times \mathcal{X}_n$ to denote the feasible region of $\theta_i$ and $\theta$.

Agent $i$'s value function $V_i^\theta : \mathcal{S} \to \mathbb{R}, i \in N$ is defined as the discounted sum of future rewards starting at state $s$ via executing $\pi_\theta$, i.e.

$$V_i^\theta(s) := \mathbb{E}\left[\sum_{t=0}^\infty \gamma^t r_i(s_t, a_t) \Big| \pi_\theta, s_0 = s\right],$$

where the expectation is with respect to the random trajectory $\tau = (s_t, a_t, r_{i,t})_{t=0}^\infty$ where $a_t \sim \pi_\theta(\cdot|s_t), s_{t+1} = P(\cdot|s_t, a_t)$. We denote agent $i$'s total reward starting from initial state $s_0 \sim \rho$ as:

$$J_i(\theta) = J_i(\theta_1, \ldots, \theta_n) := \mathbb{E}_{s_0 \sim \rho} V_i^\theta(s_0).$$

In the game setting, Nash equilibrium is often used to characterize the performance of agents' policies.

**Definition 1.** *(Nash equilibrium) A policy $\theta^* = (\theta_1^*, \ldots, \theta_n^*)$ is called a Nash equilibrium (NE) if*

$$J_i(\theta_i^*, \theta_{-i}^*) \geq J_i(\theta_i', \theta_{-i}^*), \quad \forall \theta_i' \in \mathcal{X}_i, \quad i \in N$$

*The equilibrium is called a strict NE if the inequality holds strictly for all $\theta_i' \in \mathcal{X}_i$ and $i \in N$. The equilibrium is called a pure NE if $\theta^*$ corresponds to a deterministic policy. The equilibrium is called a mixed NE if it is not pure. Further, the equilibrium is called a fully mixed NE if every entry of $\theta^*$ is strictly positive, i.e.: $\theta_{s,a_i}^* > 0, \ \forall a_i \in \mathcal{A}_i, \ \forall s \in \mathcal{S}, \ i \in N$*

We define the *discounted state visitation distribution* $d_\theta$ of a policy $\pi_\theta$ given an initial state distribution $\rho$ as:

$$d_\theta(s) := \mathbb{E}_{s_0 \sim \rho}(1 - \gamma) \sum_{t=0}^\infty \gamma^t \Pr^\theta(s_t = s|s_0), \tag{2}$$

where $\Pr^\theta(s_t = s|s_0)$ is the state visitation probability that $s_t = s$ when executing $\pi_\theta$ starting at state $s_0$. Throughout the paper, we make the following assumption on the SGs we study.

**Assumption 1.** *The stochastic game $\mathcal{M}$ satisfies: $d_\theta(s) > 0, \ \forall s \in \mathcal{S}, \ \forall \theta \in \mathcal{X}$.*

Assumption 1 requires that every state is visited with positive probability, which is a standard assumption for convergence proofs in the RL literature (e.g. (Agarwal et al., 2020; Mei et al., 2020)).

Similar to centralized RL, we define agent $i$'s Q-function $Q_i^\theta : \mathcal{S} \times \mathcal{A} \to \mathbb{R}$ and its advantage function $A_i^\theta : \mathcal{S} \times \mathcal{A} \to \mathbb{R}$ as:

$$Q_i^\theta(s, a) := \mathbb{E}\left[ \sum_{t=0}^\infty \gamma^t r_i(s_t, a_t) \Big| \pi_\theta, s_0 = s, a_0 = a \right], \quad A_i^\theta(s, a) := Q_i^\theta(s, a) - V_i^\theta(s).$$

**'Averaged' Markov decision process (MDP):** We further define agent $i$'s *'averaged' Q-function* $\overline{Q_i^\theta} : \mathcal{S} \times \mathcal{A}_i \to \mathbb{R}$ and *'averaged' advantage-function* $\overline{A_i^\theta} : \mathcal{S} \times \mathcal{A}_i \to \mathbb{R}$ as:

$$\overline{Q_i^\theta}(s, a_i) := \sum_{a_{-i}} \pi_{\theta_{-i}}(a_{-i}|s) Q_i^\theta(s, a_i, a_{-i}), \quad \overline{A_i^\theta}(s, a_i) := \sum_{a_{-i}} \pi_{\theta_{-i}}(a_{-i}|s) A_i^\theta(s, a_i, a_{-i}). \quad (3)$$

Similarly, we define agent $i$'s *'averaged' transition probability distribution* $\overline{P_i^\theta} : \mathcal{S} \times \mathcal{S} \times \mathcal{A}_i \to \mathbb{R}$, and *'averaged' reward* $\overline{r_i^\theta} : \mathcal{S} \times \mathcal{A}_i \to \mathbb{R}$ as:

$$\overline{P_i^\theta}(s'|s, a_i) := \sum_{a_{-i}} \pi_{\theta-i}(a_{-i}|s) P(s'|s, a_i, a_{-i}), \quad \overline{r_i^\theta}(s, a_i) := \sum_{a_{-i}} \pi_{\theta-i}(a_{-i}|s) r_i(s, a_i, a_{-i})$$

From its definition, the averaged Q-function satisfies the following Bellman equation:

**Lemma 1.** $\overline{Q_i^\theta}$ *satisfies:* $\quad \overline{Q_i^\theta}(s, a_i) = \overline{r_i^\theta}(s, a_i) + \gamma \sum_{s', a_i'} \pi_{\theta_i}(a_i'|s') \overline{P_i^\theta}(s'|s, a_i) \overline{Q_i^\theta}(s', a_i') \quad (4)$

Lemma 1 suggests that the averaged Q-function $\overline{Q_i^\theta}$ is indeed the Q-function for the MDP defined on action space $\mathcal{A}_i$, with $\overline{r_i^\theta}, \overline{P_i^\theta}$ as its stage reward and transition probability respectively. We define this MDP as the *'averaged' MDP* of agent $i$, i.e., $\mathcal{M}_i^\theta = (\mathcal{S}, \mathcal{A}_i, \overline{P_i^\theta}, \overline{r_i^\theta}, \gamma, \rho)$. The notion of an 'averaged' MDP will serve as an important intuition when designing the sample-based algorithm. Note that the 'averaged' MDP is only well-defined when the policies of the other agents $\theta_{-i}$ are kept fixed. When this is indeed the case, agent $i$ can be treated as an independent learner with respect to its own 'averaged' MDP. Thus, various classical policy evaluation RL algorithms can then be applied.

## 3 EXACT GRADIENT PLAY FOR GENERAL STOCHASTIC GAMES

Under direct distributed parameterization, the gradient play algorithm is given by:

$$\textit{Exact Gradient Play:} \qquad \theta_i^{(t+1)} = Proj_{\mathcal{X}_i}(\theta_i^{(t)} + \eta \nabla_{\theta_i} J_i(\theta_i^{(t)})), \quad \eta > 0. \quad (5)$$

Gradient play can be viewed as a 'better response' strategy, where agents update their own parameters by gradient ascent with respect to their own rewards. A first-order stationary point is defined as such:

**Definition 2.** *(First-order stationary policy) A policy $\theta^* = (\theta_1^*, \ldots, \theta_n^*)$ is called a first-order stationary policy if $(\theta_i' - \theta_i^*)^\top \nabla_{\theta_i} J_i(\theta^*) \le 0, \; \forall \theta_i' \in \mathcal{X}_i, \; i \in N$.*

It is not hard to verify that $\theta^*$ is a first-order stationary policy if and only if it is a fixed point under gradient play (Equation (5)). Comparing Definition 1 (of NE) and Definition 2, we know that NEs are first-order stationary policies, but not necessarily vice versa. For each agent $i$, first-order stationarity does not imply that $\theta_i^*$ is optimal among all possible $\theta_i$ given $\theta_{-i}^*$. However, interestingly, we will show that NEs are equivalent to first-order stationary policies due to a gradient domination property that we will show later. Before that, we first calculate the explicit form of the gradient $\nabla_{\theta_i} J_i$.

Policy gradient theorem (Sutton et al., 1999) gives an efficient formula for the gradient:

$$\nabla_\theta \mathbb{E}_{s_0 \sim \rho} V_i^\theta(s_0) = \frac{1}{1-\gamma} \mathbb{E}_{s \sim d_\theta} \mathbb{E}_{a \sim \pi_\theta(\cdot|s)} [\nabla_\theta \log \pi_\theta(a|s) Q_i^\theta(s, a)], \quad i \in N. \quad (6)$$

Applying Equation (6), the gradient $\nabla_{\theta_i} J_i$ can be written explicitly as follows:

**Lemma 2.** *(Proof in Appendix D) For direct distributed parameterization (Equation (1)),*

$$\frac{\partial J_i(\theta)}{\partial \theta_{s, a_i}} = \frac{1}{1-\gamma} d_\theta(s) \overline{Q_i^\theta}(s, a_i) \quad (7)$$

**Gradient domination and the equivalence between NE and first-order stationary policy.**
Lemma 4.1 in Agarwal et al. (2020) established gradient domination for centralized tabular MDP under direct parameterization. We can show that a similar property still holds for stochastic games.

**Lemma 3.** (Gradient domination, proof in Appendix E.) *For direct distributed parameterization (Equation (1)), we have that for any $\theta = (\theta_1, \ldots, \theta_n) \in \mathcal{X}$:*

$$J_i(\theta_i', \theta_{-i}) - J_i(\theta_i, \theta_{-i}) \leq \left\| \frac{d_{\theta'}}{d_\theta} \right\|_\infty \max_{\overline{\theta}_i \in \mathcal{X}_i} (\overline{\theta}_i - \theta_i)^\top \nabla_{\theta_i} J_i(\theta), \quad \forall \theta_i' \in \mathcal{X}_i, \quad i \in N \qquad (8)$$

*where $\left\| \frac{d_{\theta'}}{d_\theta} \right\|_\infty := \max_s \frac{d_{\theta'}(s)}{d_\theta(s)}$, and $\theta' = (\theta_i', \theta_{-i})$.*

For the single-agent case ($n = 1$), Equation (8) is consistent with the result in Agarwal et al. (2020), i.e.: $J(\theta') - J(\theta) \leq \left\| \frac{d_{\theta'}}{d_\theta} \right\|_\infty \max_{\overline{\theta} \in \mathcal{X}} (\overline{\theta} - \theta)^\top \nabla J(\theta)$. However, when there are multiple agents, the condition is much weaker because the inequality requires $\theta_{-i}$ to be fixed. When $n = 1$, gradient domination rules out the existence of stationary points that are not global optima. For the multi-agent case, the property can no longer guarantee the equivalence between first-order stationarity and global optimality; instead, it links the stationary points with NEs as shown in the next theorem whose proof is in Appendix E.

**Theorem 1.** *Under Assumption 1, first-order stationary policies and NEs are equivalent.*

**Local convergence for strict NEs** Although the equivalence of NEs and stationary points under gradient play has been established, it is in fact difficult to show that gradient play converges to these stationary points. Even in the simpler static (stateless) game setup, gradient play might fail to converge (Shapley, 1964; Crawford, 1985; Jordan, 1993; Krishna & Sjöström, 1998). One major difficulty is that the vector field $\{\nabla_{\theta_i} J_i(\theta)\}_{i=1}^n$ is not a conservative vector field. Accordingly, its dynamics may display complicated behavior. Thus, as a preliminary study, instead of looking at global convergence, we focus on the local convergence and restrict our study to a special subset of NEs - the strict NEs. We begin by giving the following characterization of strict NEs:

**Lemma 4.** *Given a stochastic game $\mathcal{M}$, any strict NE $\theta^*$ is pure, meaning that for each $i$ and $s$, there exist one $a_i^*(s)$ such that $\theta_{s,a_i}^* = \mathbf{1}\{a_i = a_i^*(s)\}$. Additionally,*

$$i) \; a_i^*(s) = \arg\max_{a_i} \overline{A_i^{\theta^*}}(s, a_i), \;\; ii) \; \overline{A_i^{\theta^*}}(s, a_i^*(s)) = 0; \quad iii) \; \overline{A_i^{\theta^*}}(s, a_i) < 0, \; \forall \, a_i \neq a_i^*(s) \quad (9)$$

Based on this lemma, we define the following for studying the local convergence of a strict NE $\theta^*$:

$$\Delta_i^{\theta^*}(s) := \min_{a_i \neq a_i^*(s)} \left| \overline{A_i^{\theta^*}}(s, a_i) \right|, \quad \Delta^{\theta^*} := \min_i \min_s \frac{1}{1-\gamma} d_{\theta^*}(s) \Delta_i^{\theta^*}(s) > 0. \qquad (10)$$

**Theorem 2.** *(Local finite time convergence around strict NE) Define the metric of policy parameters as: $D(\theta||\theta') := \max_{1 \leq i \leq n} \max_{s \in \mathcal{S}} \|\theta_{i,s} - \theta_{i,s}'\|_1$, where $\|\cdot\|_1$ denote the $\ell_1$- norm. Suppose $\theta^*$ is a strict Nash equilibrium, then for any $\theta^{(0)}$ such that $D(\theta^{(0)}||\theta^*) \leq \frac{\Delta^{\theta^*}(1-\gamma)^3}{8n|\mathcal{S}|\left(\sum_{i=1}^n |\mathcal{A}_i|\right)}$, running gradient play (Equation (5)) will guarantee $D(\theta^{(t+1)}||\theta^*) \leq \max\left\{ D(\theta^{(t)}||\theta^*) - \frac{\eta \Delta^{\theta^*}}{2}, 0 \right\}$, which means that gradient play is going to converge within $\lceil \frac{2D(\theta^{(0)}||\theta^*)}{\eta \Delta^{\theta^*}} \rceil$ steps.*

Proofs of Lemma 4 and Theorem 2 are provided in Appendix F. The convergence only requires a finite number of steps and the stepsize $\eta$ can be chosen arbitrarily large so that exact convergence can happen in even just one step. However, the caveat is that we need to assume that the initial policy is sufficiently close to $\theta^*$. For numerical stability considerations, one should pick reasonable stepsizes to run the algorithm to accommodate random initializations. Theorem 2 also shows that the radius of region of attraction for strict NEs is at least $\frac{\Delta^{\theta^*}(1-\gamma)^3}{8n|\mathcal{S}|\left(\sum_{i=1}^n |\mathcal{A}_i|\right)}$, and thus $\theta^*$ with a larger $\Delta^{\theta^*}$, i.e., a larger value gap between the optimal action and other actions, will have a larger region of attraction. We would like to further remark that Theorem 2 only focuses on the local convergence property, the way to interpret the theorem is that, if there exists a strict NE, then it is locally asymptotic stable under gradient play. However, it does not claim to solve the global existence or convergence of the strict NEs.

## 4 GRADIENT PLAY FOR MARKOV POTENTIAL GAMES

We have discussed that the main problem for the global convergence of gradient play for general SGs is that the vector field $\{\nabla_{\theta_i} J_i(\theta)\}_{i=1}^n$ is not conservative. Thus, in this section, we restrict our analysis to a special subclass where the vector field is conservative, which in turn enjoys global convergence. This subclass is generally referred to as a Markov potential game (MPG) in the literature.

**Definition 3.** *(Markov potential game (Macua et al., 2018)) A stochastic game $\mathcal{M}$ is called a Markov potential game if there exists a potential function $\phi : \mathcal{S} \times \mathcal{A}_1 \times \cdots \times \mathcal{A}_n \to \mathbb{R}$ such that for any agent $i$ and any pair of policy parameters $(\theta_i', \theta_{-i}), (\theta_i, \theta_{-i})$ :*

$$\mathbb{E}\left[\sum_{t=0}^{\infty} \gamma^t r_i(s_t, a_t) \big| \pi = (\theta_i', \theta_{-i}), s_0 = s\right] - \mathbb{E}\left[\sum_{t=0}^{\infty} \gamma^t r_i(s_t, a_t) \big| \pi = (\theta_i, \theta_{-i}), s_0 = s\right]$$
$$=\mathbb{E}\left[\sum_{t=0}^{\infty} \gamma^t \phi(s_t, a_t) \big| \pi = (\theta_i', \theta_{-i}), s_0 = s\right] - \mathbb{E}\left[\sum_{t=0}^{\infty} \gamma^t \phi(s_t, a_t) \big| \pi = (\theta_i, \theta_{-i}), s_0 = s\right], \ \forall s.$$

As shown in the definition, the condition of a MPG is admittedly rather strong and difficult to verify for general SGs. Macua et al. (2018); González-Sánchez & Hernández-Lerma (2013) found that continuous MPGs can model applications such as the great fish war (Levhari & Mirman, 1980), the stochastic lake game (Dechert & O'Donnell, 2006), medium access control (Macua et al., 2018) etc. There are also efforts attempting to identify conditions such that a SG is a MPG, e.g., Macua et al. (2018); Leonardos et al. (2021); Mguni (2020). In Appendix B, we provide a more detailed discussion on MPGs, including a necessary condition (Lemma 5) of MPG, counterexamples of stage-wise potential games that are not MPG, sufficient conditions for a SG to be a MPG, and application examples of MPG. Nevertheless, identifying sufficient and necessary conditions and broadening the applications of MPG are important furture directions.

Given a policy $\theta$, we define the *'total potential function'* $\Phi(\theta) := \mathbb{E}_{s_0 \sim \rho(\cdot)}\left[\sum_{t=0}^{\infty} \gamma^t \phi(s_t, a_t) \big| \pi_\theta\right]$ for a MPG. The following proposition guarantees a MPG has at least one NE and it is a pure NE.

**Proposition 1.** *(Proof in Appendix G) For a Markov potential game, there is at least one global maximum $\theta^*$ of the total potential function $\Phi$, i.e.: $\theta^* \in \arg\max_{\theta \in \mathcal{X}} \Phi(\theta)$ that is a pure NE.*

From the definition of the total potential function we obtain the following relationship

$$J_i(\theta_i', \theta_{-i}) - J_i(\theta_i, \theta_{-i}) = \Phi(\theta_i', \theta_{-i}) - \Phi(\theta_i, \theta_{-i}). \tag{11}$$

Thus,
$$\nabla_{\theta_i} J_i(\theta) = \nabla_{\theta_i} \Phi(\theta),$$

which means that gradient play (Equation (5)) is equivalent to running projected gradient ascent with respect to the total potential function $\Phi$, i.e.: $\theta^{(t+1)} = Proj_{\mathcal{X}}(\theta^{(t)} + \eta \nabla_\theta \Phi(\theta_i^{(t)}))$, $\eta > 0$.

To measure the convergence to a NE, we define an $\epsilon$-Nash equilibrium as follows:

**Definition 4.** *($\epsilon$-Nash equilibrium) Define the 'NE-gap' of a policy $\theta$ as:*

$$\texttt{NE-gap}_i(\theta) := \max_{\theta_i' \in \mathcal{X}_i} J_i(\theta_i', \theta_{-i}) - J_i(\theta_i, \theta_{-i}); \quad \texttt{NE-gap}(\theta) := \max_i \texttt{NE-gap}_i(\theta).$$

*A policy $\theta$ is an $\epsilon$-Nash equilibrium if:* $\texttt{NE-gap}(\theta) \leq \epsilon$.

We further assume that the MPG satisfies the following assumption.

**Assumption 2.** *For $\theta \in \mathcal{X}$, the total potential function $\Phi(\theta)$ is bounded by:* $\Phi_{\min} \leq \Phi(\theta) \leq \Phi_{\max}$.

### 4.1 EXACT GRADIENT PLAY - GLOBAL CONVERGENCE AND LOCAL GEOMETRY

In this section, we first focus on the global convergence for exact gradient play (5), where the gradient $\nabla_{\theta_i} J_i$ can be calculated exactly by agent $i$. The convergence result is given as follows:

**Theorem 3.** *(Global convergence to Nash equilibria, proof in Appendix H.) Given a MPG with potential function $\phi(s, a)$, suppose the total potential function $\Phi$ satisfies Assumption 2. Then with*

*stepsize* $\eta = \frac{(1-\gamma)^3}{2 \sum_{i=1}^n |\mathcal{A}_i|}$, $\theta^{(t)}$ *asymptotically converge to a NE under gradient play (Equation (5)), i.e.,* $\lim_{t \to \infty} \text{NE-gap}(\theta^{(t)}) = 0$. *Further, we have:*

$$\frac{1}{T} \sum_{1 \leq t \leq T} \text{NE-gap}(\theta^{(t)})^2 \leq \epsilon^2, \quad \text{whenever} \quad T \geq \frac{64 M^2 (\Phi_{\max} - \Phi_{\min}) |\mathcal{S}| \sum_{i=1}^n |\mathcal{A}_i|}{(1-\gamma)^3 \epsilon^2}, \quad (12)$$

*where* $M := \max_{\theta, \theta' \in \mathcal{X}} \left\| \frac{d_\theta}{d_{\theta'}} \right\|_\infty$ *(by Assumption 1, we know that this quantity is well-defined).*

The factor $M$ is also known as the distribution mismatch coefficient that characterizes how the state visitation varies with the policies. Given an initial state distribution $\rho$ that has positive measure on every state, $M$ can be at least bounded by $M \leq \frac{1}{1-\gamma} \max_\theta \left\| \frac{d_\theta}{\rho} \right\|_\infty \leq \frac{1}{1-\gamma} \frac{1}{\min_s \rho(s)}$. Also note that Inequality (12) on the average term $\frac{1}{T} \sum_{1 \leq t \leq T} \text{NE-gap}(\theta^{(t)})^2$ could be translated to a constant probability guarantee on single $\text{NE-gap}(\theta^{(t)})$. For instance, if we randomly pick one $\theta(t)$ from $1 \leq t \leq T$, then it guarantees that $\text{NE-gap}(\theta^{(t)})^2 \leq 3 \cdot \epsilon^2$ with probability at least $\frac{2}{3}$.[2] As a comparison with centralized learning, if we parameterize the policy in a centralized way, the size of the action space will be $|\mathcal{A}| = \prod_{i=1}^n |\mathcal{A}_i|$ and the projected gradient ascent would need $O\left( \frac{|\mathcal{S}| \prod_{i=1}^n |\mathcal{A}_i|}{\epsilon^2} \right)$ steps to find an $\epsilon$-optimal policy (Agarwal et al., 2020); whereas we only need $O\left( \frac{|\mathcal{S}| \sum_{i=1}^n |\mathcal{A}_i|}{\epsilon^2} \right)$ steps to find an $\epsilon$-NE, which scales linearly with $n$. However, centralized parameterization can provably find a global optimum, while distributed parameterization can only find a NE.

Though gradient play is guaranteed to converge to a NE, the exact NE which it converges to is uncertain, and depends on the initial point as well as the local geometry around the various NEs. As a preliminary study, we have the following characterization for two special types of NEs. More future investigation is needed for general NEs.

**Theorem 4.** *For Markov potential game with* $\Phi_{\min} < \Phi_{\max}$ *(i.e.,* $\Phi$ *is not a constant function):*

- *A strict NE* $\theta^*$ *is equivalent to a strict local maximum of the total potential function* $\Phi$*, i.e.:* $\exists \delta$*, s.t.* $\forall \theta \neq \theta^*$ *that satisfies* $\|\theta - \theta^*\| \leq \delta$*,* $\theta \in \mathcal{X}$*, we have* $\Phi(\theta) < \Phi(\theta^*)$*.*

- *Any fully mixed NE* $\theta^*$ *is a saddle point with regard to the total potential function* $\Phi$*, i.e.:* $\forall \delta > 0$*,* $\exists \theta$*, s.t.* $\|\theta - \theta^*\| \leq \delta$ *and* $\Phi(\theta) > \Phi(\theta^*)$*.*

Theorem 4 implies that strict NEs are asymptotically stable under first-order methods such as gradient play; while the fully mixed NEs are not stable under gradient play. We remark that the conclusion about strict NE in Theorem 4 does not hold for settings other than tabular MPG; for instance, for continuous games, one can use quadratic functions to construct simple counterexamples (Mazumdar et al., 2020). Also, similar to the remark after Theorem 2, this theorem focuses on the local geometry of the NEs but does not claim the global existence or convergence of either strict NEs or fully mixed NEs.

## 4.2 SAMPLE-BASED LEARNING: ALGORITHM DESIGN AND SAMPLE COMPLEXITY

In this section, we no longer assume access to the exact gradient, but instead estimate it via samples. Throughout the section, we make the following additional assumption on MPGs:

**Assumption 3.** *(($\tau, \sigma_S$)-Sufficient exploration on states) There exist a positive integer* $\tau$ *and a* $\sigma_S \in (0, 1)$ *such that for any policy* $\theta$ *and any initial state-action pair* $(s, a_i)$*,* $\forall i$*, we have*

$$\Pr(s_\tau | s_0 = s, a_0 = a) \geq \sigma_S, \quad \forall s_\tau. \quad (13)$$

It says that every state has a positive probability of being visited after some time. This assumption is common in proving finite time convergence (e.g. (Qu et al., 2019; Srikant & Ying, 2019)).

We further introduction the *state transition probability under* $\theta$ $\overline{P_\mathcal{S}^\theta} : \mathcal{S} \times \mathcal{S} \to \mathbb{R}$ as:

$$\overline{P_\mathcal{S}^\theta}(s'|s) := \sum_a \pi_\theta(a|s) P(s'|s, a).$$

We consider fully decentralized learning, where agent $i$'s observation only includes state $s_t$, its own action $a_{i,t}$, and its own reward $r_{i,t} := r_i(s_t, a_t)$ at time $t$. Such fully decentralized learning is

---

[2]Here $3, \frac{2}{3}$ could be replaced by $\frac{1}{1-p}, p$ where $p \in (0, 1)$ is a probability.

plausible due to the fact that when $\theta_{-i}$ is *fixed*, agent $i$ can be treated as an independent learner with the underlying MDP being the 'averaged' MDP described in Section 2. With this key observation, we design a two-timescale 'model-based' on-policy learning algorithm, where agents perform policy evaluation in the inner loop and gradient ascent at the outer loop. The algorithm is provided in Algorithm 1. Roughly, it consists of three main steps: 1) (Inner loop) Estimate the averaged transition probability and reward using on-policy samples $\overline{P_i^\theta}, \overline{r_i^\theta}, \overline{P_S^\theta}$. 2) (Inner loop) Calculate averaged Q-function $\overline{Q_i^\theta}$ and discounted state visitation distribution $d_\theta$, and compute the estimated gradient accordingly, 3) (Outer loop) Running projected gradient ascent with estimated gradients. Before discussing our algorithm in more detail, we highlight that the idea of using the "averaged" MDP can be used to design other learning methods including model-free methods, e.g., using the temporal difference methods to perform policy evaluation. One caveat is that the "averaged" MDP is only well-defined when all the other agents use fixed policies. This makes it difficult to extend the two-timescale framework to single-timescale settings, which is an interesting future direction.

---

**Algorithm 1** Sample-based learning

---

**Require:** learning rate $\eta$, greedy parameter $\alpha$, sample trajectory length $T_J$, total iteration steps $T_G$
    For each agent $i$
    **for** $k = 0, 1 \ldots, T_G - 1$ **do**
        **for** $t = 0, 1, \ldots, T_J$ **do**
        Implement policy $\theta^{(k)}$ and collect trajectory $\mathcal{D}_i^{(k)}$: $\mathcal{D}_i^{(k)} \leftarrow \mathcal{D}_i^{(k)} \cup \{s_t, a_{i,t}, r_{i,t}\}$, $a_{i,t} \sim \pi_{\theta_i^{(k)}}(\cdot|s)$
        **end for**
        Estimate $\widehat{P_i^\theta}, \widehat{r_i^\theta}, \widehat{P_S^\theta}, \widehat{M_i^\theta}$ by Equation (14), Equation (15), Equation (16) respectively.
        Calculate $\widehat{Q_i^\theta}, \widehat{d_\theta}$ by Equation (17), Equation (18) respectively.
        Estimate the gradient by Equation (19):
        Run projected gradient ascent as in Equation (20)
    **end for**

---

**Step 1: empirical estimation of $\overline{P_i^\theta}, \overline{r_i^\theta}, \overline{P_S^\theta}$:** Given a sequence $\{s_t, a_{i,t}, r_{i,t}\}_{t=0}^{T_J}$ generated by a policy $\theta := (\theta_i, \theta_{-i})$, the empirical estimation $\widehat{P_i^\theta}$ of $\overline{P_i^\theta}$ is given by:

$$\widehat{P_i^\theta}(s'|s, a_i) := \begin{cases} \frac{\sum_{t=0}^{T_J-1} \mathbf{1}\{s_{t+1}=s', s_t=s, a_{i,t}=a_i\}}{\sum_{t=1}^{T_J-1} \mathbf{1}\{s_t=s, a_{i,t}=a_i\}}, & \sum_{t=1}^{T_J-1} \mathbf{1}\{s_t = s, a_{i,t} = a_i\} \geq 1 \\ \mathbf{1}\{s' = s\}, & \sum_{t=1}^{T_J-1} \mathbf{1}\{s_t = s, a_{i,t} = a_i\} = 0 \end{cases} \quad (14)$$

Here we separately treat the special case where the state and action pair is not visited through the whole trajectory, i.e., $\sum_{t=1}^{T_J-1} \mathbf{1}\{s_t = s, a_{i,t} = a_i\} = 0$ to make $\widehat{P_i^\theta}$ well-defined.

Similarly, the estimates $\widehat{r_i^\theta}, \widehat{P_S^\theta}$ of $\overline{r_i^\theta}, \overline{P_S^\theta}$ are given by:

$$\widehat{r_i^\theta}(s, a_i) := \begin{cases} \frac{\sum_{t=0}^{T_J} \mathbf{1}\{s_t=s, a_{i,t}=a_i\} r_{i,t}}{\sum_{t=0}^{T_J} \mathbf{1}\{s_t=s, a_{i,t}=a_i\}}, & \sum_{t=1}^{T_J-1} \mathbf{1}\{s_t = s, a_{i,t} = a_i\} \geq 1 \\ 0, & \sum_{t=1}^{T_J-1} \mathbf{1}\{s_t = s, a_{i,t} = a_i\} = 0 \end{cases} \quad (15)$$

$$\widehat{P_S^\theta}(s'|s) := \begin{cases} \frac{\sum_{t=0}^{T_J-1} \mathbf{1}\{s_{t+1}=s', s_t=s\}}{\sum_{t=1}^{T_J-1} \mathbf{1}\{s_t=s\}}, & \sum_{t=1}^{T_J-1} \mathbf{1}\{s_t = s\} \geq 1 \\ \mathbf{1}\{s' = s\}, & \sum_{t=1}^{T_J-1} \mathbf{1}\{s_t = s\} = 0 \end{cases} \quad (16)$$

**Step 2: estimation of $\overline{Q_i^\theta}, d_\theta$:** We slightly abuse notation and use $\overline{Q_i^\theta}, \overline{r_i^\theta} \in \mathbb{R}^{|\mathcal{S}||\mathcal{A}_i|}$ to also denote the vectors corresponding to the averaged Q-function and reward function of agent $i$. Similarly, $\rho, d_\theta \in \mathbb{R}^{|\mathcal{S}|}$ are used to denote the vectors for $\rho(s)$ and $d_\theta(s)$. Define $M_i^\theta \in \mathbb{R}^{|\mathcal{S}||\mathcal{A}_i| \times |\mathcal{S}||\mathcal{A}_i|}$:

$$\overline{M_i^\theta}_{(s,a_i) \to (s', a_i')} := \pi_{\theta_i}(a_i'|s') \overline{P_i^\theta}(s'|s, a_i).$$

Then from Lemma 1, $\overline{Q_i^\theta}$ is given by: $(I - \gamma\overline{M_i^\theta})\overline{Q_i^\theta} = \overline{r_i^\theta} \implies \overline{Q_i^\theta} = (I - \gamma\overline{M_i^\theta})^{-1}\overline{r_i^\theta}$.

The estimated averaged Q-function $\widehat{Q_i^\theta}$ is given by:[3]

$$\widehat{Q_i^\theta} = (I - \gamma\widehat{M_i^\theta})^{-1}\widehat{r_i^\theta}, \text{ where } \widehat{M_i^\theta}_{(s,a_i)\to(s',a_i')} := \pi_{\theta_i}(a_i'|s')\widehat{P_i^\theta}(s'|s,a_i) \tag{17}$$

Similarly, from Equation (2), we have that $d_\theta$ and $\widehat{d_\theta}$ are given by (derivation see Appendix C):

$$d_\theta = (1-\gamma)\left(I - \gamma\overline{P_\mathcal{S}^\theta}^\top\right)^{-1}\rho, \qquad \widehat{d_\theta} := (1-\gamma)\left(I - \gamma\widehat{P_\mathcal{S}^\theta}^\top\right)^{-1}\rho. \tag{18}$$

Then accordingly, the estimated gradient is computed as:

$$\widehat{\partial}_{\theta_{s,a_i}} J_i(\theta^{(k)}) = \frac{1}{1-\gamma}\widehat{d_\theta}(s)\widehat{Q_i^\theta}(s,a_i). \tag{19}$$

**Step 3: Projected gradient ascent onto the set of $\alpha$-greedy policies:** Let $U_n = [\frac{1}{n}, \ldots, \frac{1}{n}] \in \Delta(n)$ denote the $n$ dimensional uniform distribution. Define $\Delta^\alpha(n) := \{\theta|\ \exists\theta'\in\Delta(n), s.t.\ \theta = (1-\alpha)\theta'+\alpha U_n\}$. We use $\mathcal{X}_i^\alpha := \Delta^\alpha(|\mathcal{A}_i|)^{|\mathcal{S}|}$, $\mathcal{X}^\alpha := \mathcal{X}_1^\alpha \times \mathcal{X}_2^\alpha \times \cdots \times \mathcal{X}_n^\alpha$ to denote the set of the $\alpha$-greedy policies for $\theta_i$ and $\theta$ respectively. Every step after doing gradient ascent, the parameter $\theta$ will further be projected onto $\mathcal{X}^\alpha$, i.e.:

$$\theta_i^{(k+1)} = Proj_{\mathcal{X}_i^\alpha}(\theta_i^{(k)} + \eta\widehat{\nabla}_{\theta_i} J_i(\theta^{(k)})) \tag{20}$$

The reason of projecting onto $\mathcal{X}^\alpha$ instead of $\mathcal{X}$ is to make sure that every action has positive possibility of being selected in order to get a relatively accurate estimation of averaged $Q$-function. Intuitively, a larger $\alpha$ introduces a larger additional error in the NE-gap; however, a smaller $\alpha$ requires more samples to estimate the gradient. Thus the choice of $\alpha$ is the tradeoff between the two effects.

**Theorem 5.** *(Sample complexity) Assume that the MPG satisfies Assumption 3. Let $M :=$* $\max_{\theta,\theta'\in\mathcal{X}}\left\|\frac{d_\theta}{d_{\theta'}}\right\|_\infty$. *In Algorithm 1, for $\eta \le \frac{(1-\gamma)^3}{4\sum_i|\mathcal{A}_i|}$, $\alpha = \frac{(1-\gamma)\epsilon}{6M}$, and*

$$T_J \ge \frac{206976\tau nM^4|\mathcal{S}|^3\max_i|\mathcal{A}_i|^3}{(1-\gamma)^8\epsilon^4\sigma_\mathcal{S}^2}\log\left(\frac{16\tau T_G|\mathcal{S}|^2\sum_i|\mathcal{A}_i|}{\delta}\right)+\tau, \quad T_G \ge \frac{648M^2(\Phi_{\max}-\Phi_{\min})|\mathcal{S}|}{\eta\epsilon^2}$$

*with probability at least $1 - \delta$, we have that:* $\frac{1}{T_G}\sum_{k=1}^{T_G} \mathrm{NE\text{-}gap}(\theta^{(k)})^2 \le \epsilon^2$

*That is, the algorithm can find an $\epsilon$-NE with probability at least $1 - \delta$ with*

$$T_J T_G \sim \tilde{O}\left(\frac{n}{\epsilon^6}\mathrm{poly}\left(\frac{1}{1-\gamma}, |\mathcal{S}|, \max_i|\mathcal{A}_i|\right)\right) \tag{21}$$

*samples, where $\tilde{O}$ hides log factors.*

**Comparison with centralized learning:** The best known sample complexity bound for single-agent/centralized MDP is $\tilde{O}\left(\frac{|\mathcal{S}||\mathcal{A}|}{(1-\gamma)^3\epsilon^2}\right)$ (Sidford et al., 2018). Compared with Equation (21), the centralized bound scales better with respect to $\epsilon, |\mathcal{S}|, |\mathcal{A}_i|, \frac{1}{1-\gamma}$. However, as argued in the previous subsection, the total action space $|\mathcal{A}| = \prod_{i=1}^n |\mathcal{A}_i|$ in the centralized bound scales exponentially with the number of agent $n$, while our complexity bound only scales linearly. Here, we briefly state the fundamental difficulties of learning in the SG setting compared with centralized learning, which also explains why our bound scales worse with respect to the factors $\epsilon, |\mathcal{S}|, |\mathcal{A}_i|, \frac{1}{1-\gamma}$. 1) Firstly, the optimization landscape in the SG setting is more complicated. For centralized learning, the gradient domination property is stronger and accelerated gradient methods (e.g. via natural policy gradient or entropy regularization) can speed up the convergence of exact gradient from $O(\frac{1}{\epsilon^2})$ to $O(\frac{1}{\epsilon})$ (Agarwal et al., 2020), or even $O(\log(\frac{1}{\epsilon}))$ (Mei et al., 2020). In contrast, for multi-agent settings, due to the more complicated optimization landscape, these methods can no longer improve the dependency on $\epsilon$, which thus makes the outer loop complexity $T_G$ larger. 2) Secondly, the behavior of other agents makes the environment non-stationary, i.e., the averaged Q-function $\overline{Q_i^\theta}$ as well as the averaged transition probability distribution $\overline{P_i^\theta}$ depends on the policy of other agent $\theta_{-i}$. Thus, unlike centralized learning, where the state transition probability matrix can be estimated in an off-policy or

---

[3]From the Perron-Frobenius theorem, we know that the absolute values of the eigenvalues of $\widehat{M_i^\theta}$ are upper bounded by 1, which guarantees that the matrix $I - \gamma\widehat{M_i^\theta}$ is invertible.

even offline manner, i.e. using data samples from different policies, $\overline{P_i^\theta}$ can only be estimated in a online manner, using samples generated by exactly the same policy $\theta$, which increases the inner loop complexity $T_J$. 3) Thirdly, the complicated interactions amongst agents necessitate more care during the learning process. Algorithms designed for centralized learning that achieve near-optimal sample complexity are generally Q-learning type algorithms. However, in SGs, it can be shown that having each agent maximize its own averaged Q-function may actually lead to non-convergent behavior. Thus, we need to consider algorithms that update in a less aggressive manner, e.g. soft Q-learning, or policy gradient (which is considered in this paper).

**Numerical Results.** Due to the space limit, numerical performance for algorithms studied in this paper are deferred to Appendix A.

## 5   CONCLUSION AND DISCUSSION

This paper studies the optimization landscape and convergence of gradient play for SGs. For general SGs, we establish some local convergence results; for MPGs, we establish the global convergence and design a sample-based method. There are many future directions. Firstly, the assumption of MPGs is relatively strong compared with the notion of potential games in the one-shot setup, which might restrict its application to broader settings. More effort would be needed to come up with other special types of SGs which facilitate learning. It would also be meaningful to combine the insight from learning in stochastic games to investigate real-life applications, such as dynamic congestion and routing. Secondly, the sample-based learning algorithm proposed in this paper is only one choice. Other methods, such as actor-critic, natural policy gradient, Gauss-Newton methods, could also be considered, which might improve the sample complexity. We note in passing that it is an interesting and important question to characterize the fundamental complexity of MARL. Thirdly, this paper only considers direct policy parameterization. To broaden the application of MARL and to strengthen the results, other types of parameterization such as softmax parameterization and general nonlinear parameterization should be investigated.

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

## A NUMERICAL EXAMPLES

|  | $a_2 = 1$ | $a_2 = 2$ |
|---|---|---|
| $a_1 = 1$ | (-1,-1) | (-3,0) |
| $a_1 = 2$ | (0,-3) | (-2,-2) |

Table 1: Reward table for Game 1

|  | $s_2 = 1$ | $s_2 = 2$ |
|---|---|---|
| $s_1 = 1$ | 2 | 0 |
| $s_1 = 2$ | 0 | 1 |

Table 2: Reward table for Game 2

**Game 1: multi-stage prisoner's dilemma**   The first example — multi-stage prisoner's dilemma model(Arslan & Yüksel, 2016) — studies exact gradient play for general SG settings. It is a 2-agent MDP, with $\mathcal{S} = \mathcal{A}_1 = \mathcal{A}_2 = \{1, 2\}$. Assume that the reward for each agent $r_i(s, a_1, a_2)$, $i \in \{1, 2\}$ is independent of state $s$ and is given by Table 1. The state transition probability is determined by agents' previous actions:

$$P(s_{t+1} = 1|(a_{1,t}, a_{2,t}) = (1, 1)) = 1 - \epsilon, \quad P(s_{t+1} = 1|(a_{1,t}, a_{2,t}) \neq (1, 1)) = \epsilon$$

Here action $a_i = 1$ means that agent $i$ choose to *cooperate* and $a_i = 2$ means *betray*. The state $s$ serves as a noisy indicator, with accuracy $1 - \epsilon$, of whether both agents cooperated ($s_t = 1$) or not ($s_t = 2$) in the previous stage $t - 1$.

The single-stage game corresponds to the famous *Prisoner's Dilemma*, and it is well-known that there is a unique NE $(a_1, a_2) = (2, 2)$, where both agent decide to betray. The dilemma arises from the fact that there exists a joint non-NE strategy $(1, 1)$ such that both players obtain a higher reward than what they get under the NE. However, in the multi-stage case, the introduction of an additional state $s$ allows agents to make decisions based on whether they have cooperated before. Intuitively, given that both agents cooperated at the previous stage, it is more beneficial to keep this cooperation rather than destroy it by betraying. It turns out that cooperation can be achieved in this manner given that the two agents are patient (i.e., $\gamma$ is close to 1) and that the indicator $s$ is accurate enough (i.e. $\epsilon$ is close to 0). Apart from the fully betray strategy, where both agents will betray regardless of $s$,

there is another strict NE $\theta^*$ that is $\theta^*_{s=1,a_i=1} = 1$, $\theta^*_{s=2,a_i=1} = 0$, where agents will cooperate given that they have cooperated in previous stage, and betray otherwise.

We simulate gradient play for this model and mainly focus on the convergence to the cooperative equilibrium $\theta^*$. The initial policy is set as: $\theta^{(0)}_{s=1,a_i=1} = 1 - 0.4\delta_i$, $\theta^{(0)}_{s=2,a_i=1} = 0$, where the $\delta_i$'s are uniformly sampled from $[0, 1]$. The initialization implies that at the beginning, both agents are willing to cooperate to some extent given that they cooperated at the previous stage. Figure 1 shows a trial converging to the NE starting from a randomly initialized policy. The size of the region of attraction for $\theta^*$ can be reflected by the ratio of convergence ($\frac{\#\text{Trials that converge to } \theta^*}{\#\text{Total number of trials}}$) for multiple trials with different initial points. An empirical estimate of the volume of the region is the convergence ratio times the volume of the uniform sampling area; the larger the ratio, the larger the region of attraction. Table 3 demonstrates the change of the ratio with regard to different values of indicator error $\epsilon$. Intuitively speaking, the more accurately the state $s$ represents the cooperation situation of the agents, the less incentive agents will have for betraying when observing $s = 1$, that is, the larger $\Delta^{\theta^*}(s=1)$ will become, and thus the larger the convergence ratio will be. This intuition matches the simulation result as well as the theoretical guarantees on the local convergence around a strict NE in Theorem 2.

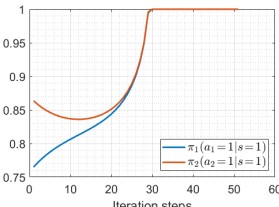

Figure 1: (Game 1:) Convergence to the cooperative NE

| $\epsilon$ | $\Delta^{\theta^*}(s=1)$ | ratio (mean $\pm$ std)% |
|---|---|---|
| 0.1 | 433.3 | (47.8$\pm$ 5.1)% |
| 0.05 | 979.3 | (66.3$\pm$ 4.3)% |
| 0.01 | 2498.6 | (77.4$\pm$ 2.8)% |

Table 3: (Game 1:) Relationship of convergence ratio and $\epsilon$. Here we fix $\gamma = 0.95$. Convergence ratio is calculated by $\frac{\#\text{Trials that converge to } \theta^*}{\#\text{Total number of trials}}$. Here we calculate one ratio using 100 trials and the mean and standard deviation (std) are calculated by computing the ratio 10 times using different trials. $\Delta^{\theta^*}(s=1)$ is calculated using Equation (10)

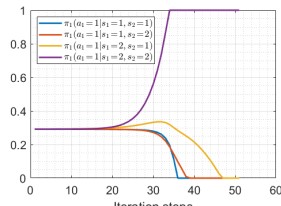

Figure 2: (Game 2:) Starting from a close neighborhood of a fully mixed NE

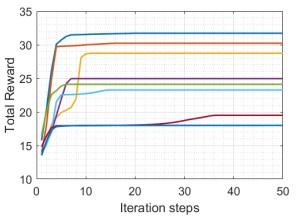

Figure 3: (Game 2:) Total reward for multiple runs

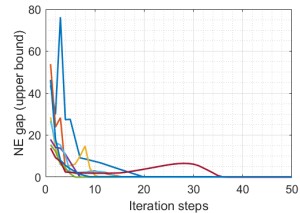

Figure 4: (Game 2:) NE-gap for multiple runs

**Game 2: coordination game** Our second numerical example studies the empirical performance of exact gradient play for an identical-reward game which is a special class of Markov potential game. Consider a 2-agent identical reward coordination game problem with state space $\mathcal{S} = \mathcal{S}_1 \times \mathcal{S}_2$ and action space $\mathcal{A} = \mathcal{A}_1 \times \mathcal{A}_2$, where $\mathcal{S}_1 = \mathcal{S}_2 = \mathcal{A}_1 = \mathcal{A}_2 = \{1, 2\}$. The state transition probability is given by:

$$P(s_{i,t+1} = 1 | a_{i,t} = 1) = 1 - \epsilon, \ P(s_{i,t+1} = 1 | a_{i,t} = 2) = \epsilon, \quad i = 1, 2.$$

The reward table is given by Table 2, where agents will only be rewarded if they are in the same state, and state 1 has a higher reward than state 2. Coordination games can be used to model the network effect in economics, where an agent reaps benefits from being in the same network as other agents. For this specific example, there are two networks with different utilities. Agents can observe the occupancy of each network, and take actions to join one of the networks based on the observation.

There is at least one fully-mixed NE where both agents join network 1 with probability $\frac{1-3\epsilon}{3(1-2\epsilon)}$ regardless of the current occupancy of networks, and there are 13 different strict NEs that can be verified numerically (computation of the NEs as well as detailed settings can be found in Appendix N). Figure 2 shows a gradient play trajectory whose initial point lies in a close neighborhood of the

mixed NE. As the algorithm progresses, we see that the trajectory in Figure 2 diverges from the mixed-NE, indicating that the fully-mixed NE is indeed a saddle point. This corroborates our finding in Theorem 4. Figure 3 shows the evolution of total reward $J(\theta^{(t)})$ for gradient play for different random initial points $\theta^{(0)}$. Different initial points converge to one of 13 different strict NEs each with a different total reward (some strict NEs with relatively small region of attraction are omitted in the figure). We see the total reward is monotonically increasing for each initial point, which makes sense since gradient play runs projected gradient ascent on the total reward function $J$. While the total rewards are different, as shown in Figure 4, we see that the NE-gap of each trajectory (corresponding to same initial points in Figure 3) converges to 0. This suggests that the algorithm is indeed able to converge to a NE. Notice that the NE-gaps do not decrease monotonically.

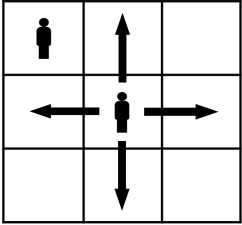 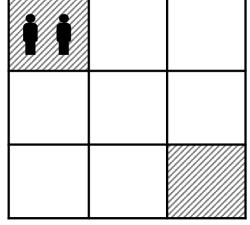

Figure 5: (Game 3:) State-based coordination game, rewards are nonzero if both players locate at the same shaded grids

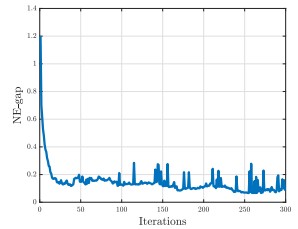

Figure 6: Total reward $J(\theta^{(t)})$ keeps increasing

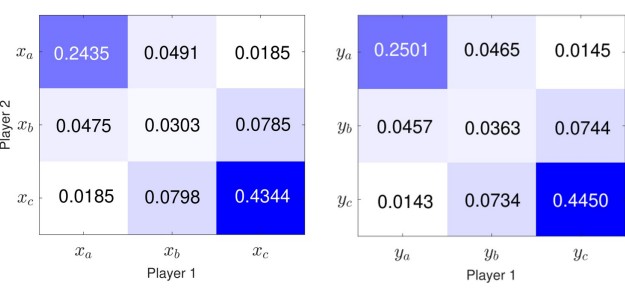

Figure 7: Marginal distribution $d_\theta^x(s_{1,x}, s_{2,x})$ and $d_\theta^y(s_{1,y}, s_{2,y})$

Figure 8: NE-gap converges to a value close to zero. Here the NE-gap is measured by $\max_i \max_{(s,a_i)\overline{A_i^\theta}(s,a_i)}$

**Game 3: state-based coordination game**  Our third numerical example studies the empirical performance of the sample-based learning algorithm, Algorithm 1. Here we consider a generalization of coordination game (Game 2) where the two players now try to coordinate on a 2D grid. The two-player state-based coordination game on a $3 \times 3$ grid is defined as follows: the state space is given by $\mathcal{S} = \mathcal{S}_1 \times \mathcal{S}_2$, $\mathcal{S}_1 = \mathcal{S}_2 = \mathcal{S}_x \times \mathcal{S}_y = \{x_a, x_b, x_c\} \times \{y_a, y_b, y_c\}$, action space is given by $\mathcal{A} = \mathcal{A}_1 \times \mathcal{A}_2$, $\mathcal{A}_1 = \mathcal{A}_2 = \{\text{Stay}, \text{Left}, \text{Right}, \text{Up}, \text{Down}\}$, i.e., agent can choose to stay at current grid or move left/right/up/down to its neighboring grids. We assume that there is random noise during the transition; for example, if agent 1 is staying at the middle grid $(x_b, y_b)$, then:

$$P(s'_{1,x} = x_a|s_{1,x} = x_b, a_1 = \text{Stay}) = P(s'_{1,x} = x_c|s_{1,x} = x_b, a_1 = \text{Stay}) = \epsilon,$$
$$P(s'_{1,x} = x_b|s_{1,x} = x_b, a_1 = \text{Stay}) = 1 - 2\epsilon,$$
$$P(s'_{1,y} = y_a|s_{1,y} = x_b, a_1 = \text{Stay}) = P(s'_{1,y} = y_c|s_{1,y} = x_b, a_1 = \text{Stay}) = \epsilon,$$
$$P(s'_{1,y} = y_b|s'_{1,y} = y_b, a_1 = \text{Stay}) = 1 - 2\epsilon.$$

The reward is given by:

$$r(s_1, s_2) = \mathbf{1}\{s_1 = s_2 = \{x_a, y_a\} \text{ or } \{x_c, y_c\}\},$$

i.e. the two agents are only rewarded if they stay at the upper-left or lower-right corner at the same time.

For numerical simulation, we take $T_G = 300, T_J = 10000, \alpha = 0.1, \eta = 10, \epsilon = 0.1$; the numerical results are as displayed in Figure 6 - 8. Figure 6 shows that total reward increases as the number of iterations increase, and Figure 8 shows that the NE-gap converges to a value close to zero. However, because we project the policy to the $\alpha$-greedy set $\mathcal{X}^\alpha$, the NE-gap cannot converge to exactly zero.

Figure 7 visualizes the discounted state visitation distribution. To make the visualization more intuitive, we look at the marginalized discounted state visitation distribution $d_\theta^x, d_\theta^y$ defined below:

$$d_\theta^x(s_{1,x}, s_{2,x}) = \sum_{s_{1,y}, s_{2,y}} d_\theta(s_{1,x}, s_{1,y}, s_{2,x}, s_{2,y}) = \mathbb{E}_{s_0 \sim \rho}(1-\gamma) \sum_{t=0}^{\infty} \gamma^t \Pr(s_{1,x,t} = s_{1,x}, s_{2,x,t} = s_{2,x} | s_0, \pi_\theta)$$

$$d_\theta^y(s_{1,y}, s_{2,y}) = \sum_{s_{1,x}, s_{2,x}} d_\theta(s_{1,x}, s_{1,y}, s_{2,x}, s_{2,y}) = \mathbb{E}_{s_0 \sim \rho}(1-\gamma) \sum_{t=0}^{\infty} \gamma^t \Pr(s_{1,y,t} = s_{1,y}, s_{2,y,t} = s_{2,y} | s_0, \pi_\theta)$$

From Figure 7 we can see that most of the probability measure concentrates on $\{(x_a, x_a), (x_c, x_c)\}$, $\{(y_a, y_a), (y_c, y_c)\}$, indicating that the two agents are able to coordinate most of the time.

## B  MORE ABOUT MARKOV POTENTIAL GAMES

This section is dedicated to a more thorough understanding of Markov potential games, which includes necessary or sufficient conditions for MPG and a few (counter)examples.

### B.1. A necessary condition and counterexamples

**Definition 5.** *(Monderer & Shapley (1996)) Define the path in the parameter space as $\tau = (\theta^{(0)}, \theta^{(1)}, \dots, \theta^{(N)})$, where $\theta^t, \theta^{t+1}$ differ only one component $i_t$, i.e. $\theta^{(t+1)} = (\theta_{i_t}^{(t+1)}, \theta_{-i_t}^{(t)})$. A closed path is a path such that $\theta^{(0)} = \theta^{(N)}$. Define:*

$$I(\tau) := \sum_{t=1}^{N} J_{i_t}(\theta^{(t)}) - J_{i_t}(\theta^{(t-1)})$$

The following theorem is a direct generalization of Theorem 2.8 in Monderer & Shapley (1996) to MPG setting:

**Lemma 5.** *For Markov potential games, $I(\tau) = 0$ for any finite closed path $\tau$.*

*Proof.* The proof is quite straightforward from the definition of MPG

$$\begin{aligned} I(\tau) &= \sum_{t=1}^{N} J_{i_t}(\theta^{(t)}) - J_{i_t}(\theta^{(t-1)}) \\ &= \sum_{t=1}^{N} \Phi(\theta^{(t)}) - \Phi(\theta^{(t-1)}) \\ &= \Phi(\theta^{(N)}) - \Phi(\theta^{(0)}) \\ &= 0. \end{aligned}$$

$\square$

Although the proof of Lemma 5 is straightforward, it serves as a useful tool in proving that a game is not a MPG. For example, applying the theorem we can get that the following conditions are not sufficient conditions for MPG.

**Proposition 2.** *None of the following conditions on a SG necessarily imply that it is a MPG:*

(1) *There exists $\phi(s, a)$ such that at each $s$, $r_i(s, a_i', a_{-i}) - r_i(s, a_i, a_{-i}) = \phi(s, a_i', a_{-i}) - \phi(s, a_i, a_{-i})$;*

(2) *There exists $\phi(s, a)$ such that for every $s, \hat{s}$, $r_i(s, a_i', a_{-i}) - r_i(\hat{s}, a_i, a_{-i}) = \phi(s, a_i', a_{-i}) - \phi(\hat{s}, a_i, a_{-i})$;*

(3) *Rewards $r_i$ are independent of $s$, and they have a potential function, i.e., $r_i(a_i, a_{-i}) - r_i(a_i', a_{-i}) = \phi(a_i, a_{-i}) - \phi(a_i', a_{-i})$.*

*Proof.* (of Proposition 2) A simple counterexample showing that the conditions in Equation (2) are not sufficient is the multi-stage prisoner's dilemma (Game 1) introduced in the numerical section (Appendix A). Since the reward table for multi-stage prisoner's dilemma is the same as the one-shot prisoner's dilemma (which is known to be a potential game), Game 1 satisfies condition (3) in Proposition 2, which implies condition (2), which in turn implies condition (1). In the following we are going to use Lemma 5 to show that Game 1 is not a MPG. We define the following individual policies:

$$
\begin{aligned}
\theta_i^{\text{Defect}} : \qquad & \theta_{s=1,a_i=1}^{\text{Defect}} = 0, \ \theta_{s=2,a_i=1}^{\text{Defect}} = 0 \\
\theta_i^{\text{Coop}} : \qquad & \theta_{s=1,a_i=1}^{\text{Coop}} = 1, \ \theta_{s=2,a_i=1}^{\text{Coop}} = 0 \\
\theta_i^{\text{Always\_coop}} : \qquad & \theta_{s=1,a_i=1}^{\text{Always\_coop}} = 1, \ \theta_{s=2,a_i=1}^{\text{Always\_coop}} = 1
\end{aligned}
$$

Let:

$$
\begin{aligned}
\theta^{(0)} = (\theta_1^{\text{Defect}}, \theta_2^{\text{Coop}}), \qquad\qquad & \theta^{(1)} = (\theta_1^{\text{Coop}}, \theta_2^{\text{Coop}}), \\
\theta^{(2)} = (\theta_1^{\text{Coop}}, \theta_2^{\text{Always\_coop}}), \qquad\qquad & \theta^{(3)} = (\theta_1^{\text{Defect}}, \theta_2^{\text{Always\_coop}})
\end{aligned}
$$

and define a path $\tau$ by:

$$
\tau = (\theta^{(0)}, \theta^{(1)}, \theta^{(2)}, \theta^{(3)}, \theta^{(4)}), \ \ \theta^{(4)} = \theta^{(0)}.
$$

For the sake of easy calculation, we set $\epsilon = 0$ and set initial state as $s_0 = 1$ in Game 1. In this example, it is not hard to see that $J_{i_t}(\theta^{(t)}) - J_{i_t}(\theta^{(t-1)}) > 0$ for each $t \in \{1, 2, 3, 4\}$, implying that $I(\tau) > 0$. This indicates that although Game 1 satisfies condition (3) (as well as conditions (1) and (2)), it is still not a MPG.[4] □

### B.2. A sufficient condition

Proposition 2 suggests that MPG is a quite restrictive assumption. Even if the reward table for a SG is the same as the reward table of a one-shot potential game, the SG may still not be a MPG. Nevertheless, we can show that the following condition is sufficient for a stochastic game to be a MPG:[5]

**Lemma 6.** *A stochastic game is a MPG if condition (1) in Proposition 2 is satisfied and that* $P(s'|s, a) = P(s'|s)$.

*Proof.* $P(s'|s, a) = P(s'|s)$ implies that the discounted state visitation distribution $d_\theta$ does not depend on $\theta$, and thus we denote it as $d(s)$ instead. Condition (1) implies that $\phi(s, a_i, a_{-i}) - r_i(s, a_i, a_{-i})$ only depends on $s$ and $a_{-i}$ but not $a_i$, and so we denote the difference as $\delta_i(s, a_{-i})$, i.e.,

$$
\phi(s, a_i, a_{-i}) - r_i(s, a_i, a_{-i}) = \delta_i(s, a_{-i}).
$$

The total reward of agent $i$ can be written as:

$$
\begin{aligned}
J_i(\theta) &= \sum_s d(s) \sum_a \pi_\theta(a|s) r_i(s, a) \\
&= \sum_s d(s) \sum_{a_i} \pi_{\theta_i}(a_i|s) \sum_{a_{-i}} \pi_{\theta_{-i}}(a_{-i}|s) r(s, a_i, a_{-i})
\end{aligned}
$$

Similarly, total potential function can be written as:

$$
\begin{aligned}
\Phi(\theta) &= \sum_s d(s) \sum_a \pi_\theta(a|s) \phi(s, a) \\
&= \sum_s d(s) \sum_{a_i} \pi_{\theta_i}(a_i|s) \sum_{a_{-i}} \pi_{\theta_{-i}}(a_{-i}|s) \phi(s, a_i, a_{-i})
\end{aligned}
$$

---

[4]We can use this counterexample to show that some propositions (e.g., Proposition 3) in the arXiv preprint (Mguni, 2020) would need more consideration since the multi-stage prisoner's dilemma satisfies the conditions given in (Mguni, 2020).

[5]The recent arXiv preprint (Leonardos et al., 2021) has a similar characterization.

Thus,

$$\Phi(\theta) - J_i(\theta) = \sum_s d(s) \sum_{a_i} \pi_{\theta_i}(a_i|s) \sum_{a_{-i}} \pi_{\theta_{-i}}(a_{-i}|s) \left(\phi(s, a_i, a_{-i}) - r(s, a_i, a_{-i})\right)$$

$$= \sum_s d(s) \sum_{a_i} \pi_{\theta_i}(a_i|s) \sum_{a_{-i}} \pi_{\theta_{-i}}(a_{-i}|s)\delta_i(s, a_{-i})$$

$$= \sum_s d(s) \sum_{a_{-i}} \pi_{\theta_{-i}}(a_{-i}|s)\delta_i(s, a_{-i}),$$

which does not depend on parameter $\theta_i$, i.e.,

$$\Phi(\theta_i', \theta_{-i}) - J_i(\theta_i', \theta_{-i}) = \Phi(\theta_i, \theta_{-i}) - J_i(\theta_i, \theta_{-i}), \quad \forall(\theta_i', \theta_{-i}), (\theta_i, \theta_{-i}) \in \mathcal{X},$$

which completes the proof. $\qquad\square$

### B.3. MPG with local states and an application example

From Proposition 2, we see that it is difficult for a SG to be a MPG even if the game is a potential game at each state. Lemma 6 only presents a very special case where the action does not affect the state, meaning that this MPG is merely a collection of potential games. To provide a MPG which is beyond the identical-interest case and the case in Lemma 6, inspired by the setting in Qu et al. (2019) and Macua et al. (2018), here we consider a special multi-agent setting where $\mathcal{S} = \mathcal{S}_1 \times \cdots \times \mathcal{S}_n$ and $\mathcal{S}_i$ is the local state space of agent $i$. In addition, the transition probability takes the decomposed form, $P(s'|s, a) = \prod_{i=1}^n P(s_i'|s_i, a_i)$. The rest of the SG setting is the same as the SG in Section 2. Deviating slightly from the main text, we consider the *localized policy* where each agent take actions based on its own state,

$$\pi_\theta(a_t|s_t) = \prod_{i=1}^n \pi_{\theta_i}(a_{i,t}|s_{i,t})$$

with the *localized direct parameterization*:

$$\pi_{\theta_i}(a_{i,t}|s_{i,t}) = \theta_{(s_i, a_i)}, \quad \theta_i \in \Delta(\mathcal{A}_i)^{|\mathcal{S}_i|}$$

We use $\mathcal{X}_i^{\text{local}} := \Delta(\mathcal{A}_i)^{|\mathcal{S}_i|}$ to denote the feasibility region of $\theta_i$, and the feasibility region of $\theta$ is denoted as $\mathcal{X}^{\text{local}} := \mathcal{X}_i^{\text{local}} \times \cdots \times \mathcal{X}_n^{\text{local}}$.

**Lemma 7.** *If there is a function $\phi(s, a)$ such that for every agent $i$, $r_i(s_i, s_{-i}, a_i, a_{-i}) = \phi(s_i, s_{-i}, a_i, a_{-i}) + \psi_i(s_{-i}, a_{-i})$ where $\psi_i$ only depends on $s_{-i}, a_{-i}$, then this SG is a MPG, i.e., for any parameters $(\theta_i', \theta_{-i}), (\theta_i, \theta_{-i}) \in \mathcal{X}^{\text{local}}$, the equation in Definition 3 is satisfied.*

The proof is straightforward given the local structure of the MDP and the localized policies. This MPG enjoys nontrivial multi-agent application examples such as medium access control (Macua et al., 2018), dynamic congestion control (Bertrand et al., 2020), etc. Below we provide medium access control as one of the examples.

*Real application - medium access control.* We consider the discretized version of the dynamic medium access control game introduced in (Macua et al., 2018), where each agent is a user that tries to transmit data via a single transmission medium by injecting power to the wireless network. Each user's goal is to maximize its data rate and battery lifespan. If multiple users transmit at the same time, they will interfere with each other and decrease their data rate. Here user $i$'s state is $s_i \in \mathcal{S}_i = \{0, 1, \ldots, B_{i,\max}\}$, which denotes its own battery level, where $B_{i,\max}$ is its initial battery level. We use $\delta_i$ to denote its discharging factor. Its action $a_i \in \mathcal{A}_i = \{0, 1, \ldots, P_{i,\max}\}$ denotes the power injected to the network at each time step, where $P_{i,\max}$ is the maximum allowed power. The state transition is deterministic, describing the discharging process of the battery proportional to the transmission power, which is given by:

$$s_{i,t+1} = s_{i,t} - \delta_i a_{i,t}.$$

The stage reward of user $i$ is given by:

$$r_i(s, a) = \log\left(1 + \frac{|h_i|^2 a_i}{1 + \sum_{j \neq i} |h_j|^2 a_j}\right) + \alpha s_i,$$

where $h_i$ is the random fading channel coefficient for user $i$.

By noticing that $r_i(s,a) = \log\left(1 + \sum_{i=1}^{n}|h_i|^2 a_i\right) + \alpha\sum_i s_i - \log\left(1 + \sum_{j\neq i}|h_j|^2 a_j\right) - \alpha\sum_{j\neq i} s_j$, we can apply Lemma 7 to verify that the medium access control problem is indeed a MPG and that the potential function $\phi$ is given as:

$$\phi(s,a) = \log\left(1 + \sum_{i=1}^{n}|h_i|^2 a_i\right) + \alpha\sum_{i=1}^{n} s_i.$$

## C DERIVATION OF EQUATION (18) (CALCULATION OF $d_\theta$)

From the definition of $d_\theta$:

$$d_\theta(s) = \mathbb{E}_{s_0\sim\rho}(1-\gamma)\sum_{t=0}^{\infty}\gamma^t\mathrm{Pr}^\theta(s_t = s|s_0),$$

we have that:

$$d_\theta = (1-\gamma)\left(\rho + \gamma\overline{P_S^\theta}^\top \rho + \gamma^2\overline{P_S^\theta}^{2^\top}\rho + \cdots\right)$$

$$= (1-\gamma)(I + \gamma\overline{P_S^\theta} + \gamma^2\overline{P_S^\theta}^2\cdots)^\top\rho$$

$$= (1-\gamma)(I - \gamma\overline{P_S^\theta})^{-\top}\rho$$

## D PROOF OF LEMMA 2

*Proof.* According to policy gradient theorem Equation (6):

$$\frac{\partial J_i(\theta)}{\partial\theta_{s,a_i}} = \frac{1}{1-\gamma}\sum_{s'}\sum_{a'}d_\theta(s')\pi_\theta(a'|s')\frac{\partial\log\pi_\theta(a'|s')}{\partial\theta_{s,a_i}}Q_i^\theta(s,a)$$

Since for direct parameterization:

$$\frac{\partial\log\pi_\theta(a'|s')}{\partial\theta_{s,a_i}} = \frac{\partial\log\pi_{\theta_i}(a_i'|s')}{\partial\theta_{s,a_i}} = \mathbf{1}\{a_i' = a_i, s' = s\}\frac{1}{\theta_{s,a_i}}$$

$$= \mathbf{1}\{a_i' = a_i, s' = s\}\frac{1}{\pi_{\theta_i}(a_i|s)}$$

Thus we have that:

$$\frac{\partial J_i(\theta)}{\partial\theta_{s,a_i}} = \frac{1}{1-\gamma}\sum_{s'}\sum_{a'}d_\theta(s')\pi_\theta(a'|s')\mathbf{1}\{a_i' = a_i, s' = s\}\frac{1}{\pi_\theta(a_i|s)}Q_i^\theta(s,a)$$

$$= \frac{1}{1-\gamma}\sum_{a'_{-i}}d_\theta(s)\pi_{\theta_i}(a_i|s)\pi_{\theta_{-i}}(a'_{-i}|s)\frac{1}{\pi_\theta(a_i|s)}Q_i^\theta(s,a_i,a'_{-i})$$

$$= \frac{1}{1-\gamma}d_\theta(s)\sum_{a'_{-i}}\pi_{\theta_{-i}}(a'_{-i}|s)Q_i^\theta(s,a_i,a'_{-i})$$

$$= \frac{1}{1-\gamma}d_\theta(s)\overline{Q_i^\theta}(s,a_i)$$

$\square$

## E PROOFS OF LEMMA 3 AND THEOREM 1

Before giving proofs for Lemma 3 and Theorem 1, we first introduce the well-known *the performance difference lemma* (Kakade & Langford, 2002) in RL literature, which is helpful throughout. A proof is also provided for completeness.

**Lemma 8.** *(Performance difference lemma) For policies $\pi_\theta, \pi_{\theta'}$,*

$$J_i(\theta') - J_i(\theta) = \frac{1}{1-\gamma}\mathbb{E}_{s\sim d_{\theta'}}\mathbb{E}_{a\sim\pi_{\theta'}}A_i^\theta(s,a), \quad i=1,2,\ldots,n \tag{22}$$

*Proof.* (of performance difference lemma) From Bellman's equation we have that:

$$
\begin{aligned}
J_i(\theta') - J_i(\theta) &= \mathbb{E}_{s_0\sim\rho}V_i^{\theta'}(s_0) - \mathbb{E}_{s_0\sim\rho}V_i^\theta(s_0)\\
&= \left(\mathbb{E}_{s\sim\rho}\mathbb{E}_{a\sim\pi_{\theta'}(\cdot|s)}Q_i^\theta(s,a) - \mathbb{E}_{s\sim\rho}V_i^\theta(s)\right) + \left(\mathbb{E}_{s\sim\rho}V_i^{\theta'}(s) - \mathbb{E}_{s\sim\rho}\mathbb{E}_{a\sim\pi_{\theta'}(\cdot|s)}Q_i^\theta(s,a)\right)\\
&= \mathbb{E}_{s\sim\rho}\mathbb{E}_{a\sim\pi_{\theta'}(\cdot|s)}A_i^\theta(s,a) + \gamma\mathbb{E}_{s_0\sim\rho,s\sim\Pr^{\theta'}(s_1=\cdot|s_0)}\left[V_i^{\theta'}(s) - V_i^\theta(s)\right]\\
&= \mathbb{E}_{s\sim\rho}\mathbb{E}_{a\sim\pi_{\theta'}(\cdot|s)}A_i^\theta(s,a) + \gamma\mathbb{E}_{s_0\sim\rho,s\sim\Pr^{\theta'}(s_1=\cdot|s_0)}\mathbb{E}_{a\sim\pi_{\theta'}(\cdot|s)}A_i^\theta(s,a)\\
&\quad + \gamma^2\mathbb{E}_{s_0\sim\rho,s\sim\Pr^{\theta'}(s_2=\cdot|s_0)}\left[V_i^{\theta'}(s) - V_i^\theta(s)\right]\\
&= \cdots\\
&= \sum_{t=0}^\infty \mathbb{E}_{s_0\sim\rho,s\sim\Pr^{\theta'}(s_t=\cdot|s_0)}\mathbb{E}_{a\sim\pi_{\theta'}}\gamma^t A_i^\theta(s,a)\\
&= \frac{1}{1-\gamma}\mathbb{E}_{s\sim d_{\theta'}}\mathbb{E}_{a\sim\pi_{\theta'}}A_i^\theta(s,a)
\end{aligned}
$$

$\square$

*Proof.* (Lemma 3) According to performance difference lemma Equation (22):

$$
\begin{aligned}
J_i(\theta'_i,\theta_{-i}) - J_i(\theta_i,\theta_{-i}) &= \frac{1}{1-\gamma}\sum_{s,a}d_{\theta'}(s)\pi_{\theta'}(a|s)A_i^\theta(s,a)\\
&= \frac{1}{1-\gamma}\sum_{s,a_i}d_{\theta'}(s)\pi_{\theta'_i}(a_i|s)\sum_{a_{-i}}\pi_{\theta_{-i}}(a_{-i}|s)A_i^\theta(s,a_i,a_{-i}) \\
&= \frac{1}{1-\gamma}\sum_{s,a_i}d_{\theta'}(s)\pi_{\theta'_i}(a_i|s)\overline{A_i^\theta}(s,a_i).
\end{aligned}
\tag{23}
$$

According to the definition of 'averaged' advantage function:

$$\sum_{a_i}\pi_{\theta_i}(a_i|s)\overline{A_i^\theta}(s,a_i) = 0, \quad \forall s\in\mathcal{S}$$

which implies:

$$\max_{a_i\in\mathcal{A}_i}\overline{A_i^\theta}(s,a_i) \geq 0,$$

thus we have that:

$$
\begin{aligned}
J_i(\theta'_i,\theta_{-i}) - J_i(\theta_i,\theta_{-i}) &= \frac{1}{1-\gamma}\sum_{s,a_i}d_{\theta'}(s)\pi_{\theta'_i}(a_i|s)\overline{A_i^\theta}(s,a_i)\\
&\leq \frac{1}{1-\gamma}\sum_s d_{\theta'}(s)\max_{a_i\in\mathcal{A}_i}\overline{A_i^\theta}(s,a_i)\\
&= \frac{1}{1-\gamma}\sum_s \frac{d_{\theta'}(s)}{d_\theta(s)}d_\theta(s)\max_{a_i\in\mathcal{A}_i}\overline{A_i^\theta}(s,a_i)\\
&\leq \frac{1}{1-\gamma}\left\|\frac{d_{\theta'}}{d_\theta}\right\|_\infty\sum_s d_\theta(s)\max_{a_i\in\mathcal{A}_i}\overline{A_i^\theta}(s,a_i).
\end{aligned}
\tag{24}
$$

We can rewrite $\frac{1}{1-\gamma} \sum_s d_\theta(s) \max_{a_i \in \mathcal{A}_i} \overline{A_i^\theta}(s, a_i)$ as:

$$
\begin{aligned}
\frac{1}{1-\gamma} \sum_s d_\theta(s) \max_{a_i \in \mathcal{A}_i} \overline{A_i^\theta}(s, a_i) &= \frac{1}{1-\gamma} \max_{\overline{\theta}_i \in \mathcal{X}_i} \sum_{s,a_i} d_\theta(s) \pi_{\overline{\theta}_i}(a_i|s) \overline{A_i^\theta}(s, a_i) \\
&= \max_{\overline{\theta}_i \in \mathcal{X}_i} \sum_{s,a_i} (\pi_{\overline{\theta}_i}(a_i|s) - \pi_{\theta_i}(a_i|s)) \frac{1}{1-\gamma} d_\theta(s) \overline{A_i^\theta}(s, a_i) \\
&= \max_{\overline{\theta}_i \in \mathcal{X}_i} \sum_{s,a_i} (\pi_{\overline{\theta}_i}(a_i|s) - \pi_{\theta_i}(a_i|s)) \frac{1}{1-\gamma} d_\theta(s) \overline{Q_i^\theta}(s, a_i) \\
&= \max_{\overline{\theta}_i \in \mathcal{X}_i} (\overline{\theta}_i - \theta_i)^\top \nabla_{\theta_i} J_i(\theta).
\end{aligned}
\tag{25}
$$

Substituting this into Equation (24), we may conclude that

$$
J_i(\theta_i', \theta_{-i}) - J_i(\theta_i, \theta_{-i}) \le \left\| \frac{d_{\theta'}}{d_\theta} \right\|_\infty \max_{\overline{\theta}_i \in \mathcal{X}_i} (\overline{\theta}_i - \theta_i)^\top \nabla_{\theta_i} J_i(\theta)
$$

and this completes the proof. $\qquad\square$

*Proof.* (Theorem 1) The definition of a Nash equilibrium naturally implies first-order stationarity, because for any $\theta_i \in \mathcal{X}_i$:

$$
J_i((1-\eta)\theta_i^* + \eta\theta_i, \theta_{-i}^*) - J_i(\theta_i^*, \theta_{-i}^*) = \eta(\theta_i - \theta)^\top \nabla_{\theta_i} J_i(\theta^*) + o(\eta\|\theta_i - \theta_i^*\|) \le 0, \quad \forall \eta > 0
$$

Letting $\eta \to 0$ gives the first-order stationary condition:

$$
(\theta_i - \theta)^\top \nabla_{\theta_i} J_i(\theta^*) \le 0, \quad \forall \theta_i \in \mathcal{X}_i,
$$

It remains to be shown that all first-order stationary policies are Nash equilibria. From Assumption 1 we know that for any pair of parameters $\theta', \theta^*$, $\left\| \frac{d_{\theta'}}{d_{\theta^*}} \right\|_\infty < +\infty$.

Take $\theta' = (\theta_i', \theta_{-i}^*), \theta^* = (\theta_i^*, \theta_{-i}^*)$. According to Lemma 3, we have that for any first-order stationary policy $\theta^*$,

$$
J_i(\theta_i', \theta_{-i}^*) - J_i(\theta_i^*, \theta_{-i}^*) \le \left\| \frac{d_{\theta'}}{d_{\theta^*}} \right\|_\infty \max_{\overline{\theta}_i \in \mathcal{X}_i} (\overline{\theta}_i - \theta_i^*)^\top \nabla_{\theta_i} J_i(\theta^*) \le 0,
$$

which completes the proof. $\qquad\square$

# F  PROOF OF THEOREM 2 AND LEMMA 4

*Proof.* (Lemma 4) For a given strict NE $\theta^*$ randomly set:

$$
a_i^*(s) \in \arg\max_{a_i} \overline{A_i^{\theta^*}}(s, a_i),
$$

and set $\theta_i$ be:

$$
\theta_{s,a_i} = \mathbf{1}\{a_i = a_i^*(s)\}.
$$

And set $\theta := (\theta_i, \theta_{-i}^*)$ From performance difference lemma Equation (22):

$$
\begin{aligned}
J_i(\theta_i, \theta_{-i}^*) - J_i(\theta_i^*, \theta_{-i}^*) &= \sum_{s,a_i} d_\theta(s) \pi_{\theta_i}(s, a_i) \overline{A_i^{\theta^*}}(s, a_i) \\
&= \sum_s d_\theta(s) \max_{a_i} \overline{A_i^{\theta^*}}(s, a_i) \ge 0
\end{aligned}
$$

Because $\theta^*$ is a strict NE, thus the inequality above forces $\theta_i^* = \theta$, and that $\max_{a_i} \overline{A_i^{\theta^*}}(s, a_i) = 0$. The uniqueness of $\theta^*$ also implies uniqueness of $a_i^*(s)$, and thus,

$$
\overline{A_i^{\theta^*}}(s, a_i) < 0, \quad \forall a_i \ne a_i^*(s),
$$

which completes the proof of the lemma. $\qquad\square$

The proof of Theorem 2 relies on the following auxiliary lemma, whose proof we defer to Appendix M.

**Lemma 9.** *Let $\mathcal{X}$ denote the probability simplex of dimension $n$. Suppose $\theta \in \mathcal{X}, g \in \mathbb{R}^n$ and that there exists $i^* \in \{1, 2, \ldots, n\}$ and $\Delta > 0$ such that:*

$$\theta_{i^*} \geq \theta_i, \quad \forall i \neq i^*$$
$$g_{i^*} \geq g_i + \Delta, \quad \forall i \neq i^*.$$

*Let*

$$\theta' = Proj_{\mathcal{X}}(\theta + g),$$

*then:*

$$\theta'_{i^*} \geq \min\{1, \theta_{i^*} + \frac{\Delta}{2}\}$$

*Proof.* (Theorem 2) For a fixed agent $i$ and state $s$, the gradient play (Equation (5)) update rule of policy $\theta_{i,s}$ is given by:

$$\theta_{i,s}^{(t+1)} = Proj_{\Delta(|\mathcal{A}_i|)}(\theta_{i,s}^{(t)} + \frac{\eta}{1-\gamma} d_{\theta^{(t)}}(s)\overline{Q_i^{\theta^{(t)}}}(s, \cdot)), \tag{26}$$

where $\Delta(|\mathcal{A}_i|)$ denotes the probability simplex in $|\mathcal{A}_i|$-th dimension and $\overline{Q_i^{\theta^{(t)}}}(s, \cdot))$ is a $|\mathcal{A}_i|$-th dimensional vector with $a_i$-th element equals to $\overline{Q_i^{\theta^{(t)}}}(s, a_i))$. We will show that this update rule satisfies the conditions in Lemma 9, which will then allow us to prove that

$$D(\theta^{(t+1)}||\theta^*) \leq \max\{0, D(\theta^{(t)}||\theta^*) - \frac{\eta\Delta^{\theta^*}}{2}\}.$$

Letting $a_i^*(s)$ be the same definition as Equation (9), we have that:

$$\frac{1}{1-\gamma}d_{\theta^{(t)}}(s)\overline{Q_i^{\theta^{(t)}}}(s, a_i^*(s)) - \frac{1}{1-\gamma}d_{\theta^{(t)}}(s)\overline{Q_i^{\theta^{(t)}}}(s, a_i)$$

$$\geq \frac{1}{1-\gamma}d_{\theta^*}(s)\overline{Q_i^{\theta^*}}(s, a_i^*(s)) - \frac{1}{1-\gamma}d_{\theta^*}(s)\overline{Q_i^{\theta^*}}(s, a_i)$$

$$- \left| \frac{1}{1-\gamma}d_{\theta^*}(s)\overline{Q_i^{\theta^*}}(s, a_i^*(s)) - \frac{1}{1-\gamma}d_{\theta^{(t)}}(s)\overline{Q_i^{\theta^{(t)}}}(s, a_i^*(s)) \right|$$

$$- \left| \frac{1}{1-\gamma}d_{\theta^*}(s)\overline{Q_i^{\theta^*}}(s, a_i) - \frac{1}{1-\gamma}d_{\theta^{(t)}}(s)\overline{Q_i^{\theta^{(t)}}}(s, a_i) \right|$$

$$\geq \frac{1}{1-\gamma}d_{\theta^*}(s)\left(\overline{A_i^{\theta^*}}(s, a_i^*(s) - \overline{A_i^{\theta^*}}(s, a_i))\right) - 2\|\nabla_{\theta_i}J_i(\theta^{(t)}) - \nabla_{\theta_i}J_i(\theta^*)\| \tag{27}$$

$$\geq \Delta^{\theta^*} - \frac{4}{(1-\gamma)^3}\left(\sum_{i=1}^{n}|\mathcal{A}_i|\right)\|\theta^{(t)} - \theta^*\| \tag{28}$$

$$\geq \Delta^{\theta^*} - \frac{4}{(1-\gamma)^3}\left(\sum_{i=1}^{n}|\mathcal{A}_i|\right)\sum_{i=1}^{n}\sum_{s}\|\theta_{i,s}^{(t)} - \theta_{i,s}^*)\|_1$$

$$\geq \Delta^{\theta^*} - \frac{4}{(1-\gamma)^3}n|\mathcal{S}|\left(\sum_{i=1}^{n}|\mathcal{A}_i|\right)D(\theta^{(t)}||\theta^*),$$

where Equation (27) to Equation (28) uses smoothness property in Lemma 18.

We use proof of induction as supposed for $\ell \leq t - 1$, we have:

$$D(\theta^{(\ell+1)}||\theta^*) \leq \max\{D(\theta^{(\ell)}||\theta^*) - \frac{\eta\Delta^{\theta^*}}{2}, 0\},$$

thus

$$D(\theta^{(t)}||\theta^*) \leq D(\theta^{(0)}||\theta^*) \leq \frac{\Delta^{\theta^*}(1-\gamma)^3}{8n|\mathcal{S}|\left(\sum_{i=1}^{n}|\mathcal{A}_i|\right)}.$$

Then we can further conclude that:

$$(1-\gamma)d_{\theta^{(t)}}(s)\overline{Q_i^{\theta^{(t)}}}(s, a_i^*(s)) - (1-\gamma)d_{\theta^{(t)}}(s)\overline{Q_i^{\theta^{(t)}}}(s, a_i)$$

$$\geq \Delta^{\theta^*} - \frac{4}{(1-\gamma)^3}n|\mathcal{S}|\left(\sum_{i=1}^n |\mathcal{A}_i|\right)D(\theta^{(t)}||\theta^*)$$

$$\geq \frac{\Delta^{\theta^*}}{2}, \quad \forall\, a_i \neq a_i^*(s)$$

Additionally, for $D(\theta^{(t)}||\theta^*) \leq \frac{\Delta^{\theta^*}(1-\gamma)^3}{8n|\mathcal{S}|\left(\sum_{i=1}^n |\mathcal{A}_i|\right)}$, we may conclude:

$$\theta_{s,a_i^*(s)}^{(t)} \geq 1/2 \geq \theta_{s,a_i}^{(t)} \quad \forall a_i \neq a_i^*(s),$$

then by applying Lemma 9 to Equation (26) we have:

$$\theta_{s,a_i^*(s)}^{(t+1)} \geq \min\{1, \theta_{s,a_i^*(s)}^{(t)} + \frac{\eta\Delta^{\theta^*}}{4}\}$$

$$\implies \|\theta_{i,s}^{(t+1)} - \theta_{i,s}^*\|_1 = 2\left(1 - \theta_{s,a_i^*(s)}^{(t+1)}\right)$$

$$\leq \max\{0, \|\theta_{i,s}^{(t)} - \theta_{i,s}^*\|_1 - \frac{\eta\Delta^{\theta^*}}{2}\}, \quad \forall\, s \in \mathcal{S},\ i = 1, 2, \ldots, n$$

$$\implies D(\theta^{(t+1)}||\theta^*) \leq \max\{0, D(\theta^{(t)}||\theta^*) - \frac{\eta\Delta^{\theta^*}}{2}\},$$

which completes the proof. $\qquad\square$

## G  PROOF OF PROPOSITION 1

*Proof.* First of all, from the definition of NE, the global maximum of the potential function is a NE. We now show that this global maximum is a deterministic policy. From classical results (e.g. (Sutton & Barto, 2018)) we know that there is an optimal deterministic centralized policy

$$\pi^*(a = (a_1, \ldots, a_n)|s) = \mathbf{1}\{a = a^*(s) = (a_1^*(s), \ldots, a_n^*(s))\}$$

that maximizes:

$$\pi^* = \arg\max_{\pi:\mathcal{S}\to\Delta(\mathcal{A})} \mathbb{E}\left[\sum_{t=0}^\infty \gamma^t \phi(s_t, a_t)\big|\pi, s_0 = s\right].$$

We now show that this centralized policy can also be represented by direct distributed policy parameterization. Set $\theta^*$ as:

$$\pi_{\theta_i^*}(a_i|s) = \mathbf{1}\{a_i = a_i^*(s)\},$$

then:

$$\pi^*(a|s) = \prod_{i=1}^n \pi_{\theta_i^*}(a_i|s)$$

Since $\pi^*$ globally maximizes the discounted summation of potential function $\phi$ among centralized policies, which includes all possible direct distributedly parameterized policies, $\theta^*$ also maximizes the total potential function $\Phi$ globally among all direct distributed parameterization, which completes the proof. $\qquad\square$

## H  PROOF OF THEOREM 3

### H.1  USEFUL OPTIMIZATION LEMMAS

**Lemma 10.** *Let $\Phi(\theta)$ be $\beta$-smooth in $\theta$, define gradient mapping:*

$$G^\eta(\theta) := \frac{1}{\eta}\left(Proj_\mathcal{X}(\theta + \eta\nabla\Phi(\theta)) - \theta\right).$$

*The update rule for projected gradient is:*

$$\theta^+ = \theta + \eta G^\eta(\theta) = Proj_\mathcal{X}(\theta + \eta\nabla\Phi(\theta)).$$

*Then:*

$$(\theta' - \theta^+)^\top \nabla\Phi(\theta^+) \leq (1 + \eta\beta)\|G^\eta(\theta)\|\|\theta' - \theta^+\| \quad \forall \theta' \in \mathcal{X}.$$

*Proof.* By a standard property of Euclidean projections onto a convex set, we get that

$$
\begin{aligned}
&(\theta + \eta\nabla\Phi(\theta) - \theta^+)^\top(\theta' - \theta^+) \leq 0 \\
\implies & \eta\nabla\Phi(\theta)^\top(\theta' - \theta^+) + (\theta - \theta^+)^\top(\theta' - \theta^+) \leq 0 \\
\implies & \eta\nabla\Phi(\theta)^\top(\theta' - \theta^+) - \eta G^\eta(\theta)^\top(\theta' - \theta^+) \leq 0 \\
\implies & \nabla\Phi(\theta)^\top(\theta' - \theta^+) \leq \|G^\eta(\theta)\|\|\theta' - \theta^+\| \\
\implies & \nabla\Phi(\theta^+)^\top(\theta' - \theta^+) \leq \|G^\eta(\theta)\|\|\theta' - \theta^+\| + (\nabla\Phi(\theta^+) - \nabla\Phi(\theta))^\top(\theta' - \theta^+)\| \\
& \qquad\qquad\qquad\quad \leq \|G^\eta(\theta)\|\|\theta' - \theta^+\| + \beta\|\theta^+ - \theta\|\|\theta' - \theta^+\| \\
& \qquad\qquad\qquad\quad = (1 + \eta\beta)\|G^\eta(\theta)\|\|\theta' - \theta^+\|
\end{aligned}
$$

$\square$

**Lemma 11.** *(Sufficient ascent) Suppose $\Phi(\theta)$ is $\beta$-smooth. Let $\theta^+ = Proj_\mathcal{X}(\theta + \eta\nabla\Phi(\theta))$. Then for $\eta \leq \frac{1}{\beta}$,*

$$\Phi(\theta^+) - \Phi(\theta) \geq \frac{\eta}{2}\|G^\eta(\theta)\|^2$$

*Proof.* From the smoothness property we have that:

$$\Phi(\theta^+) - \Phi(\theta) \geq \nabla_\theta\Phi(\theta)^\top(\theta^+ - \theta) - \frac{\beta}{2}\|\theta^+ - \theta\|^2 \tag{29}$$

Since $\theta^+ = Proj_\mathcal{X}(\theta + \eta\nabla\Phi(\theta))$, we have that:

$$(\theta + \eta\nabla\Phi(\theta) - \theta^+)^\top(\theta' - \theta^+) \leq 0, \ \ \forall \theta' \in \mathcal{X}$$

take $\theta' = \theta$, we get:

$$\nabla\Phi(\theta)^\top(\theta^+ - \theta) \geq \frac{1}{\eta}\|\theta^+ - \theta\|^2.$$

Thus:

$$
\begin{aligned}
\Phi(\theta^+) - \Phi(\theta) &\geq \nabla_\theta\Phi(\theta)^\top(\theta^+ - \theta) - \frac{\beta}{2}\|\theta^+ - \theta\|^2 \\
&\geq (\frac{1}{\eta} - \frac{\beta}{2})\|\theta^+ - \theta\|^2 \\
&\geq \frac{1}{2\eta}\|\theta^+ - \theta\|^2 \\
&= \frac{\eta}{2}\|G^\eta(\theta)\|^2,
\end{aligned}
$$

which completes the proof. $\square$

**Lemma 12.** *(Corollary of Lemma 11) For $\Phi(\theta)$ that is $\beta$ smooth and bounded $\Phi_{\min} \leq \Phi(\theta) \leq \Phi_{\max}$, running projected gradient ascent:*

$$\theta^{(t+1)} = Proj_\mathcal{X}(\theta + \eta\nabla\Phi(\theta^{(t)}))$$

*with $\eta = \frac{1}{\beta}$, will guarantee that:*

$$\lim_{t\to+\infty} \|G^\eta(\theta^{(t)})\| = 0.$$

*Further, we have that:*

$$\frac{1}{T}\sum_{t=0}^{T-1}\|G^\eta(\theta^{(t)})\|^2 \leq \frac{2\beta(\Phi_{\max} - \Phi_{\min})}{T} \tag{30}$$

*Proof.* From Lemma 11 we have:

$$\Phi(\theta^{(t+1)}) - \Phi(\theta^{(t)}) \geq \frac{1}{2\beta}\|G^\eta(\theta^{(t)})\|^2 \geq 0$$

Thus $\Phi(\theta^{(t)})$ is non decreasing, and since it is bounded, we know that $\Phi(\theta^{(t)})$ asymptotically convergence to some value $\Phi^*$, and thus show that

$$\lim_{t\to\infty}\|G^\eta(\theta^{(t)})\| = 0.$$

Additionally, from Equation (30),

$$\Phi(\theta^{(T)}) - \Phi(\theta^{(0)}) \geq \sum_{t=0}^{T-1}\frac{1}{2\beta}\|G^\eta(\theta^{(t)})\|^2$$

$$\implies \frac{1}{T}\sum_{t=0}^{T-1}\|G^\eta(\theta^{(t)})\|^2 \leq \frac{2\beta(\Phi_{\max} - \Phi_{\min})}{T},$$

which completes the proof. $\qquad\square$

### H.2 PROOF OF THEOREM 3

*Proof.* Recall the definition of gradient mapping:

$$G^\eta(\theta) = \frac{1}{\eta}\left(Proj_{\mathcal{X}}(\theta + \eta\nabla\Phi(\theta)) - \theta\right).$$

From gradient domination property Equation (8) we have that:

$$\texttt{NE-gap}_i(\theta^{(t+1)}) = \max_{\theta'_i \in \mathcal{X}_i} J_i(\theta'_i.\theta_{-i}^{(t+1)}) - J_i(\theta_i^{(t+1)}, \theta_{-i}^{(t+1)})$$

$$\leq \max_{\theta'_i \in \mathcal{X}_i}\left\|\frac{d_{(\theta'_i.\theta_{-i}^{(t+1)})}}{d_{(\theta_i^{(t+1)},\theta_{-i}^{(t+1)})}}\right\|_\infty \max_{\overline{\theta}_i \in \mathcal{X}_i}\left(\overline{\theta}_i - \theta_i^{(t+1)}\right)^\top \nabla_{\theta_i}J_i(\theta^{(t+1)})$$

$$\leq M\max_{\overline{\theta}_i \in \mathcal{X}_i}\left(\overline{\theta}_i - \theta_i^{(t+1)}\right)^\top \nabla_{\theta_i}\Phi(\theta^{(t+1)})$$

$$\leq M(1 + \eta\beta)\max_{\overline{\theta}_i \in \mathcal{X}_i}\|\overline{\theta}_i - \theta_i^{(t+1)}\|\|G^\eta(\theta^{(t)})\|$$

$$\leq M(1 + \eta\beta)2\sqrt{|\mathcal{S}|}\|G^\eta(\theta^{(t)})\|$$

$$= 4M\sqrt{|\mathcal{S}|}\|G^\eta(\theta^{(t)})\|$$

where the last step follows as $\|\overline{\theta}_i - \theta_i^{(t+1)}\| \leq 2\sqrt{|\mathcal{S}|}$. Thus

$$\texttt{NE-gap}(\theta^{(t+1)}) \leq 4M\sqrt{|\mathcal{S}|}\|G^\eta(\theta^{(t)})\|$$

Then from Lemma 12 we have that:

$$\lim_{t\to\infty}\|G^\eta(\theta^{(t)})\| = 0 \implies \lim_{t\to\infty}\texttt{NE-gap}(\theta^{(t)}) = 0,$$

and that:

$$\frac{1}{T}\sum_{t=0}^{T-1}\|G^\eta(\theta^{(t)})\|^2 \leq \frac{2\beta(\Phi_{\max} - \Phi_{\min})}{T}$$

$$\implies \frac{1}{T}\sum_{t=1}^{T}\texttt{NE-gap}(\theta^{(t)})^2 \leq \frac{32\beta M^2|\mathcal{S}|(\Phi_{\max} - \Phi_{\min})}{T}$$

we can get our required bound of $\epsilon$ if we set:

$$\frac{32\beta M^2|\mathcal{S}|(\Phi_{\max} - \Phi_{\min})}{T} \leq \epsilon^2,$$

or equivalently

$$T \geq \frac{32M^2\beta(\Phi_{\max} - \Phi_{\min})|\mathcal{S}|}{\epsilon^2}$$
$$= \frac{64M^2(\Phi_{\max} - \Phi_{\min})|\mathcal{S}|\sum_i |\mathcal{A}_i|}{\epsilon^2(1-\gamma)^3},$$

which completes the proof. □

## I    PROOF OF THEOREM 4

*Proof.* (of the first claim) The proof requires knowledge of Lemma 4 in Section 3 thus we would recommend readers to first go through Lemma 4 first. The lemma immediately leads to the conclusion that a strict NE $\theta^*$ should be deterministic. Let $a_i^*(s), \Delta_i^{\theta^*}(s), \Delta_i^{\theta^*}$ be the same definition as Equation (9) Equation (10) respectively.

For any $\theta \in \mathcal{X}$, Taylor expansion suggests that:

$$\begin{aligned}
\Phi(\theta) - \Phi(\theta^*) &= (\theta - \theta^*)^\top \nabla \Phi(\theta^*) + o(\|\theta - \theta^*\|) \\
&= \sum_i (\theta_i - \theta_i^*)^\top \nabla_{\theta_i} J_i(\theta^*) + o(\|\theta - \theta^*\|) \\
&= \frac{1}{1-\gamma} \sum_i \sum_s \sum_{a_i} d_{\theta^*}(s) \overline{A_i^{\theta^*}}(s, a_i)(\theta_{s,a_i} - \theta_{s,a_i}^*) + o(\|\theta - \theta^*\|) \\
&\leq -\frac{1}{1-\gamma} \sum_i \sum_s d_{\theta^*}(s) \Delta_i^{\theta^*}(s) \left( \sum_{a_i \neq a_i^*(s)} (\theta_{s,a_i} - \theta_{s,a_i}^*) \right) + o(\|\theta - \theta^*\|) \\
&= -\frac{1}{1-\gamma} \sum_i \sum_s d_{\theta^*}(s) \Delta_i^{\theta^*}(s) \frac{1}{2} \|\theta_{i,s} - \theta_{i,s}^*\|_1 + o(\|\theta - \theta^*\|) \\
&\leq -\frac{\Delta^{\theta^*}}{2} \sum_i \sum_s \|\theta_{i,s} - \theta_{i,s}^*\|_1 + o(\|\theta - \theta^*\|) \\
&\leq -\frac{\Delta^{\theta^*}}{2} \|\theta - \theta^*\| + o(\|\theta - \theta^*\|).
\end{aligned}$$

Thus for $\|\theta - \theta^*\|$ sufficiently small,

$$\Phi(\theta) - \Phi(\theta^*) < 0 \text{ holds,}$$

this suggests that strict NEs are strict local maxima. We now show that this also holds vice versa.

Strict local maxima satisfy first-order stationarity by definition, and thus by Theorem 1 they are also NEs, we only need to show that they are strict. We prove by contradiction, suppose that there exists a local maximum $\theta^*$ such that it is non-strict NE, i.e., there exists $\theta_i' \in \mathcal{X}_i, \theta_i' \neq \theta_i^*$ such that:

$$J_i(\theta_i', \theta_{-i}^*) = J_i(\theta_i^*, \theta_{-i}^*)$$

According to Equation (25) and first-order stationarity of $\theta^*$:

$$\frac{1}{1-\gamma} \sum_s d_{\theta^*}(s) \max_{a_i \in \mathcal{A}_i} \overline{A_i^{\theta^*}}(s, a_i) = \max_{\overline{\theta}_i \in \mathcal{X}_i} (\overline{\theta}_i - \theta_i^*)^\top \nabla_{\theta_i} J_i(\theta^*) \leq 0.$$

Since $\max_{a_i \in \mathcal{A}_i} \overline{A_i^\theta}(s, a_i) \geq 0$ for all $\theta$, we may conclude:

$$\max_{a_i \in \mathcal{A}_i} \overline{A_i^{\theta^*}}(s, a_i) = 0, \ \ \forall\, s \in \mathcal{S}.$$

We denote $\theta' := (\theta_i', \theta_{-i^*})$, according to Equation (23)

$$0 = J_i(\theta_i', \theta_{-i}^*) - J_i(\theta_i^*, \theta_{-i}^*) = \frac{1}{1-\gamma} \sum_{s,a_i} d_{\theta'}(s) \pi_{\theta_i'}(a_i|s) \overline{A_i^{\theta^*}}(s, a_i) \leq 0.$$

Since $d_{\theta'}(s) > 0$, $\forall\, s$, this further implies that

$$\sum_{a_i} \pi_{\theta_i'}(a_i|s)\overline{A_i^{\theta^*}}(s, a_i) = 0, \ \ \forall\, s \in \mathcal{S},$$

i.e., $\pi_{\theta_i'}(a_i|s)$ is nonzero only if $\overline{A_i^{\theta^*}}(s, a_i) = 0$. Define $\theta_i^\eta := \eta\theta_i' + (1-\eta)\theta_i^*$, then

$$\sum_{a_i} \pi_{\theta_i^\eta}(a_i|s)\overline{A_i^{\theta^*}}(s, a_i) = 0, \ \ \forall\, s \in \mathcal{S}.$$

Thus let $\theta^\eta := (\theta_i^\eta, \theta_{-i}^*)$

$$J_i(\theta_i^\eta, \theta_{-i}^*) - J_i(\theta_i^*, \theta_{-i}^*) = \frac{1}{1-\gamma}\sum_{s,a_i} d_{\theta^\eta}(s)\pi_{\theta_i^\eta}(a_i|s)\overline{A_i^{\theta^*}}(s, a_i) = 0.$$

Since $\|\theta_i^\eta - \theta_i^*\| \to 0$ as $\eta \to 0$, this contradicts the assumption that $\theta^*$ is a strict local maximum. This suggests that all strict local maxima are strict NEs, which completes the proof. $\qquad\square$

*Proof.* (of the second claim) First, we define the corresponding value function, $Q$-function and advantage function for potential function $\phi$.

$$V_\phi^\theta(s) := \mathbb{E}\left[\sum_{t=0}^\infty \gamma^t \phi(s_t, a_t)\,\middle|\, \pi = \theta, s_0 = s\right]$$

$$Q_\phi^\theta(s, a) := \left[\sum_{t=0}^\infty \gamma^t \phi(s_t, a_t)\,\middle|\, \pi = \theta, s_0 = s, a_0 = a\right]$$

$$A_\phi^\theta(s, a) := Q_\phi^\theta(s, a) - V_\phi^\theta(s).$$

For an index set $\mathcal{I} \subseteq \{1, 2, \ldots, n\}$ we define the following averaged advantage potential function of index set $\mathcal{I}$ as:

$$\overline{A_{\phi,\mathcal{I}}^\theta}(s, a_\mathcal{I}) := \sum_{a_{-\mathcal{I}}} A_\phi^\theta(s, a_\mathcal{I}, a_{-\mathcal{I}}).$$

We choose an index set $\mathcal{I} \subseteq \{1, 2, \ldots, n\}$ such that there exists $s^*, a_\mathcal{I}^*$ such that:

$$\overline{A_{\phi,\mathcal{I}}^{\theta^*}}(s^*, a_\mathcal{I}^*) > 0, \tag{31}$$

and that for any other index set $\mathcal{I}'$ with smaller cardinality:

$$\overline{A_{\phi,\mathcal{I}'}^{\theta^*}}(s, a_{\mathcal{I}'}) \le 0, \ \ \forall\, s, a_{\mathcal{I}'}, \ \ \forall\, |\mathcal{I}'| < |\mathcal{I}|. \tag{32}$$

Because $\Phi$ is not a constant, this guarantees the existence of such an index set $\mathcal{I}$. Further, since

$$\sum_{a_{\mathcal{I}'}} \pi_{\theta_{\mathcal{I}'}^*}(a_{\mathcal{I}'}|s)\overline{A_{\phi,\mathcal{I}'}^{\theta^*}}(s, a_{\mathcal{I}'}) = 0, \ \ \forall\, s,$$

and that $\theta^*$ is fully-mixed, we have that:

$$\overline{A_{\phi,\mathcal{I}'}^{\theta^*}}(s, a_{\mathcal{I}'}) = 0, \ \ \forall\, s, a_{\mathcal{I}'}, \ \ \forall\, |\mathcal{I}'| < |\mathcal{I}|. \tag{33}$$

We set $\theta := (\theta_\mathcal{I}, \theta_{-\mathcal{I}}^*)$, where $\theta_\mathcal{I}$ is a convex combination of $\theta_\mathcal{I}^*, \theta_\mathcal{I}' \in \mathcal{X}$:

$$\theta_\mathcal{I} = (1-\eta)\theta_\mathcal{I}^* + \eta\theta_\mathcal{I}', \ \ \eta > 0.$$

According to performance difference lemma Equation (22) we have:

$$(1-\gamma)\left(\Phi(\theta_\mathcal{I}, \theta_{-\mathcal{I}}^*) - \Phi(\theta_\mathcal{I}^*, \theta_{-\mathcal{I}}^*)\right) = \sum_{s,a_\mathcal{I}} d_\theta(s)\pi_{\theta_\mathcal{I}}(a_\mathcal{I}|s)\overline{A_{\phi,\mathcal{I}}^{\theta^*}}(s, a_\mathcal{I})$$

$$= \sum_{s,a_\mathcal{I}} d_\theta(s)\prod_{i\in\mathcal{I}}\left((1-\eta)\pi_{\theta_i^*}(a_i|s) + \eta\pi_{\theta_i'}(a_i|s)\right)\overline{A_{\phi,\mathcal{I}}^{\theta^*}}(s, a_\mathcal{I})$$

$$= \sum_{s,a_\mathcal{I}} d_\theta(s)\left((1-\eta)\pi_{\theta_{i_0}^*}(a_{i_0}|s) + \eta\pi_{\theta_{i_0}'}(a_{i_0}|s)\right)\prod_{i\in\mathcal{I}\backslash\{i_0\}}\left((1-\eta)\pi_{\theta_i^*}(a_i|s) + \eta\pi_{\theta_i'}(a_i|s)\right)\overline{A_{\phi,\mathcal{I}}^{\theta^*}}(s, a_\mathcal{I}), \ \ (\forall\, i_0 \in \mathcal{I})$$

$$= (1 - \eta) \sum_{s, a_\mathcal{I}} d_\theta(s) \prod_{i \in \mathcal{I} \setminus \{i_0\}} \left( (1-\eta)\pi_{\theta_i^*}(a_i|s) + \eta\pi_{\theta_i'}(a_i|s) \right) \overline{A_{\phi, \mathcal{I} \setminus \{i_0\}}^{\theta^*}}(s, a_{\mathcal{I} \setminus \{i_0\}})$$

$$+ \eta \sum_{s, a_\mathcal{I}} d_\theta(s)\pi_{\theta_{i_0}'}(a_{i_0}|s) \prod_{i \in \mathcal{I} \setminus \{i_0\}} \left( (1-\eta)\pi_{\theta_i^*}(a_i|s) + \eta\pi_{\theta_i'}(a_i|s) \right) \overline{A_{\phi, \mathcal{I}}^{\theta^*}}(s, a_\mathcal{I}).$$

According to Equation (33), we know that:

$$\overline{A_{\phi, \mathcal{I} \setminus \{i_0\}}^{\theta^*}}(s, a_{\mathcal{I} \setminus \{i_0\}}) = 0,$$

thus

$$(1 - \gamma)\left(\Phi(\theta_\mathcal{I}, \theta_{-\mathcal{I}}^*) - \Phi(\theta_\mathcal{I}^*, \theta_{-\mathcal{I}}^*)\right) =$$
$$\eta \sum_{s, a_\mathcal{I}} d_\theta(s)\pi_{\theta_{i_0}'}(a_{i_0}|s) \prod_{i \in \mathcal{I} \setminus \{i_0\}} \left( (1-\eta)\pi_{\theta_i^*}(a_i|s) + \eta\pi_{\theta_i'}(a_i|s) \right) \overline{A_{\phi, \mathcal{I}}^{\theta^*}}(s, a_\mathcal{I}).$$

Applying similar procedures recursively and using the fact that:

$$\overline{A_{\phi, \mathcal{I} \setminus \{i\}}^{\theta^*}}(s, a_{\mathcal{I} \setminus \{i\}}) = 0, \;\; \forall\, i \in \mathcal{I},$$

we get:

$$\Phi(\theta_\mathcal{I}, \theta_{-\mathcal{I}}^*) - \Phi(\theta_\mathcal{I}^*, \theta_{-\mathcal{I}}^*) = \frac{\eta^{|\mathcal{I}|}}{1 - \gamma} \sum_{s, a_\mathcal{I}} d_\theta(s) \prod_{i \in \mathcal{I}} \pi_{\theta_i'}(a_i|s) \overline{A_{\phi, \mathcal{I}}^{\theta^*}}(s, a_\mathcal{I}).$$

Set $\pi_{\theta_i'}(a_i|s)$ as:

$$\pi_{\theta_i'}(a_i|s^*) = \begin{cases} 1 & a_i = a_i^* \\ 0 & \text{otherwise} \end{cases}$$

$$\pi_{\theta_i'}(a_i|s) = \pi_{\theta_i^*}(a_i|s), \quad s \neq s^*,$$

where $s^*, a_i^*$ are defined in Equation (31). Then:

$$\Phi(\theta_\mathcal{I}, \theta_{-\mathcal{I}}^*) - \Phi(\theta_\mathcal{I}^*, \theta_{-\mathcal{I}}^*) = \frac{\eta^{|\mathcal{I}|}}{1 - \gamma} d_\theta(s^*) \overline{A_{\phi, \mathcal{I}}^{\theta^*}}(s^*, a_\mathcal{I}^*) > 0,$$

which completes the proof. $\qquad\square$

## J  BOUNDING THE GRADIENT ESTIMATION ERROR OF ALGORITHM 1

The accuracy of gradient estimation is essential in the sample-based algorithm 1. In this section, we will give a high probability bound of the estimation error, which is stated in the following theorem:

**Theorem 6.** *(Error bound for gradient estimation) Assume that the stochastic game that satisfies Assumption 3. In Algorithm 1, for*

$$T_J \geq \frac{32\tau(1 + \alpha)^2 |\mathcal{S}|^3 \sum_i |\mathcal{A}_i| \max_i |\mathcal{A}_i|^2}{(1 - \gamma)^6 \epsilon_g^2 \alpha^2 \sigma_S^2} \log\left( \frac{16\tau T_G |\mathcal{S}|^2 \sum_i |\mathcal{A}_i|}{\delta} \right) + 1,$$

*with probability at least $1 - \delta$, we have:*

$$\|\widehat{\nabla}\Phi(\theta^{(k)}) - \nabla\Phi(\theta^{(k)})\|_2 \leq \epsilon_g, \;\; \forall\, 0 \leq k \leq T_G - 1.$$

The proof of the theorem includes bounding the estimation error of $\overline{Q_i^\theta}$ (J.1) and $d_\theta$ (J.2). Let's first introduce the definition of 'sufficient exploration' which is going to play an important role in this section.

In the main text Assumption 3 we have introduced $(\tau, \sigma_S)$-sufficient exploration on states. In this section we introduce a similar definition $(\tau, \sigma)$-sufficient exploration:

**Definition 6.** *($(\tau, \sigma)$-Sufficient Exploration) A stochastic game and a policy $\theta$ is said to satisfy $(\tau, \sigma)$-sufficient exploration condition if there exists positive integer $\tau$ and $\sigma \in (0, 1)$ such that for policy $\theta$ and any initial state-action pair $(s, a_i)$, $\forall i$, we have*

$$\Pr(s_\tau, a_{i,\tau}|s_0 = s, a_0 = a) \geq \sigma, \;\; \forall s_\tau, a_{i,\tau}$$

Note that '$(\tau, \sigma)$-sufficient exploration' is a stronger condition compared with '$(\tau, \sigma_S)$-sufficient exploration on states'. Additionally it is not hard to verify that for any stochastic game that satisfies $(\tau, \sigma_S)$-sufficient exploration on states, and any $\theta \in \mathcal{X}^\alpha$, it will also satisfy $(\tau, \frac{\alpha\sigma_S}{\max_i |\mathcal{A}_i|})$-sufficient exploration condition.

### J.1 BOUNDING THE ESTIMATION ERROR OF THE AVERAGED-Q FUNCTION

We first state the main theorem in this subsection:

**Theorem 7.** *(Estimation error of averaged Q-functions) Assume that the stochastic game with policy $\theta$ satisfies $(\sigma, \tau)$-sufficient exploration condition (Definition 6), then for a fixed $i$, running Algorithm 1 will guarantee that:*

$$\Pr\left(\|\widehat{Q_i^\theta} - \overline{Q_i^\theta}\|_\infty \geq \epsilon\right) \leq 4\tau|\mathcal{S}|^2|\mathcal{A}_i| \exp\left(-\frac{(1-\gamma)^4\epsilon^2\sigma^2\lfloor\frac{T}{\tau}\rfloor}{32|\mathcal{S}|^2}\right),$$

*further:*

$$\Pr\left(\|\widehat{Q_i^\theta} - \overline{Q_i^\theta}\|_\infty \geq \epsilon, \exists i\right) \leq 8\tau|\mathcal{S}|^2\left(\sum_{i=1}^n |\mathcal{A}_i|\right) \exp\left(-\frac{(1-\gamma)^4\epsilon^2\sigma^2\lfloor\frac{T}{\tau}\rfloor}{32|\mathcal{S}|^2}\right),$$

*i.e., when*

$$T_J \geq \frac{32\tau|\mathcal{S}|^2}{(1-\gamma)^4\epsilon^2\sigma^2} \log\left(\frac{8\tau|\mathcal{S}|^2\sum_i |\mathcal{A}_i|}{\delta}\right) + \tau$$

*with probability at least $1 - \delta$, $\|\widehat{Q_i^\theta} - \overline{Q_i^\theta}\|_\infty \leq \epsilon, \forall i$*

In the following, we will introduce some lemmas which will play an important role in bounding the estimation error of the averaged-Q function:

**Lemma 13.** *Assume that the stochastic game with policy $\theta$ satisfies $(\sigma, \tau)$-sufficient exploration condition (Definition 6), then fix $s', s, a_i$, for $\epsilon \leq 1$,*

$$\Pr\left(\left|\widehat{P_i^\theta}(s'|s,a_i) - \overline{P_i^\theta}(s'|s,a_i)\right| \geq \epsilon\right) \leq 4\tau \exp\left(-\frac{\epsilon^2\sigma^2\lfloor\frac{T}{\tau}\rfloor}{32}\right)$$

*Proof.* According to the definition of $\widehat{P_i^\theta}$, we have that

$$\left\{\widehat{P_i^\theta}(s'|s,a_i) - \overline{P_i^\theta}(s'|s,a_i) \geq \epsilon\right\}$$

$$\subseteq \left\{\sum_{t=0}^{T-1}\left(\mathbf{1}\{s_{t+1}=s', s_t=s, a_{i,t}=a_i\} - (\overline{P_i^\theta}(s'|s,a_i)+\epsilon)\mathbf{1}\{s_t=s, a_{i,t}=a_i\}\right) \geq 0\right\}$$

$$\cup\left\{\sum_{t=0}^{T-1}\mathbf{1}\{s_t=s, a_{i,t}=a_i\} = 0\right\}$$

$$\subseteq \bigcup_{m=0}^{\tau-1}\left\{\sum_{k=0}^{\lfloor\frac{T-1-m}{\tau}\rfloor}\left(\mathbf{1}\{s_{k\tau+m+1}=s', s_{k\tau+m}=s, a_{i,k\tau+m}=a_i\} - (\overline{P_i^\theta}(s'|s,a_i)+\epsilon)\mathbf{1}\{s_{k\tau+m}=s, a_{i,k\tau+m}=a_i\}\right) \geq 0\right\}$$

$$\bigcup_{m=0}^{\tau-1}\left\{\sum_{k=0}^{\lfloor\frac{T-1-m}{\tau}\rfloor}\mathbf{1}\{s_{k\tau+m}=s, a_{i,k\tau+m}=a_i\} = 0\right\}$$

Let:

$$A_m := \left\{\sum_{k=0}^{\lfloor\frac{T-1-m}{\tau}\rfloor}\left(\mathbf{1}\{s_{k\tau+m+1}=s', s_{k\tau+m}=s, a_{i,k\tau+m}=a_i\} - (\overline{P_i^\theta}(s'|s,a_i)+\epsilon)\mathbf{1}\{s_{k\tau+m}=s, a_{i,k\tau+m}=a_i\}\right) \geq 0\right\}$$

$$A'_m := \left\{\sum_{k=0}^{\lfloor\frac{T-1-m}{\tau}\rfloor}\mathbf{1}\{s_{k\tau+m}=s, a_{i,k\tau+m}=a_i\} = 0\right\}$$

$$X_{k,m} := \mathbf{1}\{s_{k\tau+m+1}=s', s_{k\tau+m}=s, a_{i,k\tau+m}=a_i\} - (\overline{P_i^\theta}(s'|s,a_i)+\epsilon)\mathbf{1}\{s_{k\tau+m}=s, a_{i,k\tau+m}=a_i\}$$

$$X'_{k,m} := \mathbf{1}\{s_{k\tau+m}=s, a_{i,k\tau+m}=a_i\}$$

$$Y_{k,m} := X_{k,m} - \mathbb{E}[X_{k,m}|\mathcal{F}_{(k-1)\tau+m}]$$
$$Y'_{k,m} := X'_{k,m} - \mathbb{E}[X'_{k,m}|\mathcal{F}_{(k-1)\tau+m}]$$

Then $\{Y_{k,m}\}_{k=0}^{\lfloor\frac{T-1-m}{\tau}\rfloor}$ is a martingale difference sequence. Because $\epsilon \leq 1$, it is easy to verify that $|X_{k,m}| \leq 2, |X'_{k,m}| \leq 1$. We have that:

$$|Y_{k,m}| \leq |X_{k,m}| + \mathbb{E}[|X_{k,m}||\mathcal{F}_{(k-1)\tau+m}] \leq 4, \quad |Y'_{k,m}| \leq |X'_{k,m}| + \mathbb{E}[|X'_{k,m}||\mathcal{F}_{(k-1)\tau+m}] \leq 2.$$

Further,

$$\mathbb{E}[X'_{k,m}|\mathcal{F}_{(k-1)\tau+m}] = \mathbb{E}[\mathbf{1}\{s_{k\tau+m} = s, a_{i,k\tau+m} = a_i\}|\mathcal{F}_{(k-1)\tau+m}] \geq \sigma,$$

and that

$$\mathbb{E}[X_{k,m}|\mathcal{F}_{(k-1)\tau+m}]$$
$$= \mathbb{E}[\mathbf{1}\{s_{k\tau+m+1} = s', s_{k\tau+m} = s, a_{i,k\tau+m} = a_i\} - (\overline{P_i^\theta}(s'|s,a_i) + \epsilon)\mathbf{1}\{s_{k\tau+m} = s, a_{i,k\tau+m} = a_i\}|\mathcal{F}_{(k-1)\tau+m}] \tag{34}$$
$$= -\epsilon\mathbb{E}[\mathbf{1}\{s_{k\tau+m} = s, a_{i,k\tau+m} = a_i\}|\mathcal{F}_{(k-1)\tau+m}] \leq -\epsilon\sigma. \tag{35}$$

To move from equation (34) to equation (35), we used the fact that:

$$\mathbb{E}[\mathbf{1}\{s_{t+1} = s', s_t = s, a_{i,t+m} = a_i\}|\mathcal{F}_{t-1}] = P(s|s_{t-1}, a_{t-1})\sum_{a_{-i}} \pi_\theta(a_i, a_{-i}|s)P(s'|s, a_i, a_{-i})$$
$$= P(s|s_{t-1}, a_{t-1})\pi_{\theta_i}(a_i|s)\sum_{a_{-i}} \pi_{\theta_{-i}}(a_{-i}|s)P(s'|s, a_i, a_{-i})$$
$$= P(s|s_{t-1}, a_{t-1})\pi_{\theta_i}(a_i|s)\overline{P_i^\theta}(s'|s, a_i)$$
$$= \mathbb{E}[\overline{P_i^\theta}(s'|s, a_i)\mathbf{1}\{s_t = s, a_{i,t+m} = a_i\}|\mathcal{F}_{t-1}]$$

and the inequality in equation (35) is derived directly from Definition 6.

According to Azuma-Hoeffding inequality (Hoeffding, 1994; Azuma, 1967):

$$\Pr(A_m) = \Pr\left(\sum_{k=0}^{\lfloor\frac{T-1-m}{\tau}\rfloor} X_{k,m} \geq 0\right)$$
$$= \Pr\left(\sum_{k=0}^{\lfloor\frac{T-1-m}{\tau}\rfloor} Y_{k,m} \geq -\sum_{k=0}^{\lfloor\frac{T-1-m}{\tau}\rfloor} \mathbb{E}[X_{k,m}|\mathcal{F}_{(k-1)\tau+m}]\right)$$
$$\leq \Pr\left(\sum_{k=0}^{\lfloor\frac{T-1-m}{\tau}\rfloor} Y_{k,m} \geq \left\lfloor\frac{T-1-m+\tau}{\tau}\right\rfloor\epsilon\sigma\right)$$
$$\leq \exp\left(-\frac{\epsilon^2\sigma^2\lfloor\frac{T}{\tau}\rfloor}{32}\right)$$

Similarly, from Azuma-Hoeffding inequality,

$$\Pr(A'_m) = \Pr\left(\sum_{k=0}^{\lfloor\frac{T-1-m}{\tau}\rfloor} X'_{k,m} = 0\right)$$
$$= \Pr\left(\sum_{k=0}^{\lfloor\frac{T-1-m}{\tau}\rfloor} Y'_{k,m} = -\sum_{k=0}^{\lfloor\frac{T-1-m}{\tau}\rfloor} \mathbb{E}[X'_{k,m}|\mathcal{F}_{(k-1)\tau+m}]\right)$$
$$\leq \Pr\left(\sum_{k=0}^{\lfloor\frac{T-1-m}{\tau}\rfloor} Y'_{k,m} \leq -\left\lfloor\frac{T-1-m+\tau}{\tau}\right\rfloor\sigma\right)$$

$$\leq \exp\left(-\frac{\sigma^2 \lfloor \frac{T}{\tau} \rfloor}{8}\right) \tag{36}$$

Thus

$$\Pr\left(\widehat{P_i^\theta}(s'|s,a_i) - \overline{P_i^\theta}(s'|s,a_i) \geq \epsilon\right) \leq \sum_{m=0}^{\tau-1} \Pr(A_m) + \Pr(A'_m)$$

$$\leq \tau \exp\left(-\frac{\epsilon^2 \sigma^2 \lfloor \frac{T}{\tau} \rfloor}{32}\right) + \tau \exp\left(-\frac{\sigma^2 \lfloor \frac{T}{\tau} \rfloor}{8}\right) \leq 2\tau \exp\left(-\frac{\epsilon^2 \sigma^2 \lfloor \frac{T}{\tau} \rfloor}{32}\right)$$

Similarly

$$\Pr\left(\widehat{P_i^\theta}(s'|s,a_i) - \overline{P_i^\theta}(s'|s,a_i) \leq -\epsilon\right) \leq 2\tau \exp\left(-\frac{\epsilon^2 \sigma^2 \lfloor \frac{T}{\tau} \rfloor}{32}\right)$$

$$\implies \Pr\left(\left|\widehat{P_i^\theta}(s'|s,a_i) - \overline{P_i^\theta}(s'|s,a_i)\right| \geq \epsilon\right) \leq 4\tau \exp\left(-\frac{\epsilon^2 \sigma^2 \lfloor \frac{T}{\tau} \rfloor}{32}\right)$$

which completes the proof. $\qquad\square$

**Lemma 14.** *Assume that the stochastic game with policy $\theta$ satisfies $(\sigma,\tau)$-sufficient exploration condition (Definition 6), then fix $s', s, a_i$, for $\epsilon \leq 1$,*

$$\Pr\left(\left|\widehat{r_i^\theta}(s,a_i) - \overline{r_i^\theta}(s,a_i)\right| \geq \epsilon\right) \leq 4\tau \exp\left(-\frac{\epsilon^2 \sigma^2 \lfloor \frac{T}{\tau} \rfloor}{32}\right)$$

*Proof.* The proof is similar to Lemma 13.

$$\left\{\widehat{r_i^\theta}(s,a_i) - \overline{r_i^\theta}(s,a_i) \geq \epsilon\right\}$$

$$\subseteq \left\{\sum_{t=0}^{T} \mathbf{1}\{s_t = s, a_{i,t} = a_i\} r_i(s_t, a_t) - (\overline{r_i^\theta}(s,a_i) + \epsilon)\mathbf{1}\{s_t = s, a_{i,t} = a_i\} \geq 0\right\}$$

$$\cup \left\{\sum_{t=0}^{T-1} \mathbf{1}\{s_t = s, a_{i,t} = a_i\} = 0\right\}$$

$$\subseteq \bigcup_{m=0}^{\tau-1} \left\{\sum_{k=0}^{\lfloor \frac{T-m}{\tau} \rfloor} \mathbf{1}\{s_{k\tau+m} = s, a_{i,k\tau+m} = a_i\} r_i(s_{k\tau+m}, a_{k\tau+m}) - (\overline{r_i^\theta}(s,a_i) + \epsilon)\mathbf{1}\{s_{k\tau+m} = s, a_{i,k\tau+m} = a_i\} \geq 0\right\}$$

$$\bigcup_{m=0}^{\tau-1} \left\{\sum_{k=0}^{\lfloor \frac{T-1-m}{\tau} \rfloor} \mathbf{1}\{s_{k\tau+m} = s, a_{i,k\tau+m} = a_i\} = 0\right\}$$

Let:

$$A_m := \left\{\sum_{k=0}^{\lfloor \frac{T-m}{\tau} \rfloor} \mathbf{1}\{s_{k\tau+m} = s, a_{i,k\tau+m} = a_i\} r_i(s_{k\tau+m}, a_{k\tau+m}) - (\overline{r_i^\theta}(s,a_i) + \epsilon)\mathbf{1}\{s_{k\tau+m} = s, a_{i,k\tau+m} = a_i\} \geq 0\right\}$$

$$A'_m := \left\{\sum_{k=0}^{\lfloor \frac{T-1-m}{\tau} \rfloor} \mathbf{1}\{s_{k\tau+m} = s, a_{i,k\tau+m} = a_i\} = 0\right\}$$

$$X_{k,m} := \mathbf{1}\{s_{k\tau+m} = s, a_{i,k\tau+m} = a_i\} r_i(s_{k\tau+m}, a_{k\tau+m}) - (\overline{r_i^\theta}(s,a_i) + \epsilon)\mathbf{1}\{s_{k\tau+m} = s, a_{i,k\tau+m} = a_i\}$$

$$Y_{k,m} := X_{k,m} - \mathbb{E}[X_{k,m}|\mathcal{F}_{(k-1)\tau+m}]$$

Then $\{Y_{k,m}\}_{k=0}^{\lfloor \frac{T-1-m}{\tau} \rfloor}$ is a martingale difference sequence. Because $\epsilon \leq 1$, it is easy to verify that $|X_{k,m}| \leq 2$. We have that:

$$|Y_{k,m}| \leq |X_{k,m}| + \mathbb{E}[|X_{k,m}||\mathcal{F}_{(k-1)\tau+m}] \leq 4.$$

Further,

$$\mathbb{E}[X_{k,m}|\mathcal{F}_{(k-1)\tau+m}]$$

$$= \mathbb{E}[\mathbf{1}\{s_{k\tau+m} = s, a_{i,k\tau+m} = a_i\}r_i(s_{k\tau+m}, a_{k\tau+m}) - (\overline{r_i^\theta}(s,a_i) + \epsilon)\mathbf{1}\{s_{k\tau+m} = s, a_{i,k\tau+m} = a_i\}|\mathcal{F}_{(k-1)\tau+m}]$$

$$= -\epsilon\mathbb{E}[\mathbf{1}\{s_{k\tau+m} = s, a_{i,k\tau+m} = a_i\}|\mathcal{F}_{(k-1)\tau+m}] \leq -\epsilon\sigma$$

the second line to the third line of the equation is derived by the fact that:

$$\mathbb{E}[\mathbf{1}\{s_t = s, a_{i,t+m} = a_i\}r_i(s_t, a_t)|\mathcal{F}_{t-1}] = P(s|s_{t-1}, a_{t-1})\sum_{a_{-i}} \pi_\theta(a_i, a_{-i}|s)r_i(s, a_i, a_{-i})$$

$$= P(s|s_{t-1}, a_{t-1})\pi_{\theta_i}(a_i|s)\overline{r_i^\theta}(s, a_i)$$

$$= \mathbb{E}[\overline{r_i^\theta}(s, a_i)\mathbf{1}\{s_t = s, a_{i,t} = a_t\}|\mathcal{F}_{t-1}]$$

and the inequality in the third line is derived directly from Definition 6.

According to Azuma-Hoeffding inequality:

$$\Pr(A_m) = \Pr\left(\sum_{k=0}^{\lfloor\frac{T-m}{\tau}\rfloor} X_{k,m} \geq 0\right)$$

$$= \Pr\left(\sum_{k=0}^{\lfloor\frac{T-m}{\tau}\rfloor} Y_{k,m} \geq -\sum_{k=0}^{\lfloor\frac{T-m}{\tau}\rfloor} \mathbb{E}[X_{k,m}|\mathcal{F}_{(k-1)\tau+m}]\right)$$

$$\leq \Pr\left(\sum_{k=0}^{\lfloor\frac{T-m}{\tau}\rfloor} Y_{k,m} \geq \left\lfloor\frac{T-m+\tau}{\tau}\right\rfloor\epsilon\sigma\right)$$

$$\leq \exp\left(-\frac{\epsilon^2\sigma^2\lfloor\frac{T}{\tau}\rfloor}{32}\right)$$

Same as Equation (36), we have that:

$$\Pr(A'_m) \leq \exp\left(-\frac{\sigma^2\lfloor\frac{T}{\tau}\rfloor}{8}\right)$$

Thus

$$\Pr\left(\widehat{r_i^\theta}(s, a_i) - \overline{r_i^\theta}(s, a_i) \geq \epsilon\right) \leq \sum_{m=0}^{\tau-1} \Pr(A_m) + \Pr(A'_m) \leq 2\tau\exp\left(-\frac{\epsilon^2\sigma^2\lfloor\frac{T}{\tau}\rfloor}{32}\right)$$

Similarly

$$\Pr\left(\widehat{r_i^\theta}(s, a_i) - \overline{r_i^\theta}(s, a_i) \leq -\epsilon\right) \leq 2\tau\exp\left(-\frac{\epsilon^2\sigma^2\lfloor\frac{T}{\tau}\rfloor}{32}\right)$$

$$\implies \Pr\left(\left|\widehat{r_i^\theta}(s, a_i) - \overline{r_i^\theta}(s, a_i)\right| \geq \epsilon\right) \leq 4\tau\exp\left(-\frac{\epsilon^2\sigma^2\lfloor\frac{T}{\tau}\rfloor}{32}\right)$$

which completes the proof. $\square$

Lemma 13 and 14 lead to the following corollary:

**Corollary 1.**

$$\Pr(\|\widehat{M_i^\theta} - \overline{M_i^\theta}\|_\infty \geq \epsilon) \leq 4\tau|\mathcal{S}|^2|\mathcal{A}_i|\exp\left(-\frac{\epsilon^2\sigma^2\lfloor\frac{T}{\tau}\rfloor}{32|\mathcal{S}|^2}\right) \tag{37}$$

$$\Pr(\|\widehat{r_i^\theta} - \overline{r_i^\theta}\|_\infty \geq \epsilon) \leq 4\tau|\mathcal{S}||\mathcal{A}_i|\exp\left(-\frac{\epsilon^2\sigma^2\lfloor\frac{T}{\tau}\rfloor}{32}\right) \tag{38}$$

*Proof.* We first prove Equation (37)

$$\|\widehat{M_i^\theta} - \overline{M_i^\theta}\|_\infty = \max_{(s,a_i)} \sum_{(s',a_i')} \pi_{\theta_i}(a_i'|s') \left| \widehat{P_i^\theta}(s'|s,a_i) - \overline{P_i^\theta}(s'|s,a_i) \right|$$

$$= \max_{(s,a_i)} \sum_{s'} \left| \widehat{P_i^\theta}(s'|s,a_i) - \overline{P_i^\theta}(s'|s,a_i) \right|$$

Thus,

$$\left\{ \|\widehat{M_i^\theta} - \overline{M_i^\theta}\|_\infty \geq \epsilon \right\} = \bigcup_{(s,a_i)} \left\{ \sum_{s'} \left| \widehat{P_i^\theta}(s'|s,a_i) - \overline{P_i^\theta}(s'|s,a_i) \right| \geq \epsilon \right\}$$

$$\subseteq \bigcup_{(s,a_i)} \bigcup_{s'} \left\{ \left| \widehat{P_i^\theta}(s'|s,a_i) - \overline{P_i^\theta}(s'|s,a_i) \right| \geq \frac{\epsilon}{|\mathcal{S}|} \right\}$$

Then according to Lemma 13,

$$\Pr\left( \|\widehat{M_i^\theta} - \overline{M_i^\theta}\|_\infty \geq \epsilon \right) \leq \sum_{(s',s,a_i)} \Pr\left( \left| \widehat{P_i^\theta}(s'|s,a_i) - \overline{P_i^\theta}(s'|s,a_i) \right| \geq \frac{\epsilon}{|\mathcal{S}|} \right)$$

$$\leq 4\tau|\mathcal{S}|^2|\mathcal{A}_i| \exp\left( -\frac{\epsilon^2\sigma^2\lfloor\frac{T}{\tau}\rfloor}{32|\mathcal{S}|^2} \right).$$

Now we prove Equation (38). Since

$$\left\{ \|\widehat{r_i^\theta} - \overline{r_i^\theta}\|_\infty \geq \epsilon \right\} = \bigcup_{(s,a_i)} \left\{ \left| \widehat{r_i^\theta}(s,a_i) - \overline{r_i^\theta}(s,a_i) \right| \geq \epsilon \right\},$$

according to Lemma 14,

$$\Pr\left( \|\widehat{r_i^\theta} - \overline{r_i^\theta}\|_\infty \geq \epsilon \right) \leq \sum_{(s,a_i)} \Pr\left( \left| \widehat{r_i^\theta}(s,a_i) - \overline{r_i^\theta}(s,a_i) \right| \geq \epsilon \right)$$

$$\leq 4\tau|\mathcal{S}||\mathcal{A}_i| \exp\left( -\frac{\epsilon^2\sigma^2\lfloor\frac{T}{\tau}\rfloor}{32} \right),$$

which completes the proof of the corollary. □

We are now ready to prove Theorem 7.

*Proof.* (of Theorem 7) From the definition of $\widehat{Q_i^\theta}, \overline{Q_i^\theta}$,

$$\overline{Q_i^\theta} = (I - \gamma\overline{M_i^\theta})^{-1}\overline{r_i^\theta},$$

$$\widehat{Q_i^\theta} = (I - \gamma\widehat{M_i^\theta})^{-1}\widehat{r_i^\theta},$$

we have that

$$\|\widehat{Q_i^\theta} - \overline{Q_i^\theta}\|_\infty = \left\| (I - \gamma\widehat{M_i^\theta})^{-1}\widehat{r_i^\theta} - (I - \gamma\overline{M_i^\theta})^{-1}\overline{r_i^\theta} \right\|_\infty$$

$$= \left\| (I - \gamma\overline{M_i^\theta})^{-1}(\widehat{r_i^\theta} - \overline{r_i^\theta}) + \left( (I - \gamma\widehat{M_i^\theta})^{-1} - (I - \gamma\overline{M_i^\theta})^{-1} \right) \widehat{r_i^\theta} \right\|_\infty$$

$$\leq \left\| (I - \gamma\overline{M_i^\theta})^{-1}(\widehat{r_i^\theta} - \overline{r_i^\theta}) \right\|_\infty + \left\| \gamma(I - \gamma\overline{M_i^\theta})^{-1}(\widehat{M_i^\theta} - \overline{M_i^\theta})(I - \gamma\widehat{M_i^\theta})^{-1}\widehat{r_i^\theta} \right\|_\infty.$$

Because both $\overline{M_i^\theta}$ and $\widehat{M_i^\theta}$ are transition probability matrices, thus:

$$\|\overline{M_i^\theta}x\|_\infty \leq \|x\|_\infty$$

$$\|\widehat{M_i^\theta}x\|_\infty \leq \|x\|_\infty$$

$$\|(I - \gamma\overline{M_i^\theta})^{-1}x\|_\infty \leq \frac{1}{1-\gamma}\|x\|_\infty$$

$$\|(I - \gamma \widehat{M_i^\theta})^{-1} x\|_\infty \le \frac{1}{1-\gamma} \|x\|_\infty$$

Thus,

$$\|\widehat{Q_i^\theta} - \overline{Q_i^\theta}\|_\infty \le \left\|(I - \gamma \overline{M_i^\theta})^{-1} (\widehat{r_i^\theta} - \overline{r_i^\theta})\right\|_\infty + \left\|\gamma (I - \gamma \overline{M_i^\theta})^{-1} (\widehat{M_i^\theta} - \overline{M_i^\theta})(I - \gamma \widehat{M_i^\theta})^{-1} \widehat{r_i^\theta}\right\|_\infty$$

$$\le \frac{1}{1-\gamma} \|\widehat{r_i^\theta} - \overline{r_i^\theta}\|_\infty + \frac{\gamma}{(1-\gamma)^2} \|\widehat{M_i^\theta} - \overline{M_i^\theta}\|_\infty \|\widehat{r_i^\theta}\|_\infty$$

$$\le \frac{1}{1-\gamma} \|\widehat{r_i^\theta} - \overline{r_i^\theta}\|_\infty + \frac{\gamma}{(1-\gamma)^2} \|\widehat{M_i^\theta} - \overline{M_i^\theta}\|_\infty$$

Thus if

$$\|\widehat{r_i^\theta} - \overline{r_i^\theta}\|_\infty \le (1-\gamma)^2 \epsilon, \quad \|\widehat{M_i^\theta} - \overline{M_i^\theta}\|_\infty \le (1-\gamma)^2 \epsilon,$$

we have that:

$$\|\widehat{Q_i^\theta} - \overline{Q_i^\theta}\|_\infty \le \epsilon,$$

Thus from Corollary 1,

$$\Pr\left(\|\widehat{Q_i^\theta} - \overline{Q_i^\theta}\|_\infty \ge \epsilon\right) \le \Pr\left(\|\widehat{r_i^\theta} - \overline{r_i^\theta}\|_\infty \ge (1-\gamma)^2 \epsilon\right) + \Pr\left(\|\widehat{M_i^\theta} - \overline{M_i^\theta}\|_\infty \ge (1-\gamma)^2 \epsilon\right)$$

$$\le 4\tau |\mathcal{S}||\mathcal{A}_i| \exp\left(-\frac{(1-\gamma)^4 \epsilon^2 \sigma^2 \lfloor \frac{T}{\tau} \rfloor}{32}\right) + 4\tau |\mathcal{S}|^2 |\mathcal{A}_i| \exp\left(-\frac{(1-\gamma)^4 \epsilon^2 \sigma^2 \lfloor \frac{T}{\tau} \rfloor}{32 |\mathcal{S}|^2}\right)$$

$$\le 8\tau |\mathcal{S}|^2 |\mathcal{A}_i| \exp\left(-\frac{(1-\gamma)^4 \epsilon^2 \sigma^2 \lfloor \frac{T}{\tau} \rfloor}{32 |\mathcal{S}|^2}\right),$$

which completes the proof. $\qquad\square$

## J.2 Bounding the estimation error of $d_\theta$

We first state our main result:

**Theorem 8.** *(Estimation error of $d_\theta$) Under Assumption 3,*

$$\Pr\left(\|\widehat{d_\theta} - d_\theta\|_1 \ge \epsilon\right) \le 4\tau |\mathcal{S}|^2 \exp\left(-\frac{(1-\gamma)^2 \epsilon^2 \sigma_S^2 \lfloor \frac{T}{\tau} \rfloor}{32 \gamma^2 |\mathcal{S}|^2}\right),$$

*i.e., when*

$$T \ge \frac{32 \tau |\mathcal{S}|^2}{(1-\gamma)^2 \epsilon^2 \sigma_S^2} \log\left(\frac{4\tau |\mathcal{S}|^2|}{\delta}\right) + 1,$$

*with probability at least $1 - \delta$, $\|\widehat{d_\theta} - d_\theta\|_1 \le \epsilon$.*

Similar to the previous section, the proof of the theorem begins by bounding the estimation error $|\widehat{P_\mathcal{S}^\theta}(s'|s) - \overline{P_\mathcal{S}^\theta}(s'|s)|$.

**Lemma 15.** *Under Assumption 3, fix $s', s, a_i$, for $\epsilon \le 1$,*

$$\Pr\left(\left|\widehat{P_\mathcal{S}^\theta}(s'|s) - \overline{P_\mathcal{S}^\theta}(s'|s)\right| \ge \epsilon\right) \le 4\tau \exp\left(-\frac{\epsilon^2 \sigma_S^2 \lfloor \frac{T}{\tau} \rfloor}{32}\right)$$

*Proof.* According to the definition of $\widehat{P_\mathcal{S}^\theta}$, we have that

$$\left\{\widehat{P_\mathcal{S}^\theta}(s'|s) - \overline{P_\mathcal{S}^\theta}(s'|s) \ge \epsilon\right\}$$

$$\subseteq \left\{\sum_{t=0}^{T-1} \left(\mathbf{1}\{s_{t+1} = s', s_t = s\} - (\overline{P_\mathcal{S}^\theta}(s'|s) + \epsilon)\mathbf{1}\{s_t = s\}\right) \ge 0\right\} \cup \left\{\sum_{t=0}^{T-1} \mathbf{1}\{s_t = s\} = 0\right\}$$

$$
\subseteq \bigcup_{m=0}^{\tau-1} \left\{ \sum_{k=0}^{\lfloor \frac{T-1-m}{\tau} \rfloor} \left( \mathbf{1}\{s_{k\tau+m+1} = s', s_{k\tau+m} = s\} - (\overline{P_{\mathcal{S}}^\theta}(s'|s) + \epsilon)\mathbf{1}\{s_{k\tau+m} = s\} \right) \geq 0 \right\}
$$

$$
\bigcup_{m=0}^{\tau-1} \left\{ \sum_{k=0}^{\lfloor \frac{T-1-m}{\tau} \rfloor} \mathbf{1}\{s_{k\tau+m} = s\} = 0 \right\}
$$

Let:

$$
A_m := \left\{ \sum_{k=0}^{\lfloor \frac{T-1-m}{\tau} \rfloor} \left( \mathbf{1}\{s_{k\tau+m+1} = s', s_{k\tau+m} = s\} - (\overline{P_{\mathcal{S}}^\theta}(s'|s) + \epsilon)\mathbf{1}\{s_{k\tau+m} = s\} \right) \geq 0 \right\}
$$

$$
A'_m := \left\{ \sum_{k=0}^{\lfloor \frac{T-1-m}{\tau} \rfloor} \mathbf{1}\{s_{k\tau+m} = s\} = 0 \right\}
$$

$$
X_{k,m} := \mathbf{1}\{s_{k\tau+m+1} = s', s_{k\tau+m} = s\} - (\overline{P_{\mathcal{S}}^\theta}(s'|s) + \epsilon)\mathbf{1}\{s_{k\tau+m} = s\}
$$
$$
X'_{k,m} := \mathbf{1}\{s_{k\tau+m} = s\}
$$
$$
Y_{k,m} := X_{k,m} - \mathbb{E}[X_{k,m}|\mathcal{F}_{(k-1)\tau+m}]
$$
$$
Y'_{k,m} := X'_{k,m} - \mathbb{E}[X'_{k,m}|\mathcal{F}_{(k-1)\tau+m}]
$$

Then $\{Y_{k,m}\}_{k=0}^{\lfloor \frac{T-1-m}{\tau} \rfloor}$ is a martingale difference sequence. Because $\epsilon \leq 1$, it is easy to verify that $|X_{k,m}| \leq 2, |X'_{k,m}| \leq 1$. We have that:

$$
|Y_{k,m}| \leq |X_{k,m}| + \mathbb{E}[|X_{k,m}||\mathcal{F}_{(k-1)\tau+m}] \leq 4, \quad |Y'_{k,m}| \leq |X'_{k,m}| + \mathbb{E}[|X'_{k,m}||\mathcal{F}_{(k-1)\tau+m}] \leq 2.
$$

Further,

$$
\mathbb{E}[X'_{k,m}|\mathcal{F}_{(k-1)\tau+m}] = \mathbb{E}[\mathbf{1}\{s_{k\tau+m} = s\}|\mathcal{F}_{(k-1)\tau+m}] \geq \sigma_S,
$$

and that

$$
\mathbb{E}[X_{k,m}|\mathcal{F}_{(k-1)\tau+m}]
$$
$$
= \mathbb{E}[\mathbf{1}\{s_{k\tau+m+1} = s', s_{k\tau+m} = s\} - (\overline{P_{\mathcal{S}}^\theta}(s'|s) + \epsilon)\mathbf{1}\{s_{k\tau+m} = s\}|\mathcal{F}_{(k-1)\tau+m}]
$$
$$
= -\epsilon \mathbb{E}[\mathbf{1}\{s_{k\tau+m} = s\}|\mathcal{F}_{(k-1)\tau+m}] \leq -\epsilon\sigma_S
$$

the second line to the third line of the equation is derived by the fact that:

$$
\mathbb{E}[\mathbf{1}\{s_{t+1} = s', s_t = s\}|\mathcal{F}_{t-1}] = P(s|s_{t-1}, a_{t-1}) \sum_a \pi_\theta(a|s) P(s'|s, a)
$$
$$
= P(s|s_{t-1}, a_{t-1})\overline{P_{\mathcal{S}}^\theta}(s'|s)
$$
$$
= \mathbb{E}[\overline{P_{\mathcal{S}}^\theta}(s'|s)\mathbf{1}\{s_t = s\}|\mathcal{F}_{t-1}]
$$

and the inequality in the third line is derived directly from Assumption 3.

According to Azuma-Hoeffding inequality:

$$
\Pr(A_m) = \Pr\left( \sum_{k=0}^{\lfloor \frac{T-1-m}{\tau} \rfloor} X_{k,m} \geq 0 \right)
$$
$$
= \Pr\left( \sum_{k=0}^{\lfloor \frac{T-1-m}{\tau} \rfloor} Y_{k,m} \geq - \sum_{k=0}^{\lfloor \frac{T-1-m}{\tau} \rfloor} \mathbb{E}[X_{k,m}|\mathcal{F}_{(k-1)\tau+m}] \right)
$$
$$
\leq \Pr\left( \sum_{k=0}^{\lfloor \frac{T-1-m}{\tau} \rfloor} Y_{k,m} \geq \left\lfloor \frac{T-1-m+\tau}{\tau} \right\rfloor \epsilon\sigma_S \right)
$$
$$
\leq \exp\left( -\frac{\epsilon^2 \sigma_S^2 \lfloor \frac{T}{\tau} \rfloor}{32} \right)
$$

Similarly, from Azuma-Hoeffding inequality,

$$
\Pr(A'_m) = \Pr\left( \sum_{k=0}^{\lfloor \frac{T-1-m}{\tau} \rfloor} X'_{k,m} = 0 \right)
$$

$$
= \Pr\left( \sum_{k=0}^{\lfloor \frac{T-1-m}{\tau} \rfloor} Y'_{k,m} = - \sum_{k=0}^{\lfloor \frac{T-1-m}{\tau} \rfloor} \mathbb{E}[X'_{k,m} | \mathcal{F}_{(k-1)\tau+m}] \right)
$$

$$
\leq \Pr\left( \sum_{k=0}^{\lfloor \frac{T-1-m}{\tau} \rfloor} Y'_{k,m} \leq - \left\lfloor \frac{T-1-m+\tau}{\tau} \right\rfloor \sigma_S \right)
$$

$$
\leq \exp\left( -\frac{\sigma_S^2 \lfloor \frac{T}{\tau} \rfloor}{8} \right)
$$

Thus

$$
\Pr\left( \widehat{P_S^\theta}(s'|s) - \overline{P_S^\theta}(s'|s) \geq \epsilon \right) \leq \sum_{m=0}^{\tau-1} \Pr\left( A_m \right) + \Pr\left( A'_m \right)
$$

$$
\leq \tau \exp\left( -\frac{\epsilon^2 \sigma_S^2 \lfloor \frac{T}{\tau} \rfloor}{32} \right) + \tau \exp\left( -\frac{\sigma_S^2 \lfloor \frac{T}{\tau} \rfloor}{8} \right) \leq 2\tau \exp\left( -\frac{\epsilon^2 \sigma_S^2 \lfloor \frac{T}{\tau} \rfloor}{32} \right)
$$

Similarly

$$
\Pr\left( \widehat{P_S^\theta}(s'|s) - \overline{P_S^\theta}(s'|s) \leq -\epsilon \right) \leq 2\tau \exp\left( -\frac{\epsilon^2 \sigma_S^2 \lfloor \frac{T}{\tau} \rfloor}{32} \right)
$$

$$
\implies \Pr\left( \left| \widehat{P_S^\theta}(s'|s) - \overline{P_S^\theta}(s'|s) \right| \geq \epsilon \right) \leq 4\tau \exp\left( -\frac{\epsilon^2 \sigma_S^2 \lfloor \frac{T}{\tau} \rfloor}{32} \right)
$$

which completes the proof. □

**Corollary 2.**

$$
\Pr\left( \left\| \widehat{P_S^\theta} - \overline{P_S^\theta} \right\|_\infty \geq \epsilon \right) \leq 4\tau |\mathcal{S}|^2 \exp\left( -\frac{\epsilon^2 \sigma_S^2 \lfloor \frac{T}{\tau} \rfloor}{32|\mathcal{S}|^2} \right) \tag{39}
$$

*Proof.*

$$
\left\| \widehat{P_S^\theta} - \overline{P_S^\theta} \right\|_\infty = \max_s \sum_{s'} \left| \widehat{P_S^\theta}(s'|s) - \overline{P_S^\theta}(s'|s) \right|
$$

Thus,

$$
\left\{ \left\| \widehat{P_S^\theta} - \overline{P_S^\theta} \right\|_\infty \geq \epsilon \right\} = \bigcup_s \left\{ \sum_{s'} \left| \widehat{P_S^\theta}(s'|s) - \overline{P_S^\theta}(s'|s) \right| \geq \epsilon \right\}
$$

$$
\subseteq \bigcup_s \bigcup_{s'} \left\{ \left| \widehat{P_S^\theta}(s'|s) - \overline{P_S^\theta}(s'|s) \right| \geq \frac{\epsilon}{|\mathcal{S}|} \right\}
$$

Then according to Lemma 15,

$$
\Pr\left( \| \left\| \widehat{P_S^\theta} - \overline{P_S^\theta} \right\|_\infty \geq \epsilon \right) \leq \sum_{(s',s)} \Pr\left( \left| \widehat{P_S^\theta}(s'|s) - \overline{P_S^\theta}(s'|s) \right| \geq \frac{\epsilon}{|\mathcal{S}|} \right)
$$

$$
\leq 4\tau |\mathcal{S}|^2 \exp\left( -\frac{\epsilon^2 \sigma_S^2 \lfloor \frac{T}{\tau} \rfloor}{32|\mathcal{S}|^2} \right).
$$

which completes the proof of the corollary. □

*Proof.* (Proof of Theorem 8)

$$\|\widehat{d_\theta} - d_\theta\|_1 = (1-\gamma)\left\|\left(\left(I - \gamma\widehat{P_S^\theta}^\top\right)^{-1} - \left(I - \gamma\overline{P_S^\theta}^\top\right)^{-1}\right)\rho\right\|_1$$

$$\leq (1-\gamma)\left\|\left(I - \gamma\widehat{P_S^\theta}^\top\right)^{-1} - \left(I - \gamma\overline{P_S^\theta}^\top\right)^{-1}\right\|_1 \|\rho\|_1$$

$$\leq (1-\gamma)\left\|\left(I - \gamma\widehat{P_S^\theta}\right)^{-1} - \left(I - \gamma\overline{P_S^\theta}\right)^{-1}\right\|_\infty \|\rho\|_1$$

$$= \gamma(1-\gamma)\left\|\left(I - \gamma\overline{P_S^\theta}\right)^{-1}\left(\widehat{P_S^\theta} - \overline{P_S^\theta}\right)\left(I - \gamma\widehat{P_S^\theta}\right)^{-1}\right\|_\infty$$

$$\leq \frac{\gamma}{1-\gamma}\left\|\widehat{P_S^\theta} - \overline{P_S^\theta}\right\|_\infty$$

Thus

$$\Pr\left(\|\widehat{d_\theta} - d_\theta\|_1 \geq \epsilon\right) \leq \Pr\left(\left\|\widehat{P_S^\theta} - \overline{P_S^\theta}\right\|_\infty \geq \frac{1-\gamma}{\gamma}\epsilon\right)$$

$$\leq 4\tau|\mathcal{S}|^2 \exp\left(-\frac{(1-\gamma)^2\epsilon^2\sigma_S^2\lfloor\frac{T}{\tau}\rfloor}{32\gamma^2|\mathcal{S}|^2}\right)$$

$\square$

### J.3 PROOF OF THEOREM 6

*Proof.* Since the stochastic game satisfies $(\tau, \sigma_S)$-sufficient exploration on states, then for any $\theta \in \mathcal{X}^\alpha$, we know that it satisfies $(\tau, \frac{\alpha\sigma_S}{\max_i|\mathcal{A}_i|})$-sufficient exploration. Substitute this into Theorem 7, we have that for

$$T_J \geq \frac{32\tau(1+\alpha)^2|\mathcal{S}|^3\sum_i|\mathcal{A}_i|\max_i|\mathcal{A}_i|^2}{(1-\gamma)^6\epsilon_g^2\alpha^2\sigma_S^2}\log\left(\frac{16\tau T_G|\mathcal{S}|^2\sum_i|\mathcal{A}_i|}{\delta}\right) + 1, \quad (40)$$

with probability at least $1 - \frac{\delta}{2T_G}$,

$$\|\overline{Q^{\theta^{(k)}}} - \widehat{Q^{\theta^{(k)}}}\|_\infty \leq \frac{(1-\gamma)\epsilon_g}{(1+\alpha)\sqrt{|\mathcal{S}|\sum_i|\mathcal{A}_i|}}.$$

Similarly, applying Theorem 8, we have that with probability at least $1 - \frac{\delta}{2T_G}$,

$$\|d_{\theta^{(k)}} - \widehat{d_{\theta^{(k)}}}\|_1 \leq \frac{(1-\gamma)^2\epsilon_g\alpha}{(1+\alpha)\sqrt{|\mathcal{S}|\sum_i|\mathcal{A}_i|}}.$$

Since:

$$\left|\left[\nabla\Phi(\theta) - \widehat{\nabla}\Phi(\theta)\right]_{(s,a_i)}\right| = \left|\frac{1}{1-\gamma}d_\theta(s)\overline{Q_i^\theta}(s,a_i) - \frac{1}{1-\gamma}\widehat{d_\theta}(s)\widehat{Q_i^\theta}(s,a_i)\right|$$

$$= \left|\frac{1}{1-\gamma}d_\theta(s)(\overline{Q_i^\theta}(s,a_i) - \widehat{Q_i^\theta}(s,a_i))\right| + \left|\frac{1}{1-\gamma}\widehat{Q_i^\theta}(s,a_i)(d_\theta(s) - \widehat{d_\theta}(s))\right|$$

$$\leq \frac{1}{1-\gamma}|\overline{Q_i^\theta}(s,a_i) - \widehat{Q_i^\theta}(s,a_i)| + \frac{1}{(1-\gamma)^2}|d_\theta(s) - \widehat{d_\theta}(s)|$$

$$\leq \frac{\epsilon_g}{(1+\alpha)\sqrt{|\mathcal{S}|\sum_j|\mathcal{A}_j|}} + \frac{\epsilon_g\alpha}{(1+\alpha)\sqrt{|\mathcal{S}|\sum_j|\mathcal{A}_j|}}$$

$$= \frac{\epsilon_g}{\sqrt{|\mathcal{S}|\sum_j|\mathcal{A}_j|}}$$

Thus, with probability $1 - \delta$

$$\|\nabla\Phi(\theta^{(k)} - \widehat{\nabla}\Phi(\theta^{(k)}))\|_2^2 = \sum_i\sum_s\sum_{a_i}\left|\left[\nabla\Phi(\theta) - \widehat{\nabla}\Phi(\theta)\right]_{(s,a_i)}\right|^2 \leq \epsilon_g^2, \ \forall 1 \leq k \leq T_G$$

$\square$

## K  PROOF OF THEOREM 5

**Notations:**  We define the following variables that will be useful in the analysis:

$$\widehat{G}^{\eta}(\theta) := \frac{1}{\eta}\left(Proj_{\mathcal{X}}(\theta + \eta\widehat{\nabla}\Phi(\theta)) - \theta\right)$$

$$\widehat{G}^{\eta,\alpha}(\theta) := \frac{1}{\eta}\left(Proj_{\mathcal{X}^{\alpha}}(\theta + \eta\widehat{\nabla}\Phi(\theta)) - \theta\right).$$

### K.1  OPTIMIZATION LEMMAS

**Lemma 16.** *(Sufficient ascent) Suppose* $\Phi(\theta)$ *is* $\beta$*-smooth. Let* $\theta^{+} = Proj_{\mathcal{X}^{\alpha}}(\theta + \eta\widehat{\nabla}\Phi(\theta))$. *Then for* $\eta \leq \frac{1}{2\beta}$,

$$\Phi(\theta^{+}) - \Phi(\theta) \geq \frac{\eta}{4}\|\widehat{G}^{\eta,\alpha}(\theta)\|^{2} - \frac{\eta}{2}\left\|\nabla\Phi(\theta) - \widehat{\nabla}\Phi(\theta)\right\|^{2}$$

*Proof.*  From the smoothness property we have that:

$$\Phi(\theta^{+}) - \Phi(\theta) \geq \nabla_{\theta}\Phi(\theta)^{\top}(\theta^{+} - \theta) - \frac{\beta}{2}\|\theta^{+} - \theta\|^{2}$$

Since $\theta^{+} = Proj_{\mathcal{X}^{\alpha}}(\theta + \eta\widehat{\nabla}\Phi(\theta))$, we have that:

$$(\theta + \eta\widehat{\nabla}\Phi(\theta) - \theta^{+})^{\top}(\theta' - \theta^{+}) \leq 0, \quad \forall\, \theta' \in \mathcal{X}^{\alpha}$$

take $\theta' = \theta$, we get:

$$\widehat{\nabla}\Phi(\theta)^{\top}(\theta^{+} - \theta) \geq \frac{1}{\eta}\|\theta^{+} - \theta\|^{2}.$$

Thus:

$$\nabla\Phi(\theta)^{\top}(\theta^{+} - \theta) = \left(\nabla\Phi(\theta) - \widehat{\nabla}\Phi(\theta)\right)^{\top}(\theta^{+} - \theta) + \widehat{\nabla}\Phi(\theta)^{\top}(\theta^{+} - \theta)$$

$$\geq -\frac{\eta}{2}\left\|\nabla\Phi(\theta) - \widehat{\nabla}\Phi(\theta)\right\|^{2} - \frac{1}{2\eta}\left\|\theta^{+} - \theta\right\|^{2} + \widehat{\nabla}\Phi(\theta)^{\top}(\theta^{+} - \theta)$$

$$\geq -\frac{\eta}{2}\left\|\nabla\Phi(\theta) - \widehat{\nabla}\Phi(\theta)\right\|^{2} - \frac{1}{2\eta}\left\|\theta^{+} - \theta\right\|^{2} + \frac{1}{\eta}\|\theta^{+} - \theta\|^{2}$$

$$= \frac{1}{2\eta}\|\theta^{+} - \theta\|^{2} - \frac{\eta}{2}\left\|\nabla\Phi(\theta) - \widehat{\nabla}\Phi(\theta)\right\|^{2}$$

Thus from equation (29):

$$\Phi(\theta^{+}) - \Phi(\theta) \geq \left(\frac{1}{2\eta} - \frac{\beta}{2}\right)\|\theta^{+} - \theta\|^{2} - \frac{\eta}{2}\left\|\nabla\Phi(\theta) - \widehat{\nabla}\Phi(\theta)\right\|^{2}$$

$$\geq \frac{1}{4\eta}\|\theta^{+} - \theta\|^{2} - \frac{\eta}{2}\left\|\nabla\Phi(\theta) - \widehat{\nabla}\Phi(\theta)\right\|^{2}$$

$$= \frac{\eta}{4}\|\widehat{G}^{\eta,\alpha}(\theta)\|^{2} - \frac{\eta}{2}\left\|\nabla\Phi(\theta) - \widehat{\nabla}\Phi(\theta)\right\|^{2}$$

which completes the proof.  □

Lemma 16 immediately results in the following corollary:

**Corollary 3.** *(of Lemma 16) In Algorithm 1, suppose* $\|\widehat{\nabla}\Phi(\theta^{(k)}) - \nabla\Phi(\theta^{(k)})\|_{\infty} \leq \epsilon_{g}$ *holds for every* $0 \leq k \leq T_{G} - 1$, *then running algorithm 1 will guarantee that:*

$$\frac{1}{T_{G}}\sum_{k=0}^{T_{G}-1}\|\widehat{G}^{\eta,\alpha}(\theta^{(k)})\|^{2} \leq \frac{4(\Phi_{\max} - \Phi_{\min})}{\eta T_{G}} + 2\epsilon_{g}^{2}$$

*Proof.* From Lemma 16 we have that:

$$\Phi(\theta^{(k+1)}) - \Phi(\theta^{(k)}) \geq \frac{\eta}{4}\|\widehat{G}^{\eta,\alpha}(\theta^{(k)})\|^2 - \frac{\eta}{2}\left\|\nabla\Phi(\theta^{(k)}) - \widehat{\nabla}\Phi(\theta^{(k)})\right\|^2$$

$$\geq \frac{\eta}{4}\|\widehat{G}^{\eta,\alpha}(\theta^{(k)})\|^2 - \frac{\eta}{2}\epsilon_g^2.$$

Thus

$$\frac{1}{T_G}\sum_{k=0}^{T_G-1}\|\widehat{G}^{\eta,\alpha}(\theta^{(k)})\|^2 \leq \frac{4(\Phi(\theta^{(0)}) - \Phi(\theta^{(T_G)}))}{\eta T_G} + 2\epsilon_g^2$$

$$\leq \frac{4(\Phi_{\max} - \Phi_{\min})}{\eta T_G} + 2\epsilon_g^2$$

$\square$

**Lemma 17.** *(First-order stationarity and $\|\widehat{G}^{\eta,\alpha}(\theta)\|$) Suppose $\Phi(\theta)$ is $\beta$-smooth. Let $\theta^+ = Proj_{\mathcal{X}^\alpha}(\theta + \eta\widehat{\nabla}\Phi(\theta))$. Then:*

$$\nabla_\theta\Phi(\theta^+)^\top(\theta'-\theta^+) \leq \left[(1+\eta\beta)\|\widehat{G}^{\eta,\alpha}(\theta)\| + \|\widehat{\nabla}\Phi(\theta) - \nabla\Phi(\theta)\|\right]\|\theta'-\theta^+\|, \quad \forall\theta' \in \mathcal{X}^\alpha. \quad (41)$$

*Further:*

$$\max_{\overline{\theta}_i \in \mathcal{X}_i} \nabla_{\theta_i}\Phi(\theta^+)^\top(\overline{\theta}_i - \theta_i^+) \leq 2\sqrt{|\mathcal{S}|}\left[(1+\eta\beta)\|\widehat{G}^{\eta,\alpha}(\theta)\| + \|\widehat{\nabla}\Phi(\theta) - \nabla\Phi(\theta)\|\right] + \frac{2\alpha}{1-\gamma} \quad (42)$$

*Proof.* Since $\theta^+ = Proj_{\mathcal{X}^\alpha}(\theta + \eta\widehat{\nabla}\Phi(\theta))$, we have:

$$(\theta + \eta\widehat{\nabla}\Phi(\theta) - \theta^+)^\top(\theta' - \theta^+) \leq 0 \ \forall\,\theta' \in \mathcal{X}^\alpha$$

$$\implies \eta\widehat{\nabla}\Phi(\theta)^\top(\theta' - \theta^+) \leq (\theta - \theta^+)^\top(\theta' - \theta^+)$$

$$\implies \eta\nabla\Phi(\theta)^\top(\theta' - \theta^+) \leq (\theta - \theta^+)^\top(\theta' - \theta^+) + \eta(\nabla\Phi(\theta) - \widehat{\nabla}\Phi(\theta))^\top(\theta' - \theta^+)$$

$$\implies \eta\nabla\Phi(\theta^+)^\top(\theta' - \theta^+) \leq (\theta - \theta^+)^\top(\theta' - \theta^+) + \eta(\nabla\Phi(\theta) - \widehat{\nabla}\Phi(\theta))^\top(\theta' - \theta^+)$$

$$+ \eta(\nabla\Phi(\theta^+) - \nabla\Phi(\theta))^\top(\theta' - \theta^+)$$

$$\implies \eta\nabla\Phi(\theta^+)^\top(\theta' - \theta^+) \leq (\|\theta - \theta^+\| + \eta\|\nabla\Phi(\theta) - \widehat{\nabla}\Phi(\theta)\| + \eta\|\nabla\Phi(\theta^+) - \nabla\Phi(\theta)\|)\|\theta' - \theta^+\|$$

$$\leq (\|\theta - \theta^+\| + \eta\|\nabla\Phi(\theta) - \widehat{\nabla}\Phi(\theta)\| + \eta\beta\|\theta^+ - \theta\|)\|\theta' - \theta^+\|$$

$$= \left[(1+\eta\beta)\|\theta - \theta^+\| + \eta\|\nabla\Phi(\theta) - \widehat{\nabla}\Phi(\theta)\|\right]\|\theta' - \theta^+\|$$

$$\implies \nabla\Phi(\theta^+)^\top(\theta' - \theta^+) \leq \left[(1+\eta\beta)\|\widehat{G}^{\eta,\alpha}(\theta)\| + \|\nabla\Phi(\theta) - \widehat{\nabla}\Phi(\theta)\|\right]\|\theta' - \theta^+\|,$$

which proves equation (41). We now prove equation (42). For any $\theta'_{i,s} \in \Delta(|\mathcal{A}_i|)$, we know that $(1-\alpha)\theta'_{i,s} + \alpha U_{|\mathcal{A}_i|} \in \Delta^\alpha(|\mathcal{A}_i|)$. Let $U_i := [\underbrace{U_{|\mathcal{A}_i|}, \ldots, U_{|\mathcal{A}_i|}}_{|\mathcal{S}|\text{times}}]$, then for any $\theta'_i \in \mathcal{X}_i$, $(1-\alpha)\theta'_i + \alpha U_i \in \mathcal{X}_i^\alpha$.

Thus:

$$\nabla_{\theta_i}\Phi(\theta^+)^\top(\theta'_i - \theta_i^+) \leq \nabla_{\theta_i}\Phi(\theta^+)^\top((1-\alpha)\theta'_i + \alpha U_i - \theta_i^+) + \nabla_{\theta_i}\Phi(\theta^+)^\top(\theta'_i - (1-\alpha)\theta'_i - \alpha U_i)$$

$$\leq \left[(1+\eta\beta)\|\widehat{G}^{\eta,\alpha}(\theta)\| + \|\widehat{\nabla}\Phi(\theta) - \nabla\Phi(\theta)\|\right]\|(1-\alpha)\theta'_i + \alpha U_i - \theta_i^+\|$$

$$+ \nabla_{\theta_i}\Phi(\theta^+)^\top(\theta'_i - (1-\alpha)\theta'_i - \alpha U_i)$$

$$\leq 2\sqrt{|\mathcal{S}|}\left[(1+\eta\beta)\|\widehat{G}^{\eta,\alpha}(\theta)\| + \|\widehat{\nabla}\Phi(\theta) - \nabla\Phi(\theta)\|\right] + \alpha\nabla_{\theta_i}\Phi(\theta^+)^\top(\theta'_i - U_i)$$

Since

$$\nabla_{\theta_i}\Phi(\theta^+)^\top(\theta'_i - U_i) = \sum_s d_\theta(s)\overline{Q_{i,s}^\theta}^\top(\theta'_{i,s} - U_{|\mathcal{A}_i|})$$

$$\leq \sum_s d_\theta(s) \|\overline{Q_{i,s}^\theta}\|_\infty \|\theta_{i,s}' - U_{|\mathcal{A}_i|}\|_1$$

$$\leq \sum_s d_\theta(s) \frac{2}{1-\gamma} \leq \frac{2}{1-\gamma},$$

we have that:

$$\nabla_{\theta_i} \Phi(\theta^+)^\top (\theta_i' - \theta_i) \leq 2\sqrt{|\mathcal{S}|} \left[ (1+\eta\beta)\|\widehat{G}^{\eta,\alpha}(\theta)\| + \|\widehat{\nabla}\Phi(\theta) - \nabla\Phi(\theta)\| \right] + \frac{2\alpha}{1-\gamma}$$

$\square$

### K.2 PROOF OF THEOREM 5

*Proof.* Recall that $\Phi$ is $\beta$-smooth with $\beta = \frac{2}{(1-\gamma)^3}\left(\sum_{i=1}^n |\mathcal{A}_i|\right)$. The step size $\eta$ in Theorem 5 satisfies $\eta \leq \frac{(1-\gamma)^3}{4\sum_{i=1}^n |\mathcal{A}_i|} = \frac{1}{2\beta}$.

Recall from gradient domination property:

$$\texttt{NE-gap}_i(\theta^{(k+1)}) = \max_{\theta_i' \in \mathcal{X}_i} J_i(\theta_i', \theta_{-i}^{(k+1)}) - J_i(\theta_i^{(k+1)}, \theta_{-i}^{(k+1)})$$

$$\leq M \max_{\theta_i' \in \mathcal{X}_i} (\theta_i' - \theta_i^{(k+1)})^\top \nabla_{\theta_i} \Phi(\theta^{(k+1)})$$

Suppose $\|\widehat{\nabla}\Phi(\theta^{(k)}) - \nabla\Phi(\theta^{(k)})\|_\infty \leq \epsilon_g$, $\forall 0 \leq k \leq T_G - 1$, recall from Lemma 17,

$$\texttt{NE-gap}(\theta^{(k+1)}) \leq \max_i \texttt{NE-gap}_i(\theta^{(k+1)}) \leq M \max_i \max_{\theta_i' \in \mathcal{X}_i} (\theta_i' - \theta_i^{(k+1)})^\top \nabla_{\theta_i} \Phi(\theta^{(k+1)})$$

$$\leq 2M\sqrt{|\mathcal{S}|} \left[ (1+\eta\beta)\|\widehat{G}^{\eta,\alpha}(\theta^{(k)})\| + \epsilon_g \right] + \frac{2\alpha M}{1-\gamma}$$

Thus,

$$\frac{1}{T_G} \sum_{k=0}^{T_G-1} \texttt{NE-gap}(\theta^{(k+1)})^2 \leq \frac{1}{T_G} \sum_{k=0}^{T_G-1} 3 \times \left[ 4M^2|\mathcal{S}|(1+\eta\beta)^2\|\widehat{G}^{\eta,\alpha}(\theta^{(k)})\|^2 + 4M^2|\mathcal{S}|\epsilon_g^2 + \frac{4\alpha^2 M^2}{(1-\gamma)^2} \right]$$

$$= 12M^2|\mathcal{S}|\epsilon_g^2 + \frac{12\alpha^2 M^2}{(1-\gamma)^2} + 12M^2|\mathcal{S}|(1+\eta\beta)^2 \left( \frac{1}{T_G} \sum_{k=0}^{T_G-1} \|\widehat{G}^{\eta,\alpha}(\theta^{(k)})\|^2 \right)$$

From Corollary 3, we have that

$$\frac{1}{T_G} \sum_{k=0}^{T_G-1} \texttt{NE-gap}(\theta^{(k+1)})^2 \leq 12M^2|\mathcal{S}|\epsilon_g^2 + \frac{12\alpha^2 M^2}{(1-\gamma)^2} + 12M^2|\mathcal{S}|(1+\eta\beta)^2 \left( \frac{4(\Phi_{\max} - \Phi_{\min})}{\eta T_G} + 2\epsilon_g^2 \right)$$

$$\leq 66M^2|\mathcal{S}|\epsilon_g^2 + \frac{12\alpha^2 M^2}{(1-\gamma)^2} + \frac{108M^2|\mathcal{S}|(\Phi_{\max} - \Phi_{\min})}{\eta T_G} \qquad (43)$$

Substitute

$$\alpha = \frac{(1-\gamma)\epsilon}{6M}, \quad \epsilon_g = \frac{\epsilon}{2\sqrt{33}M\sqrt{|\mathcal{S}|}} \text{ and } T_G \geq \frac{648M^2(\Phi_{\max} - \Phi_{\min})|\mathcal{S}|}{\eta\epsilon^2}$$

into the above inequality we get that:

$$\frac{1}{T_G} \sum_{k=0}^{T_G-1} \texttt{NE-gap}(\theta^{(k+1)})^2 \leq \frac{\epsilon^2}{2} + \frac{\epsilon^2}{3} + \frac{\epsilon^2}{6} = \epsilon^2$$

Substitute the value of $\alpha, \epsilon_g$ in Equation (43) into Theorem 6 will give us:

$$T_J \geq \frac{206976\tau n M^4 |\mathcal{S}|^3 \max_i |\mathcal{A}_i|^3}{(1-\gamma)^8 \epsilon^4 \sigma_S^2} \log\left( \frac{16\tau T_G |\mathcal{S}|^2 \sum_i |\mathcal{A}_i|}{\delta} \right) + 1$$

which completes the proof. $\square$

## L    SMOOTHNESS

**Lemma 18.** *(Smoothness for Direct Distributed Parameterization) Assume that $0 \leq r_i(s,a) \leq 1$, $\forall s, a$, $i = 1, 2, \ldots, n$, then:*

$$\|g(\theta') - g(\theta)\| \leq \frac{2}{(1-\gamma)^3} \left( \sum_{i=1}^{n} |\mathcal{A}_i| \right) \|\theta' - \theta\|, \tag{44}$$

*where $g(\theta) = \{\nabla_{\theta_i} J_i(\theta)\}$.*

The proof of Lemma 18 depends on the following lemma:
**Lemma 19.**

$$\|\nabla_{\theta_i} J_i(\theta') - \nabla_{\theta_i} J_i(\theta)\| \leq \frac{2}{(1-\gamma)^3} \sqrt{|\mathcal{A}_i|} \sum_{j=1}^{n} \sqrt{|\mathcal{A}_j|} \|\theta'_j - \theta_j\| \tag{45}$$

Lemma 18 is a simple corollary of Lemma 19.

*Proof.* (Proof of Lemma 18)

$$\|g(\theta') - g(\theta)\|^2 = \sum_{i=1}^{n} \|\nabla_{\theta_i} J_i(\theta') - \nabla_{\theta_i} J_i(\theta)\|^2$$

$$\leq \left( \frac{2}{(1-\gamma)^3} \right)^2 \sum_i |\mathcal{A}_i| \left( \sum_{j=1}^{n} \sqrt{|\mathcal{A}_j|} \|\theta'_j - \theta_j\| \right)^2$$

$$\leq \left( \frac{2}{(1-\gamma)^3} \right)^2 \sum_i |\mathcal{A}_i| \left( \sum_{j=1}^{n} |\mathcal{A}_j| \right) \left( \sum_{j=1}^{n} \|\theta'_j - \theta_j\|^2 \right)$$

$$= \left( \frac{2}{(1-\gamma)^3} \right)^2 \left( \sum_{i=1}^{n} |\mathcal{A}_i| \right)^2 \|\theta' - \theta\|^2,$$

which completes the proof. $\square$

Lemma 19 is equivalent to the following lemma:
**Lemma 20.**

$$\left| \frac{\partial J_i(\theta'_i + \alpha u_i, \theta'_{-i}) - \partial J_i(\theta_i + \alpha u_i, \theta_{-i})}{\partial \alpha} \bigg|_{\alpha=0} \right| \leq \frac{2}{(1-\gamma)^3} \sqrt{|\mathcal{A}_i|} \sum_{j=1}^{n} \sqrt{|\mathcal{A}_j|} \|\theta'_j - \theta_j\|, \quad \forall \|u\| = 1 \tag{46}$$

*Proof.* (Lemma 20) Define:

$$\pi_{i,\alpha}(a_i|s) := \pi_{\theta_i + \alpha u_i}(a_i|s) = \theta_{s,a_i} + \alpha u_{a_i,s}$$
$$\pi'_{i,\alpha}(a_i|s) := \pi'_{\theta_i + \alpha u_i}(a_i|s) = \theta'_{s,a_i} + \alpha u_{a_i,s}$$
$$\pi_\alpha(a|s) := \pi_{\theta_i + \alpha u_i}(a_i|s) \pi_{\theta_{-i}}(a_{-i}|s)$$
$$\pi'_\alpha(a|s) := \pi_{\theta'_i + \alpha u_i}(a_i|s) \pi'_{\theta_{-i}}(a_{-i}|s)$$
$$Q_i^\alpha(s,a) := Q_{(\theta_i + \alpha u_i, \theta_{-i})}(s,a)$$
$$d'_\alpha(s) := d_{(\theta'_i + \alpha u_i, \theta_{-i})}(s)$$

According to cost difference lemma,

$$\left| \frac{\partial J_i(\theta'_i + \alpha u_i, \theta'_{-i}) - \partial J_i(\theta_i + \alpha u_i, \theta_{-i})}{\partial \alpha} \bigg|_{\alpha=0} \right|$$

$$= \frac{1}{1-\gamma} \left| \frac{\partial \sum_{s,a} d_{\alpha'}(s) \pi'_\alpha(a|s) A_i^\alpha(s,a)}{\partial \alpha} \Big|_{\alpha=0} \right|$$

$$= \frac{1}{1-\gamma} \left| \frac{\partial \sum_{s,a} d_{\alpha'}(s) (\pi'_\alpha(a|s) - \pi_\alpha(a|s)) Q_i^\alpha(s,a)}{\partial \alpha} \Big|_{\alpha=0} \right|$$

$$\leq \frac{1}{1-\gamma} \left( \underbrace{\left| \sum_{s,a} d'_\theta(s) \frac{\partial \pi'_\alpha(a|s) - \partial \pi_\alpha(a|s)}{\partial \alpha} \Big|_{\alpha=0} Q_i^\theta(s,a) \right|}_{\text{Part A}} \right.$$

$$+ \underbrace{\left| \sum_{s,a} d'_\theta(s)(\pi'_\theta(a|s) - \pi_\theta(a|s)) \frac{\partial Q_i^\alpha(s,a)}{\partial \alpha} \Big|_{\alpha=0} \right|}_{\text{Part B}}$$

$$+ \left. \underbrace{\left| \sum_{s,a} \frac{\partial d'_\alpha(s)}{\partial \alpha} \Big|_{\alpha=0} (\pi'_\theta(a|s) - \pi_\theta(a|s)) Q_i^\theta(s,a) \right|}_{\text{Part C}} \right)$$

Thus:

$$\text{Part A} = \left| \sum_{s,a} d'_\theta(s) \frac{\partial \pi'_\alpha(a|s) - \partial \pi_\alpha(a|s)}{\partial \alpha} \Big|_{\alpha=0} Q_i^\theta(s,a) \right|$$

$$= \left| \sum_{s,a} d'_\theta(s) u_{a_i,s} (\pi_{\theta'_{-i}}(a_{-i}|s) - \pi_{\theta_{-i}}(a_{-i}|s)) Q_i^\theta(s,a) \right| \tag{47}$$

$$\leq \frac{1}{1-\gamma} \left| \sum_s d'_\theta(s) \sum_{a_i} |u_{a_i,s}| \sum_{a_{-i}} \left| \pi_{\theta'_{-i}}(a_{-i}|s) - \pi_{\theta_{-i}}(a_{-i}|s) \right| \right| \tag{48}$$

$$\leq \frac{1}{1-\gamma} \left( \max_s \sum_{a_i} |u_{a_i,s}| \right) \sum_s d'_\theta(s) 2 d_{\text{TV}}(\pi_{\theta'_{-i}}(\cdot|s) || \pi_{\theta_{-i}}(\cdot|s)) \tag{49}$$

$$\leq \frac{1}{1-\gamma} \left( \max_s \sum_{a_i} |u_{a_i,s}| \right) \sum_s d'_\theta(s) \sum_{j \neq i} 2 d_{\text{TV}}(\pi_{\theta'_j}(\cdot|s) || \pi_{\theta_j}(\cdot|s)) \tag{50}$$

$$= \frac{1}{1-\gamma} \left( \max_s \sum_{a_i} |u_{a_i,s}| \right) \sum_s d'_\theta(s) \sum_{j \neq i} \|\theta'_{j,s} - \theta_{j,s}\|_1 \tag{51}$$

$$\leq \frac{1}{1-\gamma} \sqrt{|\mathcal{A}_i|} \sum_s d'_\theta(s) \sum_{j \neq i} \sqrt{|\mathcal{A}_j|} \|\theta'_{j,s} - \theta_{j,s}\| \tag{52}$$

$$\leq \frac{1}{1-\gamma} \sqrt{|\mathcal{A}_i|} \sum_{j \neq i} \sqrt{|\mathcal{A}_j|} \sqrt{\sum_s d_{\theta'}(s)^2} \sqrt{\sum_s \|\theta'_{j,s} - \theta_{j,s}\|^2} \tag{53}$$

$$= \frac{1}{1-\gamma} \sqrt{|\mathcal{A}_i|} \sum_{j \neq i} \sqrt{|\mathcal{A}_j|} \sqrt{\sum_s d_{\theta'}(s)^2} \|\theta'_j - \theta_j\|$$

$$\leq \frac{1}{1-\gamma} \sqrt{|\mathcal{A}_i|} \sum_{j \neq i} \sqrt{|\mathcal{A}_j|} \|\theta'_j - \theta_j\|$$

$$\leq \frac{1}{1-\gamma} \sqrt{|\mathcal{A}_i|} \sum_{j=1}^n \sqrt{|\mathcal{A}_j|} \|\theta'_j - \theta_j\|,$$

where Equation (47) to Equation (48) is derived from the fact that $|Q_i^\theta(s,a)| \leq \frac{1}{1-\gamma}$. Equation (49) to Equation (50) relies on the property of total variation distance:

$$d_{\text{TV}}(\pi_{\theta'_{-i}}(\cdot|s)||\pi_{\theta_{-i}}(\cdot|s)) \leq \sum_{j \neq i} d_{\text{TV}}(\pi_{\theta'_j}(\cdot|s)||\pi_{\theta_j}(\cdot|s))$$

Equation (51) to Equation (52) is derived from:

$$\max_s \sum_{a_i} |u_{a_i,s}| \leq \sqrt{|\mathcal{A}_i|}, \quad \|u\| \leq 1$$

$$\|\theta'_{j,s} - \theta_{j,s}\|_1 \leq \sqrt{|\mathcal{A}_j|}\|\theta'_{j,s} - \theta_{j,s}\|$$

which can be immediately verified by applying Cauchy-Schwarz inequality.

Before looking into Part B, we first define $\widetilde{P}(\alpha)$ as the state-action under $\pi_\alpha$:

$$\left[\widetilde{P}(\alpha)\right]_{(s,a)\to(s',a')} = \pi_\alpha(a'|s')P(s'|s,a)$$

Then we have that:

$$\left[\left.\frac{\partial \widetilde{P}(\alpha)}{\partial \alpha}\right|_{\alpha=0}\right]_{(s,a)\to(s',a')} = u_{a'_i,s'}\pi_{\theta_{-i}}(a'_{-i}|s')P(s'|s,a)$$

For an arbitrary vector $x$:

$$\begin{aligned}
\left[\left.\frac{\partial \widetilde{P}(\alpha)}{\partial \alpha}\right|_{\alpha=0}x\right]_{(s,a)} &= \sum_{s',a'} u_{a'_i,s'}\pi_{\theta_{-i}}(a'_{-i}|s')P(s'|s,a)x_{s',a'} \\
&\leq \|x\|_\infty \sum_{s',a'} |u_{a'_i,s'}|\pi_{\theta_{-i}}(a'_{-i}|s')P(s'|s,a) \\
&= \|x\|_\infty \sum_{s'} P(s'|s,a) \sum_{a'_i} |u_{a'_i,s'}| \sum_{a'_{-i}} \pi_{\theta_{-i}}(a'_{-i}|s') \\
&\leq \|x\|_\infty \sum_{s'} P(s'|s,a)\sqrt{|\mathcal{A}_i|} \sum_{a'_{-i}} \pi_{\theta_{-i}}(a'_{-i}|s') \\
&\leq \sqrt{|\mathcal{A}_i|}\|x\|_\infty
\end{aligned}$$

Thus:

$$\left\|\left[\left.\frac{\partial \widetilde{P}(\alpha)}{\partial \alpha}\right|_{\alpha=0}x\right]_{(s,a)}\right\|_\infty \leq \sqrt{|\mathcal{A}_i|}\|x\|_\infty$$

Similarly we can define $\widetilde{P}(\alpha)'$ as the state-action under $\pi'_\alpha$, and can easily check that

$$\left\|\left[\left.\frac{\partial \widetilde{P}(\alpha)'}{\partial \alpha}\right|_{\alpha=0}x\right]_{(s,a)}\right\|_\infty \leq \sqrt{|\mathcal{A}_i|}\|x\|_\infty$$

Define:

$$M(\alpha) := \left(I - \gamma\widetilde{P}(\alpha)\right)^{-1}, \quad M(\alpha)' := \left(I - \gamma\widetilde{P}(\alpha)'\right)^{-1}.$$

Because:

$$M(\alpha) = \left(I - \gamma\widetilde{P}(\alpha)\right)^{-1} = \sum_{n=0}^\infty \gamma^n \widetilde{P}(\alpha),$$

which implies that every entry of $M(\alpha)$ is nonnegative and $M(\alpha)\mathbf{1} = \frac{1}{1-\gamma}\mathbf{1}$, this implies:

$$\|M(\alpha)x\|_\infty \leq \frac{1}{1-\gamma}\|x\|_\infty,$$

and similarly
$$\|M(\alpha)'x\|_\infty \le \frac{1}{1-\gamma}\|x\|_\infty.$$

Now we are ready to bound Part B. Because:
$$Q_i^\alpha(s,a) = e_{(s,a)}^\top M(\alpha)r_i$$
$$\implies \frac{\partial Q_i^\alpha(s,a)}{\partial\alpha} = e_{(s,a)}^\top \frac{\partial M(\alpha)}{\partial\alpha}r_i = \gamma e_{(s,a)}^\top M(\alpha)\frac{\partial\widetilde{P}(\alpha)}{\partial\alpha}M(\alpha)r_i$$
$$\implies \left|\frac{\partial Q_i^\alpha(s,a)}{\partial\alpha}\right| \le \gamma\left\|M(\alpha)\frac{\partial\widetilde{P}(\alpha)}{\partial\alpha}M(\alpha)r_i\right\|_\infty$$
$$\le \frac{\gamma}{(1-\gamma)^2}\sqrt{|\mathcal{A}_i|}$$

Thus,
$$\text{Part B} = \left|\sum_{s,a} d_\theta'(s)(\pi_\theta'(a|s)-\pi_\theta(a|s))\frac{\partial Q_i^\alpha(s,a)}{\partial\alpha}\Big|_{\alpha=0}\right|$$
$$\le \sum_{s,a} d_\theta'(s)\,|\pi_\theta'(a|s)-\pi_\theta(a|s)|\left|\frac{\partial Q_i^\alpha(s,a)}{\partial\alpha}\Big|_{\alpha=0}\right|$$
$$\le \frac{\gamma}{(1-\gamma)^2}\sqrt{|\mathcal{A}_i|}\sum_s d_\theta'(s)2d_{\text{TV}}(\pi_{\theta'}(\cdot|s)||\pi_\theta(\cdot|s))$$
$$\le \frac{\gamma}{(1-\gamma)^2}\sqrt{|\mathcal{A}_i|}\sum_s d_\theta'(s)\sum_j 2d_{\text{TV}}(\pi_{\theta_j'}(\cdot|s)||\pi_{\theta_j}(\cdot|s))$$
$$= \frac{\gamma}{(1-\gamma)^2}\sqrt{|\mathcal{A}_i|}\sum_s d_\theta'(s)\sum_j \|\theta_{j,s}'-\theta_{j,s}\|_1$$
$$\le \frac{\gamma}{(1-\gamma)^2}\sqrt{|\mathcal{A}_i|}\sum_s d_\theta'(s)\sum_{j=1}^n \sqrt{|\mathcal{A}_j|}\|\theta_{j,s}'-\theta_{j,s}\|$$
$$\le \frac{\gamma}{(1-\gamma)^2}\sqrt{|\mathcal{A}_i|}\sum_{j=1}^n \sqrt{|\mathcal{A}_j|}\sqrt{\sum_s d_\theta'(s)^2}\sqrt{\sum_s \|\theta_{j,s}'-\theta_{j,s}\|^2}$$
$$\le \frac{\gamma}{(1-\gamma)^2}\sqrt{|\mathcal{A}_i|}\sum_{j=1}^n \sqrt{|\mathcal{A}_j|}\|\theta_j'-\theta_j\|$$

Now let's look at Part C:
$$d_\alpha'(s) = (1-\gamma)\sum_{s'}\rho(s')\sum_{a'}\pi_\alpha'(a'|s')e_{(s',a')}^\top M(\alpha)'\sum_{a''}e_{(s,a'')}$$
$$\implies \frac{\partial d_\alpha'(s)}{\partial\alpha} = (1-\gamma)\left(\underbrace{\sum_{s'}\rho(s')\sum_{a'}\frac{\partial\pi_\alpha'(a'|s')}{\partial\alpha}e_{(s',a')}^\top}_{v_1^\top} M(\alpha)'\sum_{a''}e_{(s,a'')}\right.$$
$$\left.+\underbrace{\sum_{s'}\rho(s')\sum_{a'}\pi_\alpha'(a'|s')e_{(s',a')}^\top}_{v_2^\top}\frac{\partial M(\alpha)'}{\partial\alpha}\sum_{a''}e_{(s,a'')}\right)$$
$$= (1-\gamma)\left(v_1^\top M(\alpha)'+\gamma v_2^\top M(\alpha)'\frac{\partial\widetilde{P}(\alpha)}{\partial\alpha}M(\alpha)'\right)\sum_{a''}e_{(s,a'')}$$

Note that $v_1, v_2$ are constant vectors that are independent of the choice of $s$. Additionally:

$$\|v_1\|_1 = \left\|\sum_s \rho(s) \sum_a \frac{\partial \pi'_\alpha(a|s)}{\partial \alpha} e_{(s,a)}\right\|_1$$

$$= \sum_s \rho(s) \sum_a \left|\frac{\partial \pi'_\alpha(a|s)}{\partial \alpha}\right|$$

$$= \sum_s \rho(s) \sum_a |u_{a_i,s}| \pi_{\theta'_{-i}}(a_{-i}|s)$$

$$\leq \sum_s \rho(s) \sum_a |u_{a_i,s}| \leq \sqrt{|\mathcal{A}_i|}$$

$$\|v_2\|_1 = \|\sum_s \rho(s) \sum_a \pi'_\alpha(a|s) e_{(s,a)}\|_1$$

$$= \sum_s \rho(s) \sum_a \pi'_\alpha(a|s) = 1$$

Thus:

$$\text{Part C} = \left|\sum_{s,a} \frac{\partial d'_\alpha(s)}{\partial \alpha}\right|_{\alpha=0} (\pi'_\theta(a|s) - \pi_\theta(a|s)) Q_i^\theta(s,a)\right|$$

$$= (1-\gamma)\left|\left(v_1^\top M(0)' + \gamma v_2^\top M(0)' \frac{\partial \widetilde{P}(\alpha)}{\partial \alpha}\right|_{\alpha=0} M(0)'\right) \underbrace{\sum_{s,a}\sum_{a'} e_{(s,a')}(\pi'_\theta(a|s) - \pi_\theta(a|s)) Q_i^\theta(s,a)}_{v_3}\right|$$

$$\leq (1-\gamma)\left(\frac{1}{1-\gamma}\|v_1\|_1\|v_3\|_\infty + \frac{\gamma}{(1-\gamma)^2}\sqrt{|\mathcal{A}_i|}\|v_2\|_1\|v_3\|_\infty\right)$$

$$\leq \frac{\sqrt{|\mathcal{A}_i|}}{1-\gamma}\|v_3\|_\infty$$

Additionally:

$$\left|[v_3]_{(s_0,a_0)}\right| = \left|\sum_a (\pi_{\theta'}(a|s_0) - \pi_\theta(a|s_0)) Q_i^\theta(s_0,a)\right|$$

$$\leq \frac{1}{1-\gamma}\sum_a |\pi_{\theta'}(a|s_0) - \pi_\theta(a|s_0)|$$

$$= \frac{1}{1-\gamma} 2 d_{\text{TV}}(\pi_{\theta'}(\cdot|s_0)||\pi_\theta(\cdot|s_0))$$

$$\leq \frac{1}{1-\gamma} \sum_{j=1}^n 2 d_{\text{TV}}(\pi_{\theta'_j}(\cdot|s_0)||\pi_{\theta_j}(\cdot|s_0))$$

$$= \frac{1}{1-\gamma} \sum_{j=1}^n \|\theta'_{j,s} - \theta_{j,s}\|_1$$

$$\leq \frac{1}{1-\gamma} \sum_{j=1}^n \sqrt{|\mathcal{A}_j|}\|\theta'_{j,s} - \theta_{j,s}\|$$

$$\leq \frac{1}{1-\gamma} \sum_{j=1}^n \sqrt{|\mathcal{A}_j|}\|\theta'_j - \theta_j\|$$

Combining the above inequalities we get:

$$\text{Part C} \leq \frac{\sqrt{|\mathcal{A}_i|}}{1-\gamma}\|v_3\|_\infty \leq \frac{\sqrt{|\mathcal{A}_i|}}{(1-\gamma)^2} \sum_{j=1}^n \sqrt{|\mathcal{A}_j|}\|\theta'_j - \theta_j\|$$

Sum up Part A-C we get:

$$\left| \frac{\partial J_i(\theta_i' + \alpha u_i, \theta_{-i}') - \partial J_i(\theta_i + \alpha u_i, \theta_{-i})}{\partial \alpha} \right|_{\alpha=0} \le \frac{1}{1-\gamma} (\text{Part A} + \text{Part B} + \text{Part C})$$

$$\le \frac{2}{(1-\gamma)^3} \sqrt{|\mathcal{A}_i|} \sum_{j=1}^{n} \sqrt{|\mathcal{A}_j|} \|\theta_j' - \theta_j\|,$$

which completes the proof. $\square$

## M   AUXILIARY

We recall Lemma 9.

**Lemma 9.** *Let $\mathcal{X}$ denote the probability simplex of dimension $n$. Suppose $\theta \in \mathcal{X}, g \in \mathbb{R}^n$ and that there exists $i^* \in \{1, 2, \ldots, n\}$ and $\Delta > 0$ such that:*

$$\theta_{i^*} \ge \theta_i, \quad \forall i \ne i^*$$
$$g_{i^*} \ge g_i + \Delta, \quad \forall i \ne i^*.$$

*Let*

$$\theta' = Proj_{\mathcal{X}}(\theta + g),$$

*then:*

$$\theta_{i^*}' \ge \min\{1, \theta_{i^*} + \frac{\Delta}{2}\}$$

*Proof.* Let $y = \theta + g$, without loss of generality, assume that $i^* = 1$ and that:

$$y_1 > y_2 \ge y_3 \ge \cdots \ge y_n.$$

Using KKT condition, one can derive an efficient algorithm for solving $Proj_{\mathcal{X}}(y)$ (Wang & Carreira-Perpiñán, 2013), which consists of the following steps:

1. Find $\rho := \max\{1 \le j \le n : y_j + \frac{1}{j}\left(1 - \sum_{i=1}^{j} y_i\right) > 0\}$;

2. Set $\lambda := \frac{1}{\rho}\left(1 - \sum_{i=1}^{\rho} y_i\right)$;

3. Set $\theta_i' = \max\{y_i + \lambda, 0\}$.

From the algorithm, we have that:

$$\lambda = \frac{1}{\rho}\left(1 - \sum_{i=1}^{\rho} y_i\right) = \frac{1}{\rho}\left(1 - \sum_{i=1}^{\rho}(\theta_i + g_i)\right)$$

$$= \frac{1}{\rho}\left(1 - \sum_{i=1}^{\rho} \theta_i\right) - \frac{1}{\rho}\sum_{i=1}^{\rho} g_i$$

$$\ge -\frac{1}{\rho}\sum_{i=1}^{\rho} g_i.$$

If $\rho \ge 2$,

$$\theta_1' = \max\{y_1 + \lambda, 0\} \ge y_1 + \lambda \ge \theta_1 + g_1 - \frac{1}{\rho}\sum_{i=1}^{\rho} g_i$$

$$\ge \theta_1 + (1 - \frac{1}{\rho})g_1 - \frac{1}{\rho}\sum_{i=2}^{\rho}(g_1 - \Delta) = \theta_1 + \frac{\rho - 1}{\rho}\Delta \ge \theta_1 + \frac{\Delta}{2}.$$

If $\rho = 1$,

$$\theta_1' = y_1 + \lambda = y_1 + (1 - y_1) = 1.$$

Thus:

$$\theta_1' \ge \min\{1, \theta_1 + \frac{\Delta}{2}\},$$

which completes the proof. $\square$

## N    NUMERICAL SIMULATION DETAILS

**Verification of the fully mixed NE in Game 2**    We now verify that joining network 1 with probability $\frac{1-3\epsilon}{3(1-2\epsilon)}$,i.e.:

$$\pi_{\theta_i}(a_i = 1|s) = \frac{1-3\epsilon}{3(1-2\epsilon)}, \quad \forall s \in \mathcal{S}, \ \ i = 1,2,$$

is indeed a NE. First, observe that

$$\mathrm{Pr}^{\theta}(s_{i,t+1} = 1) = \left(\frac{1-3\epsilon}{3(1-2\epsilon)}\right) P(s_{i,t+1} = 1|a_{i,t} = 1) + \left(1 - \frac{1-3\epsilon}{3(1-2\epsilon)}\right) P(s_{i,t+1} = 1|a_{i,t} = 2)$$

$$= \left(\frac{1-3\epsilon}{3(1-2\epsilon)}\right)(1-\epsilon) + \left(1 - \frac{1-3\epsilon}{3(1-2\epsilon)}\right)\epsilon = \frac{1}{3}.$$

Thus,

$$V(s) = r(s) + \sum_{t=1}^{\infty} \mathbb{E}_{s_t}\gamma^t r(s_t) = r(s) + \frac{2\gamma}{3(1-\gamma)},$$

$$\overline{Q_i^\theta}(s,a_i) = r(s) + \gamma \sum_{s',a_{-i}} P(s'|a_i, a_{-i})\pi_{\theta_{-i}}(a_{-i}|s)V(s')$$

$$= r(s) + \gamma \sum_{s_i' \in \{1,2\}} (P(s_i'|a_i)\mathrm{Pr}^\theta(s_{-i} = 1)r(s_i, s_{-i} = 1) + P(s_i'|a_i)\mathrm{Pr}^\theta(s_{-i} = 2)r(s_i, s_{-i} = 2)) + \frac{2\gamma^2}{3(1-\gamma)}$$

$$= r(s) + \gamma P(s_i' = 1|a_i)\left(\frac{1}{3}r(s_i' = 1, s_{-i} = 1) + \frac{2}{3}r(s_i' = 1, s_{-i} = 2)\right)$$

$$+ \gamma P(s_i' = 2|a_i)\left(\frac{1}{3}(s_i' = 2, s_{-i} = 1) + \frac{2}{3}r(s_i' = 2, s_{-i} = 2)\right) + \frac{2\gamma^2}{3(1-\gamma)}$$

$$= r(s) + \frac{2}{3}\gamma + \frac{2\gamma^2}{3(1-\gamma)} = r(s) + \frac{2\gamma}{3(1-\gamma)} = V(s),$$

which implies that:

$$(\theta_i' - \theta_i)^\top \nabla_{\theta_i} J_i(\theta) = 0, \quad \forall \theta_i' \in \mathcal{X}_i, \quad i = 1,2,$$

i.e. $\theta$ satisfies first-order stationarity. Since $d_\theta(s) > 0$ holds for any valid $\theta$, by Theorem 1, $\theta$ is a NE.

**Computation of strict NEs in Game 2**    The computation of strict NEs is done numerically, using the criterion in Lemma 4. We enumerate over all $2^8$ possible deterministic policies and check whether the conditions in Lemma 4 hold. For $\epsilon = 0.1, \gamma = 0.95$, and an initial distribution set as:

$$\rho(s_1 = i, s_2 = j) = 1/4, \ \ i,j \in \{1,2\},$$

the numerical calculation shows there exist 13 different strict NEs.

