# OpenReview forum: "Gradient play in stochastic games: stationary points, convergence, and sample complexity"
_ICLR.cc/2022/Conference — ICLR 2022 Submitted_

### Official Review · Reviewer_NPmA · 2021-10-19

**Correctness:** 4
**Technical Novelty And Significance:** 4
**Empirical Novelty And Significance:** 3
**Recommendation:** 3
**Confidence:** 5

**Details Of Ethics Concerns:**

Reviewer ByWn raised an ethical issue which caught my attention during the discussion, and here is what he said.

> I contacted the PCs in October when I got the paper for review. In their submitted version to ICLR (deadline October 5) they included a sample complexity result. This result effectively appeared in June in Leonardos et al. Moreover, I have reviewed an earlier version of the paper submitted to Neurips 2021 where the sample complexity result was not there. It is quite obvious (the authors cite in their paper and agreed in the discussion) that their sample complexity result (in my opinion the most important result in the submitted paper) has first appeared in Leonardos et al, so effectively the reprove it. On top of that, they do not state clearly that the sample complexity result is subsequent to Leonardos et al. I exchanged multiple messages in openreview with them, asking for proper citation; they have a footnote claiming that the results between their paper and Leonardos et al was in parallel (they tried to defend themselves but felt that they did not want to give proper credit). You can see the full discussion.

After I read Leonardo's paper, I feel the reasoning holds. Therefore I change my score from **8** to **3**


**Main Review:**

* I really like the content in Appendix A, which provided numerical evidence to justify the effectiveness of the proposed methods. This kind of analysis offers particular value to the MARL community, unlike many existing theoretical MARL papers.



* There are existing work that set up Markov Potential Game, can the author clarify the difference to

    **Learning in Nonzero-Sum Stochastic Games with Potentials  David Mguni et al. ICML 21**

    **When Can We Learn General-Sum Markov Games with a Large Number of Players Sample-Efficiently? Song et al**


* Since this paper deals with SGs, can the author comment on the relationship between the Nash equilibrium in Definition 1 and Markov  Perfect Equilibrium?


* Through gradient dominance assumption, the author equates NE and first-order stationary point, can the author clarify if this result contradicts with the result in

    **On gradient-based learning in continuous games. Mazumdar 2020**



* For the linear dependency result on Markov Potential Game, can the author clarify the significane of your result versus

    **When Can We Learn General-Sum Markov Games with a Large Number of Players Sample-Efficiently? Song et al**


**Summary Of The Paper:**

This paper studies the optimization landscape of first-order methods in stochastic games. In particular, it studies the connections between first-order stationary points and Nash equilibrium and its convergent property and learning stability. For general-sum SGs, local convergence results is provided. For Markov potential games, global convergence results are provided.

**Summary Of The Review:**

  This paper is a rather long paper and certainly over-exceeding the content that a conference paper can hold. Given the number of pages, it is challenging to evaluate for the correctness of proof for a 50-page paper within such a short time period. However, I believe the result is generally correct, for example, the sample complexity on Markov Potential Games has been discovered, probably in parallel, by many other papers. My major concern is how this paper poses itself among the peer work and if the result is still significant.

---

> ### Author Response · Authors · 2021-11-16
> **Response to Reviewer NPmA**
>
> We sincerely thank the reviewer for the support of our paper and the valuable feedback and questions! We have added the discussions into the revised version of our paper.
>
> [Comparison with [Mguni et al.]] Thank you for bringing up this recent work that considers Markov potential game. We have added this reference to the revised version of the paper. Indeed both *Mguni et al.* and our paper focus on learning for Markov potential games, however, the settings, algorithms as well as contributions are actually very different between the two papers. First of all, our paper studies general SGs besides MPG and gives characterization of the local geometry of NEs, which is not considered in the paper by *Mguni et al.*.  Secondly, regarding MPG, we give non-asymptotic convergence rate as well as sample complexity analysis which are not provided in *Mguni et al.*. Thirdly, if we understood it correctly, the learning algorithm considered in their paper requires to store a full table of $\{a_1,a_2,\dots,a_n\}$ and thus need to be done in a centralized manner, while our algorithm can be done in a fully decentralized manner. Lastly, the conditions that we derive for MPG are actually different. We have a brief discussion about the Markov potential game in Appendix B and footnote 2. Specifically in Appendix B.1 (Proof of proposition 2), a counterexample is provided where it satisfies the state transitivity assumption in *Mguni et al.* while it is not a MPG. We can use this counterexample to show that some propositions (e.g., Proposition 4 in Appendix H.1 in *Mguni et al.* would need more consideration. For the above reasons, we argue that although we consider similar settings, our work is very different from *Mguni et al.* and have different contributions.
>
> [Comparison with [Song et al.]] We acknowledge that we are not aware of their work when submitting this paper. Our first online version was posted in early June while their paper was posted online by October 8th which is after the submission deadline of this conference, which is why we didn't cite that in our submission. Authors of this paper actually communicated with us recently and mentioned that they got inspirations from our online paper when working on their submission. Indeed, their paper cited our paper as one of their references. We do agree that their work provides great contribution to the learning in stochastic game area. And we have also added their work in our reference in the revised version of the paper to further acknowledge their contributions. Now regarding the comparison of these two papers,  the two papers consider different settings and have different contributions.  Our paper studies the optimization landscape for general stochastic games as well as convergence rate and sample complexity of converging to a Nash equilibrium for first order methods in the setting of Markov potential game. Besides sample complexity results, we also provide some characterization of the local geometry of some Nash equilibria in the stochastic game settings. On the other hand, *Song et al.* focuses more on the sample complexity of finding correlated equilibria (instead of Nash equilibrium) for the finite time-horizon stochastic games. Although they also consider the sample complexity of finding a NE for the MPG setting, in their learning algorithm agents need to update their policies in a one-by-one asynchronous fashion, thus additionally consensus effort is needed to broadcast who can update in each epoch, which is different from our learning algorithm where agents update at the same time. We personally enjoy reading their paper and it also inspires us to look into the problem from a different perspective, however the two papers are considering different aspects of the multi-agent learning problem and the contributions are different.
>
> [Markov perfect Nash equilibrium] If we consider the definition of Markov Perfect Equilibrium (MPE) to be a subgame perfect NE at every state $s$, and that all players use Markov strategies (*Markov perfect equilibrium: I. Observable actions, Maskin and Tirole, 2001*), i.e.,$$V^{\theta_i^*,\theta_{-i}^*}_i(s) \ge V^{\theta_i',\theta^* _{-i}}(s), \forall s,~~\forall~\theta_i',$$
> then yes, we expect our results to still hold if we change the notion of NE to Markov Perfect Equilibrium (under Assumption 1), i.e., the notion of first order stationary point, NE and MPE are equivalent under Assumption 1. We thank the reviewer for pointing this out.

---

> > ### Author Response · Authors · 2021-11-16
> > **Response to Reviewer NPmA**
> >
> > [Comparison with the result in [Mazumdar et al.]] As the reviewer has suggested, the key difference is the gradient domination property. In the counterexample constructed in [Mazumdar et al.], e.g. the counterexample provided in the proof of Proposition 6, the cost function $f_2$ does not satisfy gradient domination property (when fixing $x_1$, the stationary point $\nabla_{x_2} f_2 = 0$ is not a minimizer of $f_2$ and thus gradient domination is not satisfied). If the cost function satisfies gradient domination property in these counterexamples,  the equivalence between first order stationarity and NE is expected. On the other hand, our paper considers finite states and finite actions (i.e., tabular MDP) along with direct policy parameterization. As shown in Lemma 3, the gradient domination property holds. Lastly, we want to remark that our result shows the equivalence between the two notions, but not all the stationary points are locally asymptotic stable (e.g. the fully mixed stationary points), thus it does not contradict with the stability results in [Mazumdar et al.]. Hope that our explanation helps to clarify your question.

---

> ### Comment · Reviewer_NPmA · 2021-12-01
> **All of my questions are clarified, I recommend for acceptance.**
>
> Thanks to the author for providing detailed feedback. I am satisfied by the answers, especially given most of my questions are mainly regarding to the comparisons to the relevant work that came out this year. The author makes it very clear their novelty.
>
> This is undoutedbly a solid work, it will play an important role in MARL theory, I would recommend for acceptene.

---

> > ### Author Response · Authors · 2021-12-01
> > **Further response to Reviewer NPmA**
> >
> > We sincerely thank the reviewer for appreciating our work! Glad that our response clarifies your questions!

---

> ### Comment · Reviewer_NPmA · 2021-12-02
> **Downgrading to Rejection due to Potential Ethical Issue**
>
> Dear Author:
>
> After reading other reviewers' comments, it is clear to me that there might be an ethical issue involved in this paper, particularly the connection to **Leonardo**'s paper. I, therefore, downgrade my recommendation to rejection. This is only my own opinion, the author may want to provide more evidence directly to AC or SAC.

---

> > ### Author Response · Authors · 2021-12-02
> > **Sorry for the mis-communication**
> >
> > Dear Reviewer,
> >
> > We apologize for the miscommunication. In the last round of revision, we thought we clarified/explained the meaning of "parallel" in the comment.  After receiving the comment from Reviewer ByWn, we realized that putting the "parallel" there was not right because it made people thought about the sample complexity results were parallel.  So we immediately revised the discussion: we removed the parallel claim and also added "inspired by their work". (These discussions were happening before this comment). Here is the revised quote:
> >
> > "There are recent arXiv preprints (Mguni, 2020; Leonardos et al., 2021; Mguni et al., 2021) studying MPGs that are similar to our MPG setting. In particular, Leonardos et al. (2021) also studies gradient play for MPG. Both of our papers share similar results on MPGs but the sample-based methods are designed from different perspectives. \footnote{ Leonardos et al. (2021) considers Monte Carlo, model-free gradient estimation. The sample complexity is derived under the condition that the estimation is unbiased, which is difficult to hold in general. Inspired by their work, we consider incorporating some model-based concepts in designing our learning methods. Interestingly, both sample complexities are O(1/epsilon^6). It is an interesting question to study what is the fundamental sample complexity for MPG.}"
> >
> > We would like to apologize for leaving such an impression, which is not what we intended to do at all. We sincerely hope our mis-communication did not leave reviewers and readers such an impression.

---

> > ### Author Response · Authors · 2021-12-02
> > **System does not allow resubmission/revision after Nov 22--Clarification of the revision**
> >
> > Dear Reviewer,
> >
> > After seeing your quote from Reviewer ByWn, we felt that there might be a major missunderstanding that causes all the miscommunication and the inappropriate ethics accusation.
> >
> > After receiving Reviewer ByWn's first message on "the revised version has not included my remarks", we have taken immediate action (see the logs of our responses in the Openreview system). Unfortunately, the system does not allow resubmission after Nov 22, which was why the PDF on the openreview was still having the old version, making Reviewer ByWn and you feel that we were reluctant to change and we were hiding anything. To make it more clear, the latest version of explaining the connection and difference between the two papers is quoted in the next paragraph. The new update can also be checked on our ArXiv version (we updated the revision today and the new version should be shown in 2-3 days). We want to firmly assure the reviewer(s) that it was not because we did not want to give credit to Leonardos et al 2021. On the opposite, we appreciate their work very much and are glad that some group(s) are interested in similar problems. In fact, we have been communicating with the authors after ICLR submission on exchanging technical ideas. We quote a sentence from their email, "I saw your similar paper "Gradient play in stochastic games: stationary points, convergence, and sample complexity" and also think it has really interesting results and approaches different from ours."
> >
> > Since the ArXiv needs a couple of days to reflect the new update, here is the quote of the latest version of the connection and comparison: "There are recent arXiv preprints (Mguni, 2020; Leonardos et al., 2021; Mguni et al., 2021) studying MPGs that are similar to our MPG setting. In particular, Leonardos et al. (2021) also studies gradient play for MPG. Both of our papers share similar results on MPGs but the sample-based methods are designed from different perspectives. We also would like to clarify that though results on the exact policy gradient of MPG were done in parallel for these two papers, Leonardos et al. (2021) had a sample complexity algorithm and analysis earlier than this paper. Leonardos et al. (2021) considers Monte Carlo, model-free gradient estimation. The sample complexity is derived under the condition that the estimation is unbiased, which is difficult to hold in general. On the other hand, we consider incorporating some model-based concepts in designing our learning methods. Interestingly, both sample complexities are O(1/epsilon^6) despite both methods and analyses being very different. It is an interesting question to study what is the fundamental sample complexity for MPG. In addition, our paper studies general SGs besides MPG. Moreover, our concept of “averaged” MDPs could also serve as a useful tool for the design and analysis of other MARL algorithms."
> >
> > If this discussion is still unclear, we are happy to take your advice.

---

> > > ### Author Response · Authors · 2021-12-09
> > > **update**
> > >
> > > Just want to give an update. The Arxiv version was updated. Since we can't provide that link, here is also an anonymous link to an anonymous version of the paper: https://anonymous.4open.science/r/ICLR2021-anonymous-5F93/MARL_sample_based_ICLR2022__Arxiv_version2_%20(1).pdf

---

### Official Review · Reviewer_oPGh · 2021-11-02

**Correctness:** 4
**Technical Novelty And Significance:** 3
**Empirical Novelty And Significance:** Not applicable
**Recommendation:** 6
**Confidence:** 4

**Main Review:**

Strengths:

They study general Stochastic games with multiple players (more than two) and a provide a local characterization of the strict NE which is novel, as it applies for general stochastic games with no assumption. Although, here the gradient of each player does seem to depend on the information from other players.
They provide a sample-based RL algorithm with non-asymptotic guarantees for Markov Potential Games, with fully independent gradient updates for each players.


Weaknesses:
The issue is that the existence of Strict Nash for Stochastic Games maybe a strong assumption. This may be quite rare even when considering normal form games.

Given the result of [Daskalakis et al. 2021] (which is not cited in this paper), that show that independent RL (with a two time scale approach) for two player Stochastic Zero-Sum games converges to the epsilon-Nash policy globally with a non-asmyptotic convergence rate. It would have been interesting to see more specific characterization of classes of Stochastic Games that possibly can lead to a similar convergence guarantees. For instance, it might be useful to look at separable zero-sum polymatrix games [Cai and Daskalakis 2014], a subclass of multiplayer games that accommodate efficient computation of NE through linear programs. Even in normal form games if one applies no-regret online algorithms for two-player zero-sum games, the last-iterate behavior might be complicated [Bailey and Piliouras 2018, Mertikopoulos et al. 2018]. It is possible that some of these complications may carry over to the MARL settings.

Also, it would be interesting to understand possibly other notions such as Coarse Correlated Equilibrium for general settings of Stochastic Games, instead of relying purely on the Nash equilibrium.

Finally it would be useful if the authors could differentiate between the results of [Leonardos et al 2021] for Markov Potential Games as both algorithms achieve the same sample complexity and other results are mostly similar. Maybe it would be useful if they discuss algorithmic and computational differences.

References:

Daskalakis, Constantinos, Dylan J. Foster, and Noah Golowich. "Independent policy gradient methods for competitive reinforcement learning." arXiv preprint arXiv:2101.04233 (2021).

Cai, Yang, et al. "Zero-sum polymatrix games: A generalization of minmax." Mathematics of Operations Research 41.2 (2016): 648-655.

Leonardos, Stefanos, et al. "Global Convergence of Multi-Agent Policy Gradient in Markov Potential Games." arXiv preprint arXiv:2106.01969 (2021).

Mertikopoulos, Panayotis, Christos Papadimitriou, and Georgios Piliouras. "Cycles in adversarial regularized learning." Proceedings of the Twenty-Ninth Annual ACM-SIAM Symposium on Discrete Algorithms. Society for Industrial and Applied Mathematics, 2018.

Bailey, James P., and Georgios Piliouras. "Multiplicative weights update in zero-sum games." Proceedings of the 2018 ACM Conference on Economics and Computation. 2018.



**Summary Of The Paper:**

The authors study independent gradient play in Stochastic Games, with multiple players/agents. They characterize the (local) stability of strict Nash in Stochastic Games. In addition they study Markov Potential Games, a subclass of Stochastic Games and show global convergence of projected gradient ascent for direct paramterization of Tabular MDPs. In addition, they also provide a sample-based RL algorithm, where the players do not use gradient information from others and estimate based on the observed trajectories. They show non-asymptotic convergence guarantees of this algorithm for Tabular MDPs which are Markov Potential Games.

**Summary Of The Review:**

The paper has some interesting results for very generic Stochastic Game settings and Markov Potential Games (very similar to [Leonardos et al. 2021]). But I think these settings are too generic to study and extract meaningful results. Given the above comments on the strengths and weaknesses, it seems that there is more investigation and characterization that might be required.

After the response:
The authors have clarified most of the concerns and the characterization seems adequate for this work to be published.

---

> ### Author Response · Authors · 2021-11-16
> **Response to Reviewer oPGh**
>
> We sincerely thank the reviewer for the critical feedback and questions! We address your comments and questions in the review as follows. We have added the following discussions to the revised version of the paper.
>
> [Existence of strict NE] Apologize for the confusion. We agree with the reviewer that the existence of strict NE is not necessarily true for general-sum games, however, we want to clarify that results in the paper, in particular Theorem 2 and 4, do not claim to solve the existence of strict NEs or to find these strict NEs via gradient play. The way to interpret Theorem 2 is that, if there exists a strict NE, it is locally stable under gradient play. However outside the local contraction region, we cannot say much. Thus, as claimed in the paper, Theorem 2 studies the local geometry of strict NE if there is a strict NE, instead of global existence or convergence. Similarly, for Theorem 4, the way to interpret it is that if there exists a strict NE, it is a strict local maximum of the total potential function.  In the revised version, we add a statement to clarify the interpretation of Theorem 2 and 4.
>
> [Results from Zero-sum games] Thank you for bringing up the series of studies in zero-sum games. We were aware of some of the progress made in zero-sum games. We agree that it is worthwhile looking into whether the results could have interesting generalization to the general-sum multi-agent games. We have been thinking about this line while discussing with people who are working on zero-sum games. Some caveat is that the analysis for the performance of gradient based algorithm in this paper is mainly done in the MPG setting, which is different from zero-sum game. Thus the analysis techniques and the interpretation of the results are different, and the properties and difficulties mentioned in zero-sum games might not directly extend to the MPG setting. We have added relevant references and discussions into the introduction section of the revised version of the paper.
>
> [Other notions of equilibrium, e.g. coarse correlated NE] We thank the reviewer for pointing out this possible direction! This is actually our ongoing  work and there are some recent exciting work in this direction (E.g. *When Can We Learn General-Sum Markov Games with a Large Number of Players Sample-Efficiently? Song et al.*, see also discussion with Reviewer NPmA). NE is a stronger notion compared with correlated NE and is generally harder to learn. On the other hand it can be shown that coarse correlated NE can be learnt efficiently using online learning methods such as following the regularized leader. We agree with the reviewer that learning the NE is hard for general-sum stochastic game and it will be an interesting future direction to look into learning of correlated equilibrium in general-sum stochastic games.
>
> [Comparison with the results of [Leonardos et al. 2021]] We have provided a brief discussion and comparison of our work and the work by *Leonardos et al.* in the last paragraph of Section 1 of our paper (see footnote 1 on page 2). Before explaining the difference, we want to clarify that most of this work was done ''in parrallel'' with *Leonardos et al. 2021*. Our first version of the paper was finished in May (which was posted online in early June) and then we found out the above paper on ArXiv (if we were right, the first arXiv version was posted in early June). So most of the results were developed in parallel. Now regarding the different contributions between the two papers. First of all, our papers studies general SGs besides MPG and gives characterization of the local geometry of NEs, which is not considered in the other paper. Secondly, regarding MPG, although our results are similar for the exact gradient play, the sample-based methods considered in the two papers are designed from very different perspectives. *Leonardos et al.* consider the Monte Carlo, model-free gradient estimation, while we consider estimating the gradient by estimating the Q function and discounted state visitation distribution, which suffers less from the large variance caused by Monte Carlo estimation. Further, the result in *Leonardos et al.* is derived under the condition that the Monte Carlo estimation is unbiased, which is difficult to hold in general and need more careful consideration and examination on when the condition holds. Thirdly, the two papers provide different conditions for MPG. Besides sufficient conditions, in Appendix B of our paper we also give necessary conditions, counter-example, as well as a concrete practical application example (the medium access control) of MPG. Thus the contributions of the two works are different.

---

> > ### Author Response · Authors · 2021-11-16
> > **Response to Reviewer oPGh**
> >
> > [Justification of the settings considered] The reviewer mentioned in the summary of the review that the 'settings are too generic to study and extract meaningful results'. The stochastic game setting considered in this paper is a standard model considered in both game theory as well as multi-agent RL and there are a lot of works in literature in this area, e.g. *Decentralized Q-learning for stochastic teams and games. Arslan et al. 2016*, *Last-Iterate Convergence: Zero-Sum Games and Constrained Min-Max Optimization. Daskalakis et al. 2018* ,*When Can We Learn General-Sum Markov Games with a Large Number of Players Sample-Efficiently? Song et al, 2021* . Specifically for the MPG setting, we also provide some application examples of MPG (Appendix B.3) as well as conditions that imply MPG (Appendix B.1 and B.2) in the supplementary material of our paper. Thus we believe that the settings considered in our paper are meaningful and can bring some insights to the algorithm design and real-life applications of multi-agent RL. But we also agree with the reviewer that to make the research more meaningful and impactful, more work should be devoted into investigating various applications and abstracting useful model and problem properties. We have added this into the conclusion section in our revised version of the paper.

---

> > > ### Comment · Reviewer_oPGh · 2021-11-29
> > > **Acknowledgement of author's response**
> > >
> > > Thank you for your detailed clarification to the questions and detailing the differences from Leonardos et al 2021. I believe the response has clarified most of the concerns. I would request the authors to add the above discussion and relevant references. I will raise my score in light of this response.

---

> > > > ### Author Response · Authors · 2021-12-01
> > > > **Further response to Reviewer oPGh**
> > > >
> > > > We sincerely thank the reviewer for supporting our paper! We have already added some of the discussions and references into the current revised version of our paper. We will discuss the literature and comparison with related references more thoroughly in the next round of revision.

---

### Official Review · Reviewer_ByWn · 2021-11-02

**Correctness:** 4
**Technical Novelty And Significance:** 3
**Empirical Novelty And Significance:** Not applicable
**Recommendation:** 8
**Confidence:** 5

**Details Of Ethics Concerns:**

I mentioned that the sample complexity results are not in parallel and are subsequent to the paper https://arxiv.org/abs/2106.01969 . After SAC/AC/reviewer discussion. It was concluded that there was no ethics issue.

**Main Review:**

The main idea is to use the gradient domination property established in Agarwal et al., 2020 and argue that stationary points of the potential function are indeed approximate Nash policies. Then using techniques from non-convex optimization, it is known that Gradient Descent converges to stationary points and this is what they show about policy gradient on the potential function. The analysis of the stochastic variant is much more challenging, it needs different parametrization than direct parametrizatoin. In general, i feel the paper has made a valid contribution on multi-agent RL in potential games.

**Summary Of The Paper:**

The paper studies the global convergence of policy gradient for multiagent Markov potential games. The authors define a notion of potential markov games that seems natural and show convergence to Nash policies when the agents use gradient ascent independently on their policies (using direct parametrization). Observe that the authors assume that the agents have full knowledge of their utilities and their derivatives (their algorithm is deterministic). They also provide convergence for the stochastic variant of policy gradient (given samples).
They moreover show that deterministic Nash policies always exist and show that fully mixed policies are saddle points. Saddle points might be of higher order, so first order methods might fail to avoid the fully mixed policies because you need to check third (or higher) order derivatives (so techniques from Ge et al do not help to show avoidance as claimed, the authors need to remove that comment since Ge et al argue about second order stationarity).

**Summary Of The Review:**

I feel the paper has solid contributions, they address both deterministic and stochastic variant of Policy gradient. The stochastic variant is more challenging due to the limitation of the non-Lipschitz gradient.

---

> ### Author Response · Authors · 2021-11-16
> **Response to Reviewer ByWn**
>
> We sincerely thank the reviewer for the support of our paper!
>
> In response to reviewer's comments about higher order stationary points, we agree with the reviewer that we need to check the second order condition to determine whether it is a non-degenerate saddle point in order for Ge et al.'s results to apply. And indeed the fully mixed NEs can be degenerated and escaping these saddle points is more challenging.  We have removed the statement in the revised version of the paper. We thank the reviewer for pointing this out.
>
> In response to the reviewer's concern that the sample complexity results are not in parallel but are subsequent to *Global Convergence of Multi-Agent Policy Gradient in Markov Potential Games, Leonardos et.al.*, we apologize for the confusion. By ''parallel'' we meant that our first version of the paper was finished in May (which was posted online in early June) and then we found out the above paper  [Leonardos et.al.] on ArXiv (if we were right, the first arXiv version was also posted in early June). So most of the results were developed in parallel. But we agree that the paper by *Leonardos et.al.* had sample-based algorithms and sample complexity analysis ahead of us. The sample-based algorithms in their paper and our paper are developed from different perspectives (see more discussion in the last paragraph of Section 1 in our paper).  We have made some modification in the revised version of our paper to clarify this and further acknowledge their contribution.

---

> > ### Comment · Reviewer_ByWn · 2021-11-30
> > **The revised version has not included my remarks**
> >
> > Dear authors,
> >
> > In your revised paper you have not clarified that your results on the sample complexity are subsequent to Leonardos et al paper as you promised to do. Please make this clear in the next revision.

---

> > > ### Author Response · Authors · 2021-12-01
> > > **Further response to Reviewer ByWn**
> > >
> > > We apologize for not making it very clear in the previous revision. In the next round of revision, we will delete the "in parallel" claim. We quote the revised footnote on page 2 here: "Leonardos et al. (2021) considers Monte Carlo, model-free gradient estimation. The sample complexity is derived under the condition that the estimation is unbiased, which is difficult to hold in general. Interestingly, both sample complexities are $O(1/\epsilon^6 )$. It is an interesting question to study what is the fundamental sample complexity for MPG."

---

> > > > ### Comment · Reviewer_ByWn · 2021-12-01
> > > > **It is important to give proper credit**
> > > >
> > > > Dear authors,
> > > >
> > > > From technical perspective, I appreciate the footnote you added. Nevertheless, it is quite obvious that your complexity result is subsequent and this needs to be added explicitly. You need to understand that Leonardos et al paper is on arxiv since June, having the sample complexity results 3 months! before you submitted your ICLR paper and this is a sensitive matter for ethical reasons. Moreover, this is Openreview and our discussions are visible to everybody. Thank you.
> > > >
> > > > Best,
> > > > Reviewer

---

> > > > > ### Author Response · Authors · 2021-12-01
> > > > > **Thanks for the comment**
> > > > >
> > > > > Dear Reviewer,
> > > > >
> > > > > We agree with your comment and thank you for helping us clarify it. Hope by removing the "parallel" claim, we clarify the confusion that we caused before. Here we quote the entire related discussion that will be in the revision:
> > > > >
> > > > > "There are recent arXiv preprints (Mguni, 2020; Leonardos et al., 2021; Mguni et al., 2021) studying MPGs that are similar to our MPG setting. In particular, Leonardos et al. (2021) also studies gradient play for MPG. Both of our papers share similar results on MPGs but the sample-based methods are designed from different perspectives. \footnote{ Leonardos et al. (2021) considers Monte Carlo, model-free gradient estimation. The sample complexity is derived under the condition that the estimation is unbiased, which is difficult to hold in general. Inspired by their work, we consider incorporating some model-based concepts in designing our learning methods. Interestingly, both sample complexities are O(1/epsilon^6). It is an interesting question to study what is the fundamental sample complexity for MPG.}"
> > > > >
> > > > > Hope we have made it clear that their sample complexity results are ahead of this paper. We should have only stayed with the technical discussion from the beginning. Thank you again!
> > > > >
> > > > > Best,

---

> > > ### Author Response · Authors · 2021-12-02
> > > **System does not allow resubmission/revision after Nov 22--Clarification of the revision**
> > >
> > > Dear Reviewer,
> > >
> > > After seeing a quote from Reviewer NPmA about a discussion between the reviewers, we felt that there might be a major missunderstanding that causes all the miscommunication and the inappropriate ethics accusation.
> > >
> > > After receiving your first message on "the revised version has not included my remarks", we have taken immediate action (see the logs of our responses in the Openreview system). Unfortunately, the system does not allow resubmission after Nov 22, which was why the PDF on the openreview was still having the old version, making you feel that we were reluctant to change and we were hiding anything. To make it more clear, the latest version of explaining the connection and difference between the two papers is quoted in the next paragraph. The new update can also be checked on our ArXiv version (we updated the revision today and the new version should be shown in 2-3 days). We want to firmly assure the reviewer(s) that it was not because we did not want to give credit to Leonardos et al 2021. On the opposite, we appreciate their work very much and are glad that some group(s) are interested in similar problems. In fact, we have been communicating with the authors after ICLR submission on exchanging technical ideas. We quote a sentence from their email, "I saw your similar paper "Gradient play in stochastic games: stationary points, convergence, and sample complexity" and also think it has really interesting results and approaches different from ours."
> > >
> > > Since the ArXiv needs a couple of days to reflect the new update, here is the quote of the latest version of the connection and comparison: "There are recent arXiv preprints (Mguni, 2020; Leonardos et al., 2021; Mguni et al., 2021) studying MPGs that are similar to our MPG setting. In particular, Leonardos et al. (2021) also studies gradient play for MPG. Both of our papers share similar results on MPGs but the sample-based methods are designed from different perspectives. We also would like to clarify that though results on the exact policy gradient of MPG were done in parallel for these two papers, Leonardos et al. (2021) had a sample complexity algorithm and analysis earlier than this paper. Leonardos et al. (2021) considers Monte Carlo, model-free gradient estimation. The sample complexity is derived under the condition that the estimation is unbiased, which is difficult to hold in general. On the other hand, we consider incorporating some model-based concepts in designing our learning methods. Interestingly, both sample complexities are O(1/epsilon^6) despite both methods and analyses being very different. It is an interesting question to study what is the fundamental sample complexity for MPG. In addition, our paper studies general SGs besides MPG. Moreover, our concept of “averaged” MDPs could also serve as a useful tool for the design and analysis of other MARL algorithms."
> > >
> > >  If this discussion is still unclear, we are happy to take your advice.

---

> > > > ### Author Response · Authors · 2021-12-09
> > > > **update**
> > > >
> > > > Just want to give an update. The Arxiv version was updated. Since we can't provide that link, here is also an anonymous link to an anonymous version of the paper:
> > > > https://anonymous.4open.science/r/ICLR2021-anonymous-5F93/MARL_sample_based_ICLR2022__Arxiv_version2_%20(1).pdf

---

### Decision · Program_Chairs · 2022-01-20

**Decision:**

Reject

**Comment:**

This paper looks at stochastic and Markov potential games. Its different results, including the sample complexity ones, are overall interesting and relevant for the community.

This said, we had an intense discussion as several of the aforementioned results -  actually, closely related results, not the exact same one - already appeared elsewhere (in a ArXiv preprint, that has been publicly submitted at a previous conference). We do believe that there is no ethical/plagiarism issue here, however, it remained the question of "paternity" of these results.

We have decided to give the paternity of the sample complexity to the first paper (the ArXiv preprint) that proved it. We can therefore only credit to this paper the improvements in the sample complexity results (as they are not exactly similar).

However, this had an impact on the reviewers (and also mine and the SAC one) evaluation of the paper, when some substantial parts of the paper are "discarded".

Nonetheless, we think that this paper has its merits, although it does not reach the ICLR bar in its current form. We strongly encourage the authors to work on a revised version - incorporating the different comments of the reviewers - and to resubmit it at a further venue.